# Observation of quantum entanglement with top quarks at the ATLAS detector

The ATLAS Collaboration*✉

Entanglement is a key feature of quantum mechanics[1–3], with applications in fields such as metrology, cryptography, quantum information and quantum computation[4–8]. It has been observed in a wide variety of systems and length scales, ranging from the microscopic[9–13] to the macroscopic[14–16]. However, entanglement remains largely unexplored at the highest accessible energy scales. Here we report the highest-energy observation of entanglement, in top–antitop quark events produced at the Large Hadron Collider, using a proton–proton collision dataset with a centre-of-mass energy of $\sqrt{s} = 13$ TeV and an integrated luminosity of 140 inverse femtobarns (fb)$^{-1}$ recorded with the ATLAS experiment. Spin entanglement is detected from the measurement of a single observable $D$, inferred from the angle between the charged leptons in their parent top- and antitop-quark rest frames. The observable is measured in a narrow interval around the top–antitop quark production threshold, at which the entanglement detection is expected to be significant. It is reported in a fiducial phase space defined with stable particles to minimize the uncertainties that stem from the limitations of the Monte Carlo event generators and the parton shower model in modelling top-quark pair production. The entanglement marker is measured to be $D = -0.537 \pm 0.002$ (stat.) $\pm 0.019$ (syst.) for 340 GeV $< m_{t\bar{t}} <$ 380 GeV. The observed result is more than five standard deviations from a scenario without entanglement and hence constitutes the first observation of entanglement in a pair of quarks and the highest-energy observation of entanglement so far.

Particle colliders, such as the Large Hadron Collider (LHC) at CERN, probe fundamental particles and their interactions at the highest energies accessible in a laboratory, exceeded only by astrophysical sources. Beyond the fundamental interest of exploring quantum entanglement in a new setting, this observation demonstrates the potential of using high-energy colliders, such as the LHC, as tools for testing our fundamental understanding of quantum mechanics. Hadron colliders offer a truly relativistic environment and provide a rich variety of fundamental interactions, rarely considered for experiments in quantum information. Relativistic effects are expected to play a critical part in quantum information[17] and the measurement described here illustrates the potential for new approaches to explore these effects and other foundational problems in quantum mechanics using colliders.

Recently, the heaviest fundamental particle known to exist, the top quark, was proposed as a new laboratory to study quantum entanglement and quantum information[18,19]. In this Article, the spin correlation between the top quark and antitop quark is used to probe the effects of quantum entanglement, in proton–proton ($pp$) collision events recorded with the ATLAS detector with a centre-of-mass energy of 13 TeV. Entanglement is observed with a significance of more than five standard deviations for the first time in pairs of quarks.

If two particles are entangled, the quantum state of one particle cannot be described independently of the other. The simplest example of an entangled system involves a pair of quantum bits (qubits); pieces of quantum information about two particles in the same quantum state that exist in superposition. The spin quantum number of a fundamental fermion, a particle that can take spin values of ±1/2, is one of the simplest and most fundamental examples of a qubit. Among the fundamental fermions of the standard model of particle physics, the top quark is uniquely suited for high-energy spin measurements because of its unique properties: its immense mass gives it a lifetime (about $10^{-25}$ s) notably shorter than the timescale needed for the quantum numbers of a quark to be shrouded by hadronization (around $10^{-24}$ s) and spin decorrelation (approximately $10^{-21}$ s) effects[20]. As a result, its spin information is transferred to its decay products. This unique feature provides an opportunity to study a pseudo-bare quark, free of the colour-confinement properties of the strong force that shrouds other quarks.

Quarks are most commonly produced in hadron collider experiments as matter–antimatter pairs. A pair of top–antitop quarks ($t\bar{t}$) is a two-qubit system in which the spin quantum state is described by the spin density matrix $\rho$:

$$\rho = \frac{1}{4}\left[ I_4 + \sum_i \left(B_i^+ \sigma^i \otimes I_2 + B_i^- I_2 \otimes \sigma^i\right) + \sum_{i,j} C_{ij} \sigma^i \otimes \sigma^j \right].$$

The first term in the linear sum is a normalization constant, where $I_n$ is the $n \times n$ identity matrix. The second term describes the intrinsic

*A list of authors and their affiliations appears online. ✉e-mail: atlas.publications@cern.ch

polarization of the top and the antitop quarks, where $\sigma^i$ are the corresponding Pauli matrices and the real numbers $B_i^\pm$ characterize the spin polarization of each particle. The third term describes the spin correlation between the particles, encoded by the spin correlation matrix $C_{ij}$. In all expressions, an orthogonal coordinate system is represented by the indices $i, j = 1, 2, 3$.

At hadron colliders, $t\bar{t}$ pairs are produced mainly by the strong interaction and thus have no intrinsic polarization (that is, $B_i^\pm \simeq 0$) because of parity conservation and time invariance in quantum chromodynamics (QCD)[21]. However, the spins of these pairs are expected to be correlated, and this correlation has already been observed by both the ATLAS and CMS experiments at the LHC[22-26]. Entanglement in top-quark pairs can be observed by an increase in the strength of their spin correlations.

Owing to their short lifetime, top quarks cannot be detected directly in experiments. In the standard model, the top quarks decay almost exclusively into a bottom quark and a $W$ boson, and the $W$ boson subsequently decays into either a pair of lighter quarks or a charged lepton and a neutrino. In this measurement, only $W$ bosons decaying into leptons are considered because charged leptons, especially electrons and muons, are readily detected with high precision at collider experiments. To a good approximation, the degree to which the leptons carry the spin information of their parent top quarks is 100% because of the maximally parity-violating nature of the electro-weak charged current. The angular direction of each of these leptons is correlated with the direction of the spin of their parent top quark or antitop quark in such a way that the normalized differential cross-section ($\sigma$) of the process may be written as[27]

$$\frac{1}{\sigma}\frac{d\sigma}{d\Omega_+ d\Omega_-} = \frac{1 + \mathbf{B}^+ \cdot \hat{\mathbf{q}}_+ - \mathbf{B}^- \cdot \hat{\mathbf{q}}_- - \hat{\mathbf{q}}_+ \cdot C \cdot \hat{\mathbf{q}}_-}{(4\pi)^2},$$

where $\hat{\mathbf{q}}_+$ is the antilepton direction in the rest frame of its parent top quark and $\hat{\mathbf{q}}_-$ is the lepton direction in the rest frame of its parent antitop quark; and $\Omega_+$ is the solid angle associated with the antilepton and $\Omega_-$ is the solid angle associated with the lepton. The vectors $\mathbf{B}^\pm$ determine the top-quark and antitop-quark polarizations, whereas the matrix $C$ contains their spin correlations. These terms are the same as those that appear in the general form for $\rho$. As the information about the polarizations and spin correlations of the short-lived top quarks is transferred to the decay leptons, their values can be extracted from a measurement of angular observables associated with these leptons, allowing us to reconstruct the $t\bar{t}$ spin quantum state.

The experiments at the LHC ring, such as ATLAS, are the only ones currently taking data that are able to produce and study the properties of the top quark. At the LHC, $t\bar{t}$ pairs are produced mainly by gluon–gluon fusion. When they are produced close to their production threshold, that is, when their invariant mass $m_{t\bar{t}}$ is close to twice the mass of the top quark ($m_{t\bar{t}} \sim 2 \cdot m_t \sim 350$ GeV), approximately 80% of the production cross-section of $t\bar{t}$ pairs arises from a spin-singlet state[28-30], which is maximally entangled. After averaging over all possible top-quark directions, entanglement only survives close to the threshold because of the rotational invariance of the spin-singlet. This invariance implies that the trace (the sum of all of the diagonal elements) of the correlation matrix $C$, in which each diagonal element corresponds to the spin correlation in a particular direction, is a good entanglement witness. It is an observable that can signal the presence of entanglement, with $\mathrm{tr}(C) + 1 < 0$ as a sufficient condition for entanglement[18]. It can be understood as a violation of a Cauchy–Schwarz inequality, a notable entanglement criterion in fields such as quantum optics, condensed matter or analogue gravity[31-34].

It is more convenient to define an entanglement marker by using $D = \mathrm{tr}[C]/3$ (ref. 18), which can be experimentally measured as

$$D = -3\langle\cos\varphi\rangle,$$

where $\langle\cos\varphi\rangle$ is the average value of the cosine of the angle (dot product) between the charged-lepton directions after they have been subjected to Lorentz boosting into the $t\bar{t}$ rest frame and then the rest frames of their parent top-quark and antitop-quark, which can be measured experimentally in an ensemble dataset. The existence of an entangled state is demonstrated if the measurement satisfies $D < -1/3$, derived from the Peres–Horodecki criterion[35,36] and is independent of the order of the calculation. It should be noted that the CMS collaboration has already measured $D = -0.237 \pm 0.011$ (ref. 26) inclusively, showing no signal of entanglement.

The standard model is a quantum theory, and entanglement is implicitly present in its predictions. Nevertheless, a demonstration of spin entanglement in $t\bar{t}$ pairs is challenging because of the inability to control the internal degrees of freedom in the initial state[19]. Currently, entanglement can be detected only with the help of a dedicated analysis in a restricted phase space such as the one presented here.

## The ATLAS detector and event samples

The ATLAS experiment[37-39] at the LHC is a multipurpose particle detector with a forward–backward symmetric cylindrical geometry and a solid-angle coverage of almost 4π. It is used to record particles produced in LHC collisions through a combination of particle position and energy measurements. The coordinate system is defined in the section 'Object identification in the ATLAS detector'. It consists of an inner-tracking detector surrounded by a thin superconducting solenoid providing a 2 T axial magnetic field, electromagnetic and hadronic calorimeters, and a muon spectrometer. The muon spectrometer surrounds the calorimeters and is based on three large superconducting air-core toroidal magnets with eight coils each providing a field integral of between 2.0 T m and 6.0 T m across the detector. An extensive software suite[40] is used in data simulation, the reconstruction and analysis of real and simulated data, detector operations, and the trigger and data acquisition systems of the experiment. The complete dataset of $pp$ collision events with a centre-of-mass energy of $\sqrt{s} = 13$ TeV collected with the ATLAS experiment during 2015–2018 is used, corresponding to an integrated luminosity of 140 fb$^{-1}$. This analysis focuses on the data sample recorded using single-electron or single-muon triggers[41].

A unique feature of particle physics is that very precise simulations of the standard model can be realized through the use of Monte Carlo event generators. These simulations replicate real collisions and their resultant particles on an event-by-event basis, and these events can be passed through sophisticated simulations of the ATLAS detector to produce simulated data. Comparing these simulated events with those recorded by the detector is one way to test the predictions of the standard model. Another is to use the simulated data to model how the ATLAS detector responds to a particular physics process, such as the pair production of top quarks, and to use these data to create corrections to undo the effect of the detector response on real data and then to compare these corrected data with theoretical predictions. This measurement uses the latter strategy.

Three distinct types of real and simulated data are used, each with associated physics objects. Detector level refers to real data before they have been corrected for detector effects and simulated data after they have been passed through simulation of the ATLAS detector. Parton level refers to simulated Monte Carlo events in which the particles arise from the fundamental interaction being simulated, such as quarks and bosons, or to real collision data that have been corrected to this level. Particle level refers to simulated data with physics objects that are built only from the stable particles that remain after the decay of the particles that exist at parton level, that is, particles that live long enough to interact with the detector, or to real data that have been corrected to this level. This measurement relies on the selection and reconstruction of muons, electrons, quarks and gluons as hadronic jets, neutrinos as missing transverse momentum ($\mathbf{p}_T^{\mathrm{miss}}$), $W$ bosons and

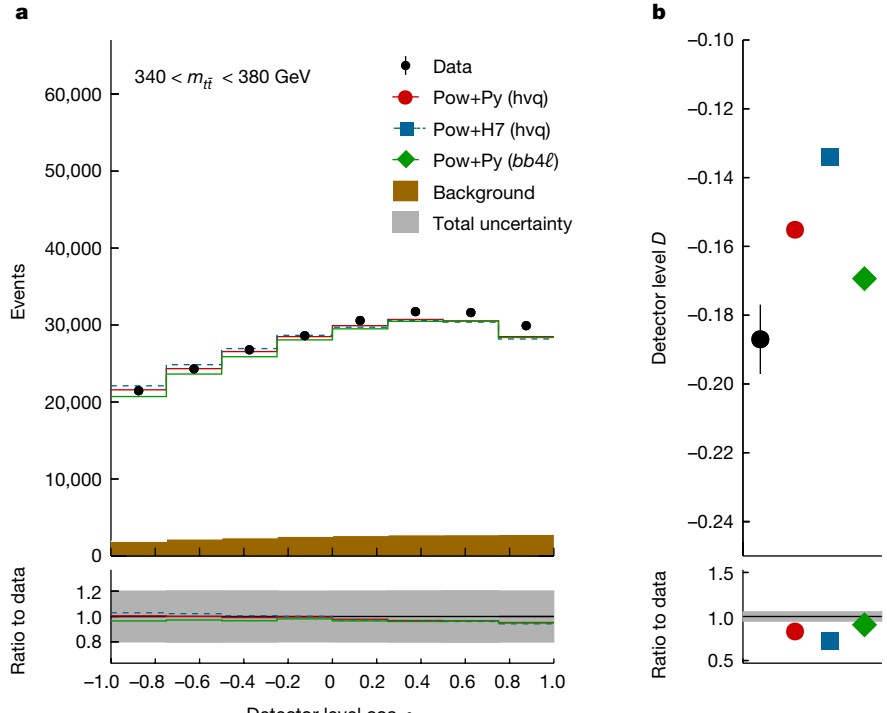

**Fig. 1 | Detector-level results. a**, The cos $\varphi$ observable in the signal region at the detector level. **b**, The entanglement marker $D$, calculated from the detector-level distributions, from three different Monte Carlo generators; the POWHEG + PYTHIA and POWHEG + HERWIG heavy-quark models, labelled Pow+Py (hvq) and Pow+H7 (hvq), respectively, and the POWHEG + PYTHIA $bb4\ell$ model, labelled Pow + Py ($bb4\ell$), are shown after background processes are subtracted. The uncertainty band shows the uncertainties from all sources added in quadrature. The ratios of the predictions to the data are shown at the bottom of **a** and **b**. The quoted value for $D$ for the $bb4\ell$ model also includes subtraction of the single-top-quark background.

top quarks. These objects are each reconstructed at the detector level, particle level and parton level. Details of how these objects are reconstructed in ATLAS and Monte Carlo simulations are provided in the section 'Object identification in the ATLAS detector'.

Monte Carlo event simulations are used to model the $t\bar{t}$ signal and the expected standard model background processes. The production of $t\bar{t}$ events was modelled using the POWHEG BOX v.2 heavy-quark (hvq) (refs. 42–45) generator at next-to-leading order (NLO) precision in QCD and the events were interfaced to either PYTHIA 8.230 (ref. 46) or HERWIG 7.2.1 (refs. 47,48) to model the parton shower and hadronization. The decays of the top quarks, including their spin correlations, were modelled at leading-order (LO) precision in QCD. An additional sample that generates $t\bar{t}$ events at full NLO accuracy in production and decay was generated using the POWHEG BOX RES ($bb4\ell$) (refs. 49,50) generator, interfaced to PYTHIA. Further details of the setup and tuning of these generators are provided in the section 'Monte Carlo simulation'. An important difference between PYTHIA and HERWIG is that the former uses a $p_\text{T}$-ordered shower, whereas the latter uses an angular-ordered shower (see section 'Parton shower and hadronization effects'). Another important consideration is that full information on the spin density matrix is not passed to the parton shower programs and, therefore, is not fully preserved during the shower.

The standard model background processes that contribute to the analysis are the production of a single top quark with a $W$ boson ($tW$), pair production of top quarks with an additional boson $t\bar{t} + X$ ($X = H, W, Z$) and the production of dileptonic events from either one or two massive gauge bosons ($W$ and $Z$ bosons). The generators for the hard-scatter processes and the showering are listed in the section 'Monte Carlo simulation'. The procedure for identifying and reconstructing detector-level objects is the same for data and Monte Carlo events.

## Analysis procedure

Only events taken during stable-beam conditions, and for which all relevant components of the detector were operational, are considered. To be selected, events must have exactly one electron and one muon with opposite-sign electric charges. A minimum of two jets is required, and at least one of them must be identified to originate from a $b$-hadron ($b$-tagged).

The background contribution of events with reconstructed objects that are misidentified as leptons, referred to as the 'fake-lepton' background, is estimated using a combination of Monte Carlo prediction and correction based on data. This data-driven correction is obtained from a control region dominated by fake leptons. It is defined by using the same selection criteria as above, except that the two leptons must have the same-sign electric charges. The difference between the numbers of observed events and predicted events in this region is taken as a scale factor and applied to the predicted fake-lepton events in the signal region.

Events that pass the event selection are separated into three analysis regions, based on the detector-level, particle-level or parton-level $m_{t\bar{t}}$, depending on the region. The signal region is constructed to be dominated by events that are as close to the production threshold as the resolution of the reconstruction method will allow, as this is the region in which the entanglement of the top quarks is expected to be maximized.

The optimal mass window for the signal region was determined to be $340 < m_{t\bar{t}} < 380$ GeV. Two additional validation regions are defined to validate the method used for the measurement. First, a region is defined close to the limit in which entanglement is not expected to be observable, and also with sizeable dilution from mis-reconstructed events from non-entangled regions, by requiring $380 < m_{t\bar{t}} < 500$ GeV. Second, a region in which no signal of entanglement is expected is

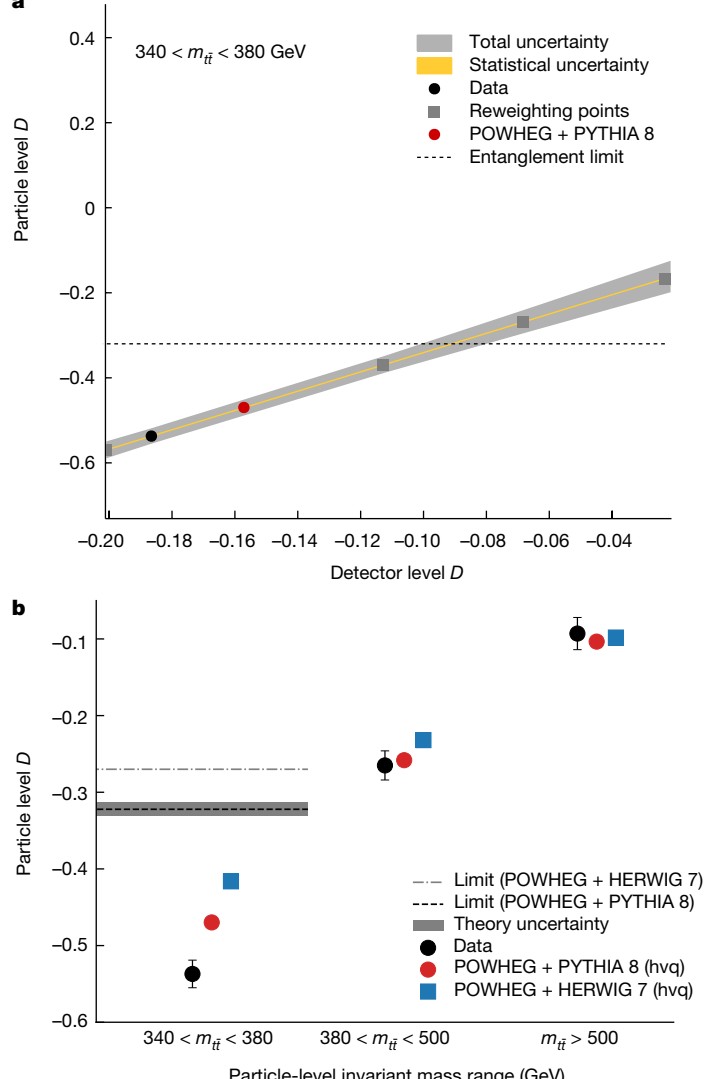

**a**

340 < $m_{t\bar{t}}$ < 380 GeV

Legend:
- Total uncertainty
- Statistical uncertainty
- ● Data
- ■ Reweighting points
- ● POWHEG + PYTHIA 8
- - - - Entanglement limit

(y-axis: Particle level D; x-axis: Detector level D)

**b**

Legend:
- –·– Limit (POWHEG + HERWIG 7)
- – – Limit (POWHEG + PYTHIA 8)
- Theory uncertainty
- ● Data
- ● POWHEG + PYTHIA 8 (hvq)
- ■ POWHEG + HERWIG 7 (hvq)

(y-axis: Particle level D; x-axis: Particle-level invariant mass range (GeV))

340 < $m_{t\bar{t}}$ < 380    380 < $m_{t\bar{t}}$ < 500    $m_{t\bar{t}}$ > 500

**Fig. 2 | Summary of results. a**, Calibration curve for the dependence between the particle-level value of $D$ and the detector-level value of $D$ in the signal region. The yellow band represents the statistical uncertainty, and the grey band represents the total uncertainty obtained by adding the statistical and systematic uncertainties in quadrature. The measured values and expected values from POWHEG + PYTHIA 8 (hvq) are marked with black and red circles, respectively, and the entanglement limit is shown as a dashed line. **b**, The particle-level $D$ results in the signal and validation regions compared with various Monte Carlo models. The entanglement limit shown is a conversion from its parton-level value of $D = -1/3$ to the corresponding value at the particle level, and the uncertainties that are considered for the band are described in the text.

defined with $m_{t\bar{t}}$ > 500 GeV. Each of the regions has a $t\bar{t}$-event purity of more than 90%. The dominant sources of background processes arise from $tW$ and fake-lepton, accounting for 56% and 27% of the background in the signal region, respectively. The remaining 17% of background events arise from $t\bar{t} + X$ and the production of dileptonic events from either one or two massive gauge bosons. The distribution of cos $\varphi$ in the signal region and the detector-level $D_{detector}$ value, built from the cos $\varphi$ at the reconstructed detector level and after background subtraction, are shown in Fig. 1a,b.

To compare the data with calculations and correct for detector effects, we must also define an event selection using the 'truth' information in the Monte Carlo event record. This selection uses particle-level objects to match as closely as possible the selection at the detector

level and is called a fiducial particle-level selection. Particle-level events are required to contain exactly one electron and one muon with opposite-sign electric charges and at least two particle-level jets, one of which must contain a $b$-hadron. The cos $\varphi$ distribution is then constructed from the particle-level top quarks and charged leptons in the same manner as at the detector level.

The response of the detector, the event selections and the top-quark reconstruction distort the shape of the cos $\varphi$ distribution. The observed distribution is corrected for these effects with a simple method: a simulation-based calibration curve that connects any value at the detector level to the corresponding value at the particle level. We correct the data for detector effects by using a unique calibration curve built for each signal and validation region based on the expected signal model, after subtracting the expected contribution from background processes. Owing to the limited resolution of the reconstructed mass of the $t\bar{t}$ system, some events that truly belong to the validation regions can enter the signal region at the detector level. These events are treated as detector effects.

To build these curves, Monte Carlo event samples are created with alternative values of $D$ by reweighting the events, following the procedure described in the section 'Reweighting the cos $\varphi$ distribution'. The calibration curve corrects the value $D_{detector}$ measured at the detector level to a corresponding value $D_{particle}$ at the particle level. To construct the calibration curve, several hypotheses for different values of $D$, denoted by $D'_{particle}$ with a corresponding $D'_{detector}$ value, are created corresponding to the changes in the expected value of entanglement.

The pairs of $D'_{detector}$ and $D'_{particle}$ are plotted in Fig. 2a. A straight line interpolates between the points. With this calibration curve, any value for $D_{detector}$ can be calibrated to the particle level.

Three categories of uncertainties are included in the calibration curves: uncertainties in modelling $t\bar{t}$ production and decay, uncertainties in modelling the backgrounds and detector-related uncertainties for both the $t\bar{t}$ signal and the standard model background processes. Each source of systematic uncertainty can result in a different calibration curve because it changes the shape of the cos $\varphi$ distribution at the particle level and/or detector level. For each source of systematic uncertainty, the data are corrected using this new calibration curve, and the resultant deviation from the data corrected by the nominal curve is taken as the systematic uncertainty of the data due to that source. Systematic uncertainties from all sources are summed in quadrature to determine the final uncertainty in the result.

For all of the detector-related uncertainties, the particle-level quantity is not affected and only detector-level values change. For signal modelling uncertainties, the effects at the particle level propagate to the detector level, resulting in shifts in both. Uncertainties in modelling the background processes affect how much background is subtracted from the expected or observed data and can, therefore, cause changes in the calibration curve. These uncertainties are treated as fully correlated between the signal and background (that is, if a source of systematic uncertainty is expected to affect both the signal and background processes, this is estimated simultaneously and not separately).

A summary of the different sources of systematic uncertainty and their impact on the result is given in Table 1. The size of each systematic uncertainty depends on the value of $D$ and is given in Table 1 for the standard model prediction, calculated with POWHEG + PYTHIA. The systematic uncertainties considered in the analysis are described in detail in the section 'Systematic uncertainties'.

To compare the particle-level result with the parton-level entanglement limit $D < -1/3$, the limit must be folded to the particle level. A second calibration curve is constructed to relate the value of $D_{parton}$ to the corresponding $D_{particle}$. The definitions of parton-level top quarks and leptons in the Monte Carlo generator follow ref. 24 and correspond approximately to those of stable top quarks and leptons in a fixed-order calculation. Only systematic uncertainties related to the modelling of the $t\bar{t}$ production and decay process are considered while building this

## Table 1 | Summary of uncertainties

| Source of uncertainty | $\Delta D_{observed}$ ($D=-0.537$) | $\Delta D_{observed}$ (%) | $\Delta D_{expected}$ ($D=-0.470$) | $\Delta D_{expected}$ (%) |
|---|---|---|---|---|
| Signal modelling | 0.017 | 3.2 | 0.015 | 3.2 |
| Electrons | 0.002 | 0.4 | 0.002 | 0.4 |
| Muons | 0.001 | 0.2 | 0.001 | 0.1 |
| Jets | 0.004 | 0.7 | 0.004 | 0.8 |
| $b$-Tagging | 0.002 | 0.4 | 0.002 | 0.4 |
| Pile-up | <0.001 | <0.1 | <0.001 | <0.1 |
| $E_T^{miss}$ | 0.002 | 0.4 | 0.002 | 0.4 |
| Backgrounds | 0.005 | 0.9 | 0.005 | 1.1 |
| Total statistical uncertainty | 0.002 | 0.3 | 0.002 | 0.4 |
| Total systematic uncertainty | 0.019 | 3.5 | 0.017 | 3.6 |
| Total uncertainty | 0.019 | 3.5 | 0.017 | 3.6 |

A summary of the effect of the groups of uncertainties at the expected standard model value of $D_{expected}=-0.470$, corresponding to the POWHEG+PYTHIA modelling, and the observed value $D_{observed}=-0.537$, both in the signal region. $E_T^{miss}$ denotes the magnitude of the missing transverse momentum. The total systematic uncertainty is calculated as the sum in quadrature of the individual groups of systematic uncertainties.

calibration curve. The migration of the parton-level events from the signal region into the validation regions at the particle level and vice versa is very small.

The calibration procedure is performed in the signal region and the two validation regions to correct the data to a fiducial phase space at the particle level, as described in the previous section. All systematic uncertainties are included in the three regions. The observed (expected) results are

$$D = -0.537 \pm 0.002 \text{ (stat.)}$$
$$\pm 0.019 \text{ (syst.)} \ (-0.470 \pm 0.002 \text{ (stat.)} \pm 0.017 \text{ (syst.)}),$$

in the signal region of $340 < m_{t\bar{t}} < 380$ GeV and

$$D = -0.265 \pm 0.001 \text{ (stat.)}$$
$$\pm 0.019 \text{ (syst.)} \ (-0.258 \pm 0.001 \text{ (stat.)} \pm 0.019 \text{ (syst.)}),$$

$$D = -0.093 \pm 0.001 \text{ (stat.)}$$
$$\pm 0.021 \text{ (syst.)} \ (-0.103 \pm 0.001 \text{ (stat.)} \pm 0.021 \text{ (syst.)}),$$

in the validation regions of $380 < m_{t\bar{t}} < 500$ GeV and $m_{t\bar{t}} > 500$ GeV, respectively. The expected values are those predicted by POWHEG + PYTHIA. The calibration curve for the signal region and a summary of the results in all regions are presented in Fig. 2.

The observed values of the entanglement marker $D$ are compared with the entanglement limit in Fig. 2b. The parton-level bound $D = -1/3$ is converted to a particle-level bound by folding the limit to particle level to better highlight the differences between the predictions using different parton shower orderings. For POWHEG + PYTHIA, this yields $-0.322 \pm 0.009$, in which the uncertainty includes all uncertainties in the POWHEG + PYTHIA model except the parton shower uncertainty (for more details of these uncertainties, see section 'Systematic uncertainties'). Similarly, for POWHEG + HERWIG, with an angular-ordered parton shower, a value of $-0.27$ is obtained. No uncertainties are assigned in this case because it is merely used as an alternative model.

## Discussion

In both of the validation regions, with no entanglement signal, the measurements are found to agree with the predictions from different Monte Carlo setups within the uncertainties. This serves as a consistency check to validate the method used for the measurement.

Although the different models yield different predictions, the current precision of the measurements in the validation regions does not allow us to rule out any of the Monte Carlo setups that were used. It is important to note that close to the threshold, non-relativistic QCD processes, such as Coulomb bound state effects, affect the production of $t\bar{t}$ events[28] and are not accounted for in the Monte Carlo generators. The main impact of these effects is to change the line shape of the $m_{t\bar{t}}$ spectrum. The impact of these missing effects was tested by introducing them with an ad hoc reweighting of the Monte Carlo based on theoretical predictions, and the effect was found to be 0.5%. Other systematic uncertainties on the top-quark decay (1.6%) and top-quark mass (0.7%) also similarly change the line shape within our experimental resolution and have a much larger impact. Therefore, the ad hoc reweighting is not included by default in the measurement because including it would not change the sensitivity of the result within the precision quoted.

In the signal region, the POWHEG + PYTHIA and POWHEG + HERWIG generators yield different predictions. The size of the observed difference is consistent with changing the method of shower ordering and is discussed in detail in the section 'Parton shower and hadronization effects'.

In the signal region, the observed and expected significances with respect to the entanglement limit are well beyond five standard deviations, independently of the Monte Carlo model used to correct the entanglement limit to account for the fiducial phase space of the measurement. This is shown in Fig. 2b, in which the hypothesis of no entanglement is shown. The observed result in the region with $340$ GeV $< m_{t\bar{t}} < 380$ GeV establishes the formation of entangled $t\bar{t}$ states. This constitutes the first observation of entanglement in a quark–antiquark pair.

Apart from the fundamental interest in testing quantum entanglement in a new environment, this measurement in top quarks paves the way to use high-energy colliders, such as the LHC, as a laboratory to study quantum information and foundational problems in quantum mechanics. From a quantum information perspective, high-energy colliders are particularly interesting because of their relativistic nature and the richness of the interactions and symmetries that can be probed there. Furthermore, highly demanding measurements, such as measuring quantum discord and reconstructing the steering ellipsoid, can be naturally implemented at the LHC because of the vast number of available $t\bar{t}$ events[51]. From a high-energy physics perspective, borrowing concepts from quantum information theory inspires new approaches and observables that can be used to search for physics beyond the standard model[52–55].

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

The ATLAS Collaboration

G. Aad[1], B. Abbott[2], K. Abeling[3], N. J. Abicht[4], S. H. Abidi[5], A. Aboulhorma[6], H. Abramowicz[7], H. Abreu[8], Y. Abulaiti[9], B. S. Acharya[10,11,12], C. Adam Bourdarios[13], L. Adamczyk[14], S. V. Addepalli[15], M. J. Addison[16], J. Adelman[17], A. Adiguzel[18], T. Adye[19], A. A. Affolder[20], Y. Afik[21], M. N. Agaras[22], J. Agarwala[23,24], A. Aggarwal[25], C. Agheorghiesei[26], A. Ahmad[27], F. Ahmadov[28,271], W. S. Ahmed[29], S. Ahuja[30], X. Ai[31], G. Aielli[32,33], A. Aikot[34], M. Ait Tamlihat[6], B. Aitbenchikh[35], I. Aizenberg[36], M. Akbiyik[25], T. P. A. Åkesson[37], A. V. Akimov[272], D. Akiyama[38], N. N. Akolkar[39], S. Aktas[40], K. Al Khoury[41], G. L. Alberghi[42], J. Albert[43], P. Albicocco[44], G. L. Albouy[45], S. Alderweireldt[46], Z. L. Alegria[47], M. Aleksa[27], I. N. Aleksandrov[271], C. Alexa[48], T. Alexopoulos[49], A. Alfonsi[42], M. Algren[50], M. Alhroob[2], B. Ali[51], H. M. J. Ali[52], S. Ali[53], S. W. Alibocus[54], M. Aliev[55], G. Alimonti[56], W. Alkakhi[3], C. Allaire[57], B. M. M. Allbrooke[58], J. F. Allen[46], C. A. Allendes Flores[59], P. P. Allport[60], A. Aloisio[61,62], F. Alonso[63], C. Alpigiani[64], M. Alvarez Estevez[65], A. Alvarez Fernandez[25], M. Alves Cardoso[50], M. G. Alviggi[61,62], M. Aly[16], Y. Amaral Coutinho[66], A. Ambler[29], C. Amelung[27], M. Amerl[16], C. G. Ames[67], D. Amidei[68], S. P. Amor Dos Santos[69], K. R. Amos[34], V. Ananiev[70], C. Anastopoulos[71], T. Andeen[72], J. K. Anders[27], S. Y. Andrean[73,74], A. Andreazza[75], S. Angelidakis[76], A. Angerami[41,77], A. V. Anisenkov[272], A. Annovi[78], C. Antel[50], M. T. Anthony[71], E. Antipov[79], M. Antonelli[44], F. Anulli[80], M. Aoki[81], T. Aoki[82], J. A. Aparisi Pozo[34], M. A. Aparo[58], L. Aperio Bella[83], C. Appelt[84], A. Apyan[15], S. J. Arbiol Val[85], C. Arcangeletti[44], A. T. H. Arce[86], E. Arena[54], J-F. Arguin[87], S. Argyropoulos[88], J.-H. Arling[83], O. Arnaez[13], H. Arnold[89], G. Artoni[80,90], H. Asada[91], K. Asai[92], S. Asai[93], N. A. Asbah[93], K. Assamagan[5], R. Astalos[94], S. Atashi[95], R. J. Atkin[96], M. Atkinson[97], H. Atmani[98], P. A. Atmasiddha[99], K. Augsten[94], S. Auricchio[61,62], A. D. Auriol[60], V. A. Austrup[16], G. Avolio[27], K. Axiotis[50], G. Azuelos[87,100], D. Babal[101], H. Bachacou[102], K. Bachas[103,104], A. Bachiu[105], F. Backman[73,74], A. Badea[21], T. M. Baer[68], P. Bagnaia[80,90], M. Bahmani[84], D. Bahner[88], A. J. Bailey[34], V. R. Bailey[97], J. T. Baines[19], L. Baines[19], O. K. Baker[107], E. Bakos[108], D. Bakshi Gupta[109], V. Balakrishnan[2], R. Balasubramanian[89], E. M. Baldin[272], P. Balek[14], E. Ballabene[42,110], F. Balli[102], L. M. Baltes[111], W. K. Balunas[112], J. Balz[25], E. Banas[85], M. Bandieramonte[113], A. Bandyopadhyay[39], S. Bansal[39], L. Barak[7], M. Barakat[83], E. L. Barberio[114], D. Barberis[115,116], M. Barbero[117], M. Z. Barel[89], K. N. Barends[96], T. Barillari[117], M-S. Barisits[27], T. Barklow[118], P. Baron[119], D. A. Baron Moreno[16], A. Baroncelli[120], G. Barone[5], A. J. Barr[121], J. D. Barr[122], F. Barreiro[65], J. Barreiro Guimarães da Costa[123], U. Barron[7], M. G. Barros Teixeira[69], S. Barsov[272], F. Bartels[111], R. Bartoldus[118], A. E. Barton[52], P. Bartos[94], A. Basan[25], M. Baselga[4], A. Bassalat[57,124], M. J. Basso[100], C. R. Basson[96], R. L. Bates[125], S. Batlamous[6], J. R. Batley[112], B. Batool[126], M. Battaglia[20], D. Battulga[84], M. Bauce[80,90], M. Bauer[27], P. Bauer[39], L. T. Bazzano Hurrell[127], J. B. Beacham[86], T. Beau[128], J. Y. Beaucamp[63], P. H. Beauchemin[129], P. Bechtle[39], H. P. Beck[130,131], K. Becker[132], A. J. Beddall[133], V. A. Bednyakov[271], C. P. Bee[79], L. J. Beemster[108], T. A. Beermann[134], M. Begalli[134], M. Begel[5], A. Behera[79], J. K. Behr[83], J. F. Beirer[27], F. Beisiegel[39], M. Belfkir[135], G. Bella[7], L. Bellagamba[42], A. Bellerive[105], P. Bellos[60], K. Beloborodov[272], D. Benchekroun[35], F. Bendebba[35], Y. Benhammou[7], L. Beresford[83], M. Beretta[44], E. Bergeaas Kuutmann[136], N. Berger[13], B. Bergmann[51], J. Beringer[137], G. Bernardi[138], C. Bernius[118], F. U. Bernlochner[39], F. Bernon[1,27], A. Berrocal Guardia[22], T. Berry[30], P. Berta[13], A. Berthold[140], I. A. Bertram[52], S. Bethke[117], A. Betti[80,90], A. J. Bevan[106], N. K. Bhalla[88], M. Bhamjee[5], S. Bhatta[79], D. S. Bhattacharya[141], P. Bhattarai[118], K. D. Bhide[88], V. S. Bhopatkar[47], R. M. Bianchi[113], G. Bianco[42,110], O. Biebel[67], R. Bielski[142], M. Biglietti[143], C. S. Billingsley[144], M. Bindi[3], A. Bingul[145], C. Bini[80,90], A. Biondini[54], C. J. Birch-sykes[16], G. A. Bird[19,112], M. Birman[36], M. Biros[139], S. Biryukov[58], T. Bisanz[4], E. Bisceglie[146,147], J. P. Biswal[19], D. Biswas[126], K. Bjørke[70], I. Bloch[83], A. Blue[125], U. Blumenschein[106], J. Blumenthal[25], G. J. Bobbink[89], V. S. Bobrovnikov[272], M. Boehler[88], B. Boehm[141], D. Bogavac[27], A. G. Bogdanchikov[272], C. Bohm[73], V. Boisvert[30], P. Bokan[27], T. Bold[14], M. Bomben[138], M. Bona[106], M. Boonekamp[102], C. D. Booth[30], A. G. Borbély[125], I. S. Bordulev[272], H. M. Borecka-Bielska[87], G. Borissov[52], D. Bortoletto[121], D. Boscherini[42], M. Bosman[22], J. D. Bossio Sola[27], K. Bouaouda[35], N. Bouchhar[21], J. Boudreau[113], E. V. Bouhova-Thacker[52], D. Boumediene[148], R. Bouquet[43], A. Boveia[149], J. Boyd[27], D. Boye[5], I. R. Boyko[271], J. Bracinik[60], N. Brahimi[150], G. Brandt[151], O. Brandt[112], F. Braren[83], B. Brau[152], J. E. Brau[142], R. Brener[136], L. Brenner[89], R. Brenner[136], S. Bressler[36], D. Britton[125], D. Britzger[117], I. Brock[39], R. Brock[153], G. Brooijmans[41], W. K. Brooks[59], E. Brost[5], L. M. Brown[43], L. E. Bruce[93], T. L. Bruckler[121], P. A. Bruckman de Renstrom[85], B. Brüers[83], A. Bruni[42], G. Bruni[42], M. Bruschi[42], N. Bruscino[80,90], T. Buanes[154], Q. Buat[64], D. Buchin[117], A. G. Buckley[125], O. Bulekov[272], B. A. Bullard[118], S. Burdin[54], C. D. Burgard[4], A. M. Burger[27], B. Burghgrave[109], O. Burlayenko[88], J. T. P. Burr[112], C. D. Burton[72], J. C. Burzynski[155], E. L. Busch[41], V. Büscher[25], P. J. Bussey[125], J. M. Butler[156], C. M. Buttar[125], J. M. Butterworth[122], W. Buttinger[19], C. J. Buxo Vazquez[153], A. R. Buzykaev[272], S. Cabrera Urbán[34], L. Cadamuro[57], D. Caforio[157], H. Cai[113], Y. Cai[123,158], Y. Cai[159], V. M. M. Cairo[27], O. Cakir[160], N. Calace[27], P. Calafiura[137], G. Calderini[128], P. Calfayan[61], G. Callea[125], L. P. Caloba[66], D. Calvet[148], S. Calvet[148], M. Calvetti[78,162], R. Camacho Toro[128], S. Camarda[27], D. Camarero Munoz[15], P. Camarri[32,33], M. T. Camerlingo[61,62], D. Cameron[27], C. Camincher[43], M. Campanelli[122], A. Camplani[163], V. Canale[61,62], J. Cantero[34], Y. Cao[97], F. Capocasa[15], M. Capua[146,147], A. Carbone[56,75], R. Cardarelli[32], J. C. J. Cardenas[109], F. Cardillo[34], G. Carducci[146,147], T. Carli[27], G. Carlino[61], J. I. Carlotto[22], B. T. Carlson[113,164], E. M. Carlson[43,100], L. Carminati[56,75], A. Carnelli[102], M. Carnesale[80,90], S. Caron[165], E. Carquin[59], S. Carrá[56], G. Carratta[42,110], A. M. Carroll[142], J. W. S. Carter[166], T. M. Carter[46], M. P. Casado[22,167], M. Caspar[83], F. L. Castillo[13], L. Castillo Garcia[22], V. Castillo Gimenez[34], N. F. Castro[69,168], A. Catinaccio[27], J. R. Catmore[70], T. Cavaliere[13], V. Cavaliere[5], N. Cavalli[42,110], V. Cavasinni[78,162], Y. C. Cekmecelioglu[83], E. Celebi[40], F. Celli[121], M. S. Centonze[169,170], V. Cepaitis[50], K. Cerny[119], A. S. Cerqueira[171], A. Cerri[58], L. Cerrito[32,33], F. Cerutti[137], B. Cervato[126], A. Cervelli[42], G. Cesarini[44], S. A. Cetin[133], D. Chakraborty[17], J. Chan[137], W. Y. Chan[82], J. D. Chapman[112], E. Chapon[102], B. Chargeishvili[172], D. G. Charlton[60], M. Chatterjee[20], C. Chauhan[139], Y. Chen[159], H. Chen[159], H. Chen[5], J. Chen[174], J. Chen[155], M. Chen[121], S. Chen[82], S. J. Chen[159], X. Chen[102,174], X. Chen[175,176], Y. Chen[120], C. L. Cheng[177], H. C. Cheng[178], S. Cheong[118], A. Cheplakov[271], E. Cheremushkina[83], E. Cherepanova[89], R. Cherkaoui El Moursli[6], E. Cheu[79], K. Cheung[180], L. Chevalier[102], V. Chiarella[44], G. Chiarelli[78], G. Chiodini[169], A. S. Chisholm[60], A. Chitan[48], M. Chitishvili[34], M. V. Chizhov[271], K. Choi[72], Y. Chou[64], E. Y. S. Chow[165], K. L. Chu[36], M. C. Chu[178], X. Chu[123,158], J. Chudoba[181], J. J. Chwastowski[85], D. Cieri[117], K. M. Ciesla[14], V. Cindro[182], A. Ciocio[137],

F. Cirotto[61,62], Z. H. Citron[36,183], M. Citterio[56], D. A. Ciubotaru[48], A. Clark[50], P. J. Clark[46], C. Clarry[166], J. M. Clavijo Columbie[83], S. E. Clawson[83], C. Clement[73,74], J. Clercx[89], Y. Coadou[1], M. Cobal[10,184], A. Coccaro[116], R. F. Coelho Barrue[69], R. Coelho Lopes De Sa[152], S. Coelli[56], B. Cole[41], J. Collot[45], P. Conde Muiño[69,185], M. P. Connell[55], S. H. Connell[55], I. A. Connelly[125], E. I. Conroy[121], F. Conventi[61,186], H. G. Cooke[60], A. M. Cooper-Sarkar[121], A. Cordeiro Oudot Choi[128], L. D. Corpe[148], M. Corradi[80,90], F. Corriveau[29,187], A. Cortes-Gonzalez[84], M. J. Costa[34], F. Costanza[13], D. Costanzo[71], B. M. Cote[149], G. Cowan[30], K. Cranmer[177], D. Cremonini[42,110], S. Crépé-Renaudin[45], F. Crescioli[128], M. Cristinziani[126], M. Cristoforetti[188,189], V. Croft[89], J. E. Crosby[47], G. Crosetti[146,147], A. Cueto[65], T. Cuhadar Donszelmann[95], H. Cui[123,158], Z. Cui[179], W. R. Cunningham[125], F. Curcio[146,147], P. Czodrowski[27], M. M. Czurylo[190], M. J. Da Cunha Sargedas De Sousa[115,116], J. V. Da Fonseca Pinto[66], C. Da Via[16], W. Dabrowski[14], T. Dado[4], S. Dahbi[191], T. Dai[68], D. Dal Santo[130], C. Dallapiccola[152], M. Dam[163], G. D'amen[25], V. D'Amico[67], J. Damp[25], J. R. Dandoy[105], M. Danninger[155], V. Dao[27], G. Darbo[116], S. Darmora[173], S. J. Das[5,192], S. D'Auria[56,75], C. David[96], T. Davidek[139], B. Davis-Purcell[105], I. Dawson[106], H. A. Day-hall[51], K. De[109], R. De Asmundis[61], N. De Biase[83], S. De Castro[42,110], N. De Groot[165], P. de Jong[89], H. De la Torre[17], A. De Maria[159], A. De Salvo[80], U. De Sanctis[32,33], F. De Santis[169,170], A. De Santo[58], J. B. De Vivie De Regie[45], D. V. Dedovich[271], J. Degens[89], A. M. Deiana[144], F. Del Corso[42,110], J. Del Peso[65], F. Del Rio[111], L. Delagrange[128], F. Deliot[102], C. M. Delitzsch[4], M. Della Pietra[61,62], D. Della Volpe[50], A. Dell'Acqua[27], L. Dell'Asta[56,75], M. Delmastro[13], P. A. Delsart[45], S. Demers[107], M. Demichev[271], S. P. Denisov[272], L. D'Eramo[148], D. Derendarz[85], F. Derue[128], P. Dervan[54], K. Desch[39], C. Deutsch[39], F. A. Di Bello[115,116], A. Di Ciaccio[32,33], L. Di Ciaccio[13], A. Di Domenico[80,90], C. Di Donato[61,62], A. Di Girolamo[27], G. Di Gregorio[27], A. Di Luca[188,189], B. Di Micco[143,193], R. Di Nardo[143,193], M. Diamantopoulou[105], F. A. Dias[89], T. Dias Do Vale[155], M. A. Diaz[194,195], F. G. Diaz Capriles[39], M. Didenko[272], E. B. Diehl[68], L. Diehl[88], S. Díez Cornell[83], C. Diez Pardos[126], C. Dimitriadi[136,139], A. Dimitrievska[137], J. Dingfelder[39], I-M. Dinu[48], S. J. Dittmeier[190], F. Dittus[27], F. Djama[1], T. Djobava[172], C. Doglioni[16,37], A. Dohnalova[94], J. Dolejsi[139], Z. Dolezal[139], K. M. Dona[21], M. Donadelli[196], B. Dong[153], J. Donini[148], A. D'Onofrio[61,62], M. D'Onofrio[54], J. Dopke[19], A. Doria[61], N. Dos Santos Fernandes[69], P. Dougan[16], M. T. Dova[63], A. T. Doyle[125], M. A. Draguet[121], E. Dreyer[36], I. Drivas-koulouris[49], M. Drnevich[9], M. Drozdova[50], D. Du[20], T. A. du Pree[89], F. Dubinin[272], M. Dubovsky[94], E. Duchovni[36], G. Duckeck[67], O. A. Ducu[48], D. Duda[46], A. Dudarev[27], E. R. Duden[15], M. D'uffizi[16], L. Duflot[57], M. Dührssen[27], A. E. Dumitriu[48], M. Dunford[111], S. Dungs[4], K. Dunne[73,74], A. Duperrin[1], H. Duran Yildiz[160], M. Düren[157], A. Durglishvili[172], B. L. Dwyer[17], G. I. Dyckes[137], M. Dyndal[14], B. S. Dziedzic[85], Z. O. Earnshaw[58], G. H. Eberwein[121], B. Eckerova[94], S. Eggebrecht[3], E. Egidio Purcino De Souza[128], L. F. Ehrke[50], G. Eigen[154], K. Einsweiler[137], T. Ekelof[136], P. A. Ekman[37], S. El Farkh[6], Y. El Ghazali[197], H. El Jarrari[27], A. El Moussaouy[87], V. Ellajosyula[136], M. Ellert[136], F. Ellinghaus[151], N. Ellis[27], J. Elmsheuser[5], M. Elsing[27], D. Emeliyanov[19], Y. Enari[93], I. Ene[137], S. Epari[85], P. A. Erland[85], M. Errenst[151], M. Escalier[57], C. Escobar[34], E. Etzion[7], G. Evans[69], H. Evans[161], L. S. Evans[30], M. O. Evans[58], A. Ezhilov[272], S. Ezzarqtouni[35], F. Fabbri[125], L. Fabbri[42,110], G. Facini[122], V. Fadeyev[20], R. M. Fakhrutdinov[272], D. Fakoudis[25], S. Falciano[80], L. F. Falda Ulhoa Coelho[27], P. J. Falke[39], J. Faltova[139], C. Fan[97], Y. Fan[123], Y. Fang[123,158], M. Fanti[56,75], M. Faraj[10,11], Z. Farazpay[198], A. Farbin[109], A. Farilla[143], T. Farooque[153], S. M. Farrington[46], F. Fassi[6], D. Fassouliotis[76], M. Faucci Giannelli[32,33], W. J. Fawcett[112], L. Fayard[57], P. Federic[139], P. Federicova[181], O. L. Fedin[272], G. Fedotov[272], M. Feickert[177], L. Feligioni[1], D. E. Fellers[142], C. Feng[199], M. Feng[175], Z. Feng[89], M. J. Fenton[95], L. Ferencz[83], R. A. M. Ferguson[52], S. I. Fernandez Luengo[59], P. Fernandez Martinez[22], M. J. V. Fernoux[1], J. Ferrando[52], A. Ferrari[136], P. Ferrari[89,165], R. Ferrari[23], D. Ferrere[50], C. Ferretti[68], F. Fiedler[25], P. Fiedler[51], A. Filipčič[182], E. K. Filmer[200], F. Filthaut[165], M. C. N. Fiolhais[69,201,202], L. Fiorini[34], W. C. Fisher[153], T. Fitschen[16], P. M. Fitzhugh[102], I. Fleck[126], P. Fleischmann[68], T. Flick[151], M. Flores[203], L. R. Flores Castillo[178], L. Flores Sanz De Acedo[27], F. M. Follega[188,189], N. Fomin[154], J. H. Foo[166], A. Formica[102], A. C. Forti[16], E. Fortin[27], A. W. Fortman[137], M. G. Foti[137], L. Fountas[76,204], D. Fournier[57], H. Fox[52], P. Francavilla[78,162], S. Francescato[93], S. Franchellucci[50], M. Franchini[42,110], S. Franchino[111], D. Francis[27], L. Franco[165], V. Franco Lima[27], L. Franconi[83], M. Franklin[93], A. C. Freegard[106], W. S. Freund[66], Y. Y. Frid[7], J. Friend[125], N. Fritzsche[140], A. Froch[88], D. Froidevaux[27], J. A. Frost[121], Y. Fu[120], S. Fuenzalida Garrido[59], M. Fujimoto[117], K. Y. Fung[178], E. Furtado De Simas Filho[66], M. Furukawa[82], J. Fuster[34], A. Gabrielli[42,110], A. Gabrielli[166], P. Gadow[27], G. Gagliardi[115,116], L. G. Gagnon[137], E. J. Gallas[121], B. J. Gallop[19], K. K. Gan[149], S. Ganguly[82], Y. Gao[46], F. M. Garay Walls[194,195], B. Garcia[5], C. García[34], A. Garcia Alonso[89], A. G. Garcia Caffaro[107], J. E. García Navarro[34], M. Garcia-Sciveres[137], G. L. Gardner[99], R. W. Gardner[21], N. Garelli[129], D. Garg[205], R. B. Garg[118,206], J. M. Gargan[46], C. A. Garner[166], C. M. Garvey[96], P. Gaspar[66], V. K. Gassmann[129], G. Gaudio[23], V. Gautam[22], P. Gauzzi[80,90], I. L. Gavrilenko[272], A. Gavrilyuk[272], C. Gay[207], G. Gaycken[83], E. N. Gazis[49], A. A. Geanta[48], C. M. Gee[20], A. Gekow[149], C. Gemme[116], M. H. Genest[45], S. Gentile[80,90], A. D. Gentry[208], S. George[30], W. F. George[60], T. Geralis[209], P. Gessinger-Befurt[27], M. E. Geyik[151], M. Ghani[132], M. Ghneimat[126], K. Ghorbanian[106], A. Ghosal[126], A. Ghosh[95], A. Ghosh[179], B. Giacobbe[42], S. Giagu[80,90], T. Giani[89], P. Giannetti[78], A. Giannini[120], S. M. Gibson[30], M. Gignac[20], D. T. Gil[210], A. K. Gilbert[14], B. J. Gilbert[41], D. Gillberg[105], G. Gilles[89], L. Ginabat[128], D. M. Gingrich[100,211], M. P. Giordani[10,184], P. F. Giraud[102], G. Giugliarelli[10,184], D. Giugni[56], F. Giuli[27], I. Gkialas[76,204], L. K. Gladilin[272], C. Glasman[65], G. R. Gledhill[142], G. Glemža[83], M. Glisic[142], I. Gnesi[147,212], Y. Go[5], M. Goblirsch-Kolb[27], B. Gocke[4], D. Godin[87], B. Gokturk[40], S. Goldfarb[114], T. Golling[50], M. G. D. Gololo[191], D. Golubkov[272], J. P. Gombas[153], A. Gomes[69,213], G. Gomes Da Silva[126], A. J. Gomez Delegido[34], R. Gonçalo[69,201], L. Gonella[60], A. Gongadze[214], F. Gonnella[60], J. L. Gonski[41], R. Y. González Andana[46], S. González de la Hoz[34], R. Gonzalez Lopez[54], C. Gonzalez Renteria[137], M. V. Gonzalez Rodrigues[83], R. Gonzalez Suarez[136], S. Gonzalez-Sevilla[50], G. R. Gonzalvo Rodriguez[34], L. Goossens[27], B. Gorini[27], E. Gorini[169,170], A. Gorišek[182], T. C. Gosart[99], A. T. Goshaw[86], M. I. Gostkin[271], S. Goswami[47], C. A. Gottardo[27], S. A. Gotz[67], M. Gouighri[197], V. Goumarre[83], A. G. Goussiou[64], N. Govender[55], I. Grabowska-Bold[14], K. Graham[105], E. Gramstad[70], S. Grancagnolo[169,170], C. M. Grant[102,200], P. M. Gravila[215], F. G. Gravili[169,170], H. M. Gray[137], M. Greco[169,170], C. Grefe[39], I. M. Gregor[83], P. Grenier[118], S. G. Grewe[117], C. Grieco[22], A. A. Grillo[20], K. Grimm[216], S. Grinstein[22,217], J.-F. Grivaz[57], E. Gross[36], J. Grosse-Knetter[3], J. C. Grundy[121], L. Guan[68], W. Guan[5], C. Gubbels[207], J. G. R. Guerrero Rojas[34], G. Guerrieri[10,184], F. Guescini[117], R. Gugel[25], J. A. M. Guhit[68], A. Guida[83], E. Guilloton[19,132], S. Guindon[27], F. Guo[123,158], J. Guo[174], L. Guo[83], Y. Guo[68], R. Gupta[83], R. Gupta[113], S. Gurbuz[39], S. S. Gurdasani[88], G. Gustavino[27], M. Guth[50], P. Gutierrez[2], L. F. Gutierrez Zagazeta[99], M. Gutsche[140], C. Gutschow[122], C. Gwenlan[121], C. B. Gwilliam[54],

E. S. Haaland[70], A. Haas[9], M. Habedank[83], C. Haber[137], H. K. Hadavand[109], A. Hadef[140],
S. Hadzic[117], A. I. Hagan[52], J. J. Hahn[126], E. H. Haines[122], M. Haleem[141], J. Haley[47], J. J. Hall[71],
G. D. Hallewell[1], L. Halser[130], K. Hamano[43], M. Hamer[39], G. N. Hamity[46], E. J. Hampshire[30],
J. Han[199], K. Han[120], L. Han[159], L. Han[120], S. Han[137], Y. F. Han[166], Ka. Hanagaki[81], M. Hance[20],
D. A. Hangal[41], H. Hanif[155], M. D. Hank[99], J. B. Hansen[163], P. H. Hansen[163], K. Hara[218],
D. Harada[50], T. Harenberg[151], S. Harkusha[272], M. L. Harris[152], Y. T. Harris[121], J. Harrison[22],
N. M. Harrison[149], P. F. Harrison[132], N. M. Hartman[117], N. M. Hartmann[67], Y. Hasegawa[219],
R. Hauser[153], C. M. Hawkes[60], R. J. Hawkings[27], Y. Hayashi[82], S. Hayashida[91], D. Hayden[153],
C. Hayes[68], R. L. Hayes[89], C. P. Hays[121], J. M. Hays[106], H. S. Hayward[54], F. He[120], M. He[123,158],
Y. He[220], Y. He[83], Y. He[122], N. B. Heatley[106], V. Hedberg[37], A. L. Heggelund[70], N. D. Hehir[106,273],
C. Heidegger[88], K. K. Heidegger[88], W. D. Heidorn[221], J. Heilman[105], S. Heim[83], T. Heim[137],
J. G. Heinlein[99], J. J. Heinrich[142], L. Heinrich[117,222], J. Hejbal[181], A. Held[177], S. Hellesund[154],
C. M. Helling[207], S. Hellman[73,74], R. C. W. Henderson[52], L. Henkelmann[112],
A. M. Henriques Correia[27], H. Herde[37], Y. Hernández Jiménez[79], L. M. Herrmann[39],
T. Herrmann[140], G. Herten[88], R. Hertenberger[67], L. Hervas[27], M. E. Hesping[25], N. P. Hessey[100],
E. Hill[166], S. J. Hillier[60], J. R. Hinds[153], F. Hinterkeuser[39], M. Hirose[223], S. Hirose[218],
D. Hirschbuehl[151], T. G. Hitchings[16], K. Hiti[182], J. Hobbs[79], R. Hobincu[224], N. Hod[36],
M. C. Hodgkinson[71], B. H. Hodkinson[112], A. Hoecker[27], D. D. Hofer[68], J. Hofer[83], T. Holm[39],
M. Holzbock[117], L. B. A. H. Hommels[112], B. P. Honan[16], J. Hong[174], T. M. Hong[113],
B. H. Hooberman[97], W. H. Hopkins[173], Y. Horii[91], S. Hou[53], A. S. Howard[182], J. Howarth[125],
J. Hoya[173], M. Hrabovsky[119], A. Hrynevich[83], T. Hryn'ova[113], P. J. Hsu[180], S.-C. Hsu[64], Q. Hu[120],
Y. F. Hu[123,158], S. Huang[225], X. Huang[159], X. Huang[123,158], Y. Huang[71], Y. Huang[23], Z. Huang[16],
Z. Hubacek[51], M. Huebner[39], F. Huegging[39], T. B. Huffman[121], C. A. Hugli[83], M. Huhtinen[27],
S. K. Huiberts[154], R. Hulsken[29], N. Huseynov[28], J. Huston[153], J. Huth[93], R. Hyneman[118],
G. Iacobucci[50], G. Iakovidis[5], I. Ibragimov[28], L. Iconomidou-Fayard[27], P. Iengo[27],
P. Iengo[61,62], R. Iguchi[82], T. Iizawa[121], Y. Ikegami[82], Y. Ikemami[5], N. Ilic[166], H. Imam[35], M. Ince Lezki[50],
T. Ingebretsen Carlson[73,74], G. Introzzi[23,24], M. Iodice[143], V. Ippolito[80,90], R. K. Irwin[54],
M. Ishino[82], W. Islam[177], C. Issever[83,84], S. Istin[40,226], H. Ito[38], R. Iuppa[188,189], A. Ivina[36],
J. M. Izen[227], V. Izzo[81], P. Jacka[51,181], P. Jackson[200], B. P. Jaeger[155], C. S. Jagfeld[67], G. Jain[100],
P. Jain[88], K. Jakobs[88], T. Jakoubek[36], J. Jamieson[125], K. W. Janas[14], M. Javurkova[152], L. Jeanty[142],
J. Jejelava[228,229], P. Jenni[27,88], C. E. Jessiman[105], C. Jia[199], J. Jia[79], X. Jia[93], X. Jia[123,158], Z. Jia[159],
S. Jiggins[83], J. Jimenez Pena[22], S. Jin[159], A. Jinaru[48], O. Jinnouchi[220], P. Johansson[71],
K. A. Johns[179], J. W. Johnson[20], D. M. Jones[112], E. Jones[83], P. Jones[112], R. W. L. Jones[52],
T. J. Jones[54], H. L. Joos[3,27], R. Joshi[149], J. Jovicevic[2], X. Ju[137], J. J. Junggeburth[152],
T. Junkermann[111], A. Juste Rozas[22,217], M. K. Juzek[85], S. Kabana[230], A. Kaczmarska[85], M. Kado[117],
H. Kagan[149], M. Kagan[118], A. Kahn[41], A. Kahn[99], C. Kahra[25], T. Kaji[82], E. Kajomovitz[8], N. Kakati[36],
I. Kalaitzidou[88], C. W. Kalderon[166], A. Kamenshchikov[166], N. J. Kang[20], D. Kar[191], K. Karava[121],
M. J. Kareem[231], E. Karentzos[88], I. Karkanias[103], O. Karkout[89], S. N. Karpov[271], Z. M. Karpova[271],
V. Kartvelishvili[52], A. N. Karyukhin[272], E. Kasimi[103], J. Katzy[83], S. Kaur[105], K. Kawade[219],
M. P. Kawale[2], K. Kawamoto[232], T. Kawamoto[120], E. F. Kay[27], F. I. Kaya[129], S. Kazakos[153],
V. F. Kazanin[272], Y. Ke[79], J. M. Keaveney[96], R. Keeler[43], G. V. Kehris[93], J. S. Keller[105],
A. S. Kelly[72], J. J. Kempster[58], P. D. Kennedy[75], O. Kepka[181], B. P. Kerridge[132], S. Kersten[151],
B. P. Kerševan[182], S. Keshri[57], L. Keszeghova[94], S. Ketabchi Haghighat[166], R. A. Khan[113],
A. Khanov[47], A. G. Kharlamov[272], T. Kharlamova[272], E. E. Khoda[64], M. Kholodenko[272],
T. J. Khoo[84], G. Khoriauli[141], J. Khubua[172], Y. A. R. Khwaira[57], B. Kibirige[191], A. Kilgallon[142],
D. W. Kim[73,74], Y. K. Kim[21], N. Kimura[122], M. K. Kingston[3], A. Kirchhoff[3], C. Kirfel[39], F. Kirfel[39],
J. Kirk[19], A. E. Kiryunin[17], C. Kitsaki[49], O. Kivernyk[36], M. Klassen[11], C. Klein[105], L. Klein[141],
M. H. Klein[144], S. B. Klein[50], U. Klein[54], P. Klimek[27], A. Klimentov[5], T. Klioutchnikova[27],
P. Kluit[89], S. Kluth[117], E. Kneringer[233], T. M. Knight[166], A. Knue[4], R. Kobayashi[232],
D. Kobylianskii[36], S. F. Koch[121], M. Kocian[118], P. Kodyš[139], D. M. Koeck[142], P. T. Koenig[39],
T. Koffas[105], O. Kolay[140], I. Koletsou[13], T. Komarek[117], T. Köneke[83], A. X. Y. Kong[200], T. Kono[92],
N. Konstantinidis[122], P. Kontaxakis[50], B. Konya[37], R. Kopeliansky[161], S. Koperny[14], K. Korcyl[85],
K. Kordas[103,234], A. Korn[122], S. Korn[3], I. Korolkov[22], N. Korotkova[272], B. Kortman[89], O. Kortner[117],
S. Kortner[117], W. H. Kostecka[17], V. V. Kostyukhin[126], A. Kotsokechagia[102], A. Kotwal[86],
A. Koulouris[27], A. Kourkoumeli-Charalampidi[23,24], C. Kourkoumelis[76], E. Kourlitis[117,222],
O. Kovanda[58], R. Kowalewski[43], W. Kozanecki[102], A. S. Kozhin[272], V. A. Kramarenko[272],
G. Kramberger[182], P. Kramer[25], M. W. Krasny[128], A. Krasznahorkay[27], J. W. Kraus[151],
J. A. Kremer[83], T. Kresse[140], J. Kretzschmar[54], K. Kreul[84], P. Krieger[166], S. Krishnamurthy[152],
M. Krivos[139], K. Krizka[60], K. Kroeninger[4], H. Kroha[117], J. Kroll[181], J. Kroll[99], K. S. Krowpman[153],
U. Kruchonak[271], H. Krüger[39], N. Krumnack[80], C. Kruse[86], O. Kuchinskaia[272], S. Kuday[160],
S. Kuehn[27], R. Kuesters[88], T. Kuhl[83], V. Kukhtin[271], Y. Kulchitsky[272], S. Kuleshov[195,235],
M. Kumar[191], N. Kumari[83], P. Kumari[231], A. Kupco[181], T. Kupfer[4], A. Kupich[272], O. Kuprash[88],
H. Kurashige[236], L. L. Kurchaninov[100], O. Kurdysz[57], Y. A. Kurochkin[272], A. Kurova[272],
M. Kuze[220], A. K. Kvam[152], J. Kvita[119], T. Kwan[29], N. G. Kyriacou[166], L. A. O. Laatu[34], C. Lacasta[34],
F. Lacava[80,90], H. Lacker[84], D. Lacour[128], N. N. Lad[122], E. Ladygin[271], B. Laforge[128], T. Lagouri[48],
F. Z. Lahbabi[35], S. Lai[3], I. K. Lakomiec[14], N. Lalloue[45], J. E. Lambert[43], S. Lammers[161],
W. Lampl[179], C. Lampoudis[103,234], A. N. Lancaster[17], E. Lançon[5], U. Landgraf[88],
M. P. J. Landon[106], V. S. Lang[88], R. J. Langenberg[152], O. K. B. Langrekken[70], A. J. Lankford[95],
F. Lanni[27], K. Lantzsch[39], A. Lanza[23], A. Lapertosa[115,116], J. F. Laporte[102], T. Lari[56],
F. Lasagni Manghi[42], M. Lassnig[27], V. Latonova[181], A. Laudrain[25], A. Laurier[8], S. D. Lawlor[71],
Z. Lawrence[125], R. Lazaridou[132], M. Lazzaroni[56,75], B. Le[16], E. M. Le Boulicaut[86], B. Leban[182],
A. Lebedev[221], M. LeBlanc[7], F. Ledroit-Guillon[45], A. C. A. Lee[122], S. C. Lee[53], S. Lee[73,74],
T. F. Lee[54], L. L. Leeuw[55], H. P. Lefebvre[30], M. Lefebvre[43], C. Leggett[137], G. Lehmann Miotto[27],
M. Leigh[50], W. A. Leight[152], W. Leinonen[165], A. Leisos[103,237], M. A. L. Leite[196], C. E. Leitgeb[84],
R. Leitner[139], K. J. C. Leney[144], T. Lenz[39], S. Leone[78], C. Leonidopoulos[46], A. Leopold[238],
C. Leroy[87], R. Les[153], C. G. Lester[112], M. Levchenko[272], J. Levêque[13], L. J. Levinson[36],
G. Levrini[42,110], M. P. Lewicki[85], D. J. Lewis[13], A. Li[138], B. Li[199], C. Li[120], C-Q. Li[117], H. Li[120], H. Li[199],
H. Li[159], H. Li[175], H. Li[199], J. Li[174], K. Li[64], L. Li[174], M. Li[123,158], Q. Y. Li[120], S. Li[123,158], S. Li[150,174,239], T. Li[138],
X. Li[29], Z. Li[121], Z. Li[29], Z. Li[123,158], S. Liang[123,158], Z. Liang[123], M. Liberatore[102], B. Liberti[32],
K. Lie[240], J. Lieber Marin[66], H. Lien[161], K. Lin[153], R. E. Lindley[179], J. H. Lindon[211], E. Lipeles[99],
A. Lipniacka[154], A. Lister[207], J. D. Little[13], B. Liu[123], B. X. Liu[155], D. Liu[150,174], J. B. Liu[120], J. K. K. Liu[121],
K. Liu[150,174], M. Liu[120], M. Y. Liu[120], P. Liu[123], Q. Liu[64,150,174], X. Liu[120], X. Liu[199], Y. Liu[158,241], Y. L. Liu[199],
Y. W. Liu[120], J. Llorente Merino[155], S. L. Lloyd[106], E. M. Lobodzinska[83], P. Loch[179], T. Lohse[84],
K. Lohwasser[71], E. Loiacono[83], M. Lokajicek[181,273], J. D. Lomas[60], J. D. Long[97], I. Longarini[95],
L. Longo[169,170], R. Longo[97], I. Lopez Paz[242], A. Lopez Solis[155], N. Lorenzo Martinez[13],
A. M. Lory[67], G. Löschcke Centeno[58], O. Loseva[272], X. Lou[73,74], X. Lou[123,158], A. Lounis[57],
J. Love[173,272], P. A. Love[52], G. Lu[123,158], M. Lu[205], S. Lu[99], Y. J. Lu[180], H. J. Lubatti[64], C. Luci[80,90],

F. L. Lucio Alves[159], F. Luehring[161], I. Luise[79], O. Lukianchuk[57], O. Lundberg[238],
B. Lund-Jensen[238,273], N. A. Luongo[173], M. S. Lutz[27], A. B. Lux[156], D. Lynn[5], R. Lysak[181],
E. Lytken[37], V. Lyubushkin[271], T. Lyubushkina[271], M. M. Lyukova[79], Ha. Ma[5], K. Ma[120,271],
L. L. Ma[199,271], W. Ma[120], Y. Ma[47], D. M. Mac Donell[43], G. Maccarrone[44], J. C. MacDonald[25],
P. C. Machado De Abreu Farias[66], R. Madar[148], W. F. Mader[140], T. Madula[122], J. Maeda[236],
T. Maeno[5], H. Maguire[71], V. Maiboroda[102], A. Maio[69,213,243], K. Maj[14], O. Majersky[83],
S. Majewski[142], N. Makovec[57], V. Maksimovic[108], B. Malaescu[128], Pa. Malecki[85], V. P. Maleev[272],
F. Malek[45,244], M. Mali[182], D. Malito[30], U. Mallik[205,272], S. Maltezos[49], S. Malyukov[272],
J. Mamuzic[22], G. Mancini[44], M. N. Mancini[15], G. Manco[23,24,271], J. P. Mandalia[106], I. Mandić[182],
L. Manhaes de Andrade Filho[171], I. M. Maniatis[36], J. Manjarres Ramos[1,245], D. C. Mankad[36],
A. Mann[67], S. Manzoni[27], L. Mao[174], X. Mapekula[55], A. Marantis[103,237], G. Marchiori[138],
M. Marcisovsky[181], C. Marcon[56], M. Marinescu[60], S. Marium[83], A. Marjanovic[2], E. J. Marshall[52],
Z. Marshall[137], S. Marti-Garcia[34], T. A. Martin[132], V. J. Martin[46], B. Martin dit Latour[154],
L. Martinelli[80,90], M. Martinez[22,217], P. Martinez Agullo[34], V. I. Martinez Outschoorn[152],
P. Martinez Suarez[22], S. Martin-Haugh[19], V. S. Martoiu[48], A. C. Martyniuk[122], A. Marzin[27],
D. Mascione[188,189], L. Masetti[25], T. Mashimo[82], J. Masik[16], A. L. Maslennikov[272], L. Massarotti[61,62],
P. Mastrandrea[78,162], A. Mastroberardino[146,147], T. Masubuchi[82,272], T. Mathisen[27],
J. Matousek[139], N. Matsuzawa[82], J. Maurer[48], B. Maček[182], D. A. Maximov[272], R. Mazini[53],
I. Maznas[103], M. Mazza[153], S. M. Mazza[20,272], E. Mazzeo[56,75], C. Mc Ginn[5], J. P. Mc Gowan[29],
S. P. Mc Kee[68], C. C. McCracken[207], E. F. McDonald[114], A. E. McDougall[89], J. A. Mcfayden[58],
R. P. McGovern[99], G. Mchedlidze[172], R. P. Mckenzie[191], T. C. Mclachlan[83], D. J. Mclaughlin[122],
S. J. McMahon[19], C. M. Mcpartland[54], R. A. McPherson[43,187], S. Mehlhase[67], A. Mehta[54],
D. Melini[34], B. R. Mellado Garcia[191], A. H. Melo[3], F. Meloni[83], A. M. Mendes Jacques Da
Costa[16], H. Y. Meng[166], L. Meng[52], S. Menke[117], M. Mentink[27], E. Meoni[146,147], G. Mercado[17],
C. Merlassino[10,184], L. Merola[61,62], C. Meroni[56,75], J. Metcalfe[7], A. S. Mete[173], C. Meyer[161],
J-P. Meyer[102], R. P. Middleton[19], L. Mijović[46], G. Mikenberg[36], M. Mikestikova[181], M. Mikuž[182],
H. Mildner[25], A. Milic[27], D. W. Miller[21], E. H. Miller[118], L. S. Miller[105], A. Milov[36],
D. A. Milstead[73,74], T. Min[159], A. A. Minaenko[271], I. A. Minashvili[172], L. Mince[125], A. I. Mincer[9],
B. Mindur[14,272], M. Mineev[271], Y. Minegishi[232], Y. Mino[232], L. M. Mir[22], M. Miralles Lopez[125], M. Mironova[137,271],
A. Mishima[82], M. C. Missio[165], A. Mitra[132], V. A. Mitsou[34], Y. Mitsumori[17], O. Miu[16],
P. S. Miyagawa[106], T. Mkrtchyan[111], M. Mlinarevic[122], T. Mlinarevic[122], M. Mlynarikova[27],
S. Mobius[130], P. Mogg[67], M. H. Mohamed Farook[208], A. F. Mohammed[123,158], S. Mohapatra[41],
G. Mokgatitswane[191], L. Moleri[36], B. Mondal[126], S. Mondal[51], K. Mönig[83], E. Monnier[1],
L. Monsonis Romero[34], J. Montejo Berlingen[27], M. Montella[149], F. Montereali[143,193],
F. Monticelli[63], S. Monzani[10,184], N. Morange[57], A. L. Moreira De Carvalho[69],
M. Moreno Llácer[34], C. Moreno Martinez[50], P. Morettini[116], S. Morgenstern[27], M. Morii[93],
M. Morinaga[82], F. Morodei[80,90], L. Morvaj[27], P. Moschovakos[27], B. Moser[27], M. Mosidze[172],
T. Moskalets[88], P. Moskvitina[165], J. Moss[216,246], E. J. W. Moyse[152], O. Mtintsilana[191], S. Muanza[1],
J. Mueller[113], D. Muenstermann[52], R. Müller[130], G. A. Mullier[152], A. J. Mullin[112], J. J. Mullin[99],
D. P. Mungo[166], J. R. Muñoz de Nova[247], D. Munoz Perez[34], F. J. Munoz Sanchez[16], M. Murin[16],
W. J. Murray[19,132], M. Muškinja[137], C. Mwewa[5], A. G. Myagkov[272], A. J. Myers[109], G. Myers[161],
M. Myska[51], B. P. Nachman[137], O. Nackenhorst[4], K. Nagai[121], K. Nagano[82], J. L. Nagle[5,192],
E. Nagy[1], A. M. Nairz[27], Y. Nakahama[81], K. Nakamura[81], K. Nakkalil[138], H. Nanjo[223],
R. Narayan[144], E. A. Narayanan[208], I. Naryshkin[272], M. Naseri[105], S. Nasri[135], C. Nass[39],
G. Navarro[248,272], J. Navarro-Gonzalez[34], R. Nayak[7], A. Nayaz[84], P. Y. Nechaeva[272],
F. Nechansky[83], L. Nedic[121], T. J. Neep[60], A. Negri[23,24,272], M. Negrini[42], C. Nellist[29],
K. Nelson[68], S. Nemecek[181], M. Nessi[27,50], M. S. Neubauer[97], F. Neuhaus[25], J. Neundorf[83],
R. Newhouse[207], P. R. Newman[60], C. W. Ng[113], Y. W. Y. Ng[83], B. Ngair[249], H. D. N. Nguyen[1],
R. B. Nickerson[121], R. Nicolaidou[102], J. Nielsen[20], M. Niemeyer[3], J. Niermann[3,27], N. Nikiforou[27],
V. Nikolaenko[272], I. Nikolic-Audit[128], K. Nikolopoulos[60], P. Nilsson[5], I. Ninca[83,272],
H. R. Nindhito[50], G. Ninio[7], A. Nisati[80], N. Nishu[211], R. Nisius[17], J-E. Nitschke[140],
E. K. Nkadimeng[191], T. Nobe[82], D. L. Noel[112], T. Nommensen[250], M. B. Norfolk[71],
R. R. B. Norisam[122], B. J. Norman[105], M. Noury[35], J. Novak[182], T. Novak[83], L. Novotny[51],
R. Novotny[208], L. Nozka[119], K. Ntekas[95], N. M. J. Nunes De Moura Junior[66], E. Nurse[122],
J. Ocariz[128], A. Ochi[121], I. Ochoa[69], S. Oerdek[83,251], J. T. Offermann[21], A. Ogrodnik[139], A. Oh[16],
C. C. Ohm[238], H. Oide[81], R. Oishi[82], M. L. Ojeda[83], Y. Okumura[82], L. F. Oleiro Seabra[69],
S. A. Olivares Pino[235], D. Oliveira Damazio[5], D. Oliveira Goncalves[171], J. L. Oliver[95],
Ö. O. Öncel[88], A. P. O'Neill[130], A. Onofre[69,168], P. U. E. Onyisi[72], M. J. Oreglia[21], G. E. Orellana[63],
D. Orestano[143,193], N. Orlando[22], R. S. Orr[166], V. O'Shea[125], L. M. Osojnak[99], R. Ospanov[120],
G. Otero y Garzon[127], H. Otono[252], P. S. Ott[111], G. J. Ottino[137], M. Ouchrif[253], F. Ould-Saada[70],
M. Owen[125], R. E. Owen[19], K. Y. Oyulmaz[40], V. E. Ozcan[40], F. Ozturk[85], N. Ozturk[109], S. Ozturk[133],
H. A. Pacey[121], A. Pacheco Pages[22], C. Padilla Aranda[22], G. Padovano[80,90], S. Pagan Griso[137],
G. Palacino[161], A. Palazzo[169,170], J. Pan[107], T. Pan[78], D. K. Panchal[72], C. E. Pandini[89],
J. G. Panduro Vazquez[30], H. D. Pandya[200], H. Pang[175], P. Pani[83], G. Panizzo[10,184], L. Paolozzi[50],
S. Parajuli[97], A. Paramonov[173], C. Paraskevopoulos[44], D. Paredes Hernandez[225], K. R. Park[41],
T. H. Park[166], M. A. Parker[112], F. Parodi[115,116], E. W. Parrish[17], V. A. Parrish[46], J. A. Parsons[41],
U. Parzefall[88], B. Pascual Dias[87], L. Pascual Dominguez[1], E. Pasqualucci[80], S. Passaggio[116],
F. Pastore[30], P. Patel[85], U. M. Patel[16], J. R. Pater[16], T. Pauly[27], J. Pearkes[64], M. Pedersen[70],
R. Pedro[69], S. V. Peleganchuk[272], O. Penc[27], E. A. Pender[46], G. D. Penn[107], K. E. Penski[67,272],
M. Penzin[272], B. S. Peralva[134], A. P. Pereira Peixoto[45], L. Pereira Sanchez[73,74],
D. V. Perepelitsa[5,192,272], E. Perez Codina[100], M. Perganti[49], H. Pernegger[27], O. Perrin[148],
K. Peters[83], R. F. Y. Peters[16], B. A. Petersen[27], T. C. Petersen[163], E. Petit[1], V. Petousis[51],
C. Petridou[103,234], A. Petrukhin[126], M. Pettee[137], N. E. Pettersson[27], A. Petukhov[272],
K. Petukhova[139], R. Pezoa[59], L. Pezzotti[27], G. Pezzullo[107,272], T. M. Pham[177], T. Pham[114],
P. W. Phillips[19], G. Piacquadio[79], E. Pianori[137], F. Piazza[142], R. Piegaia[127], D. Pietreanu[48],
A. D. Pilkington[16], M. Pinamonti[10,184], J. L. Pinfold[2], B. C. Pinheiro Pereira[69],
A. E. Pinto Pinoargote[25,102], L. Pintucci[10,184], K. M. Piper[58], A. Pirttikoski[50], D. A. Pizzi[105],
L. Pizzimento[225], A. Pizzini[89], M.-A. Pleier[5], V. Plesanovs[88], V. Pleskot[139], E. Plotnikova[271],
G. Poddar[13], R. Poettgen[37], L. Poggioli[128], I. Pokharel[3,271], S. Polacek[139], G. Polesello[23],
A. Poley[100,155], A. Polini[42], C. S. Pollard[132], Z. B. Pollock[149], E. Pompa Pacchi[80,90],
D. Ponomarenko[165], L. Pontecorvo[27], S. Popa[24], G. A. Popeneciu[255], A. Poreba[27],
D. M. Portillo Quintero[100], S. Pospisil[51], M. A. Postill[71], P. Postolache[26], K. Potamianos[132],
P. A. Potepa[14], I. N. Potrap[271], C. J. Potter[112], H. Potti[200], T. Poulsen[83], J. Poveda[34,271],
M. E. Pozo Astigarraga[27], A. Prades Ibanez[34], J. Pretel[88], D. Price[16], M. Primavera[169],
M. A. Principe Martin[65], R. Privara[119], T. Procter[125], M. L. Proffitt[64], N. Proklova[99],
K. Prokofiev[240], G. Proto[117], J. Proudfoot[173], M. Przybycien[14], W. W. Przygoda[210], A. Psallidas[209],
J. E. Puddefoot[71], D. Pudzha[272], D. Pyatiizbyantseva[272], J. Qian[68], D. Qichen[16], Y. Qin[16,272],

 

T. Qiu[46,272], A. Quadt[3], M. Queitsch-Maitland[16], G. Quetant[50], R. P. Quinn[207],
G. Rabanal Bolanos[93], D. Rafanoharana[88], F. Ragusa[56,75], J. L. Rainbolt[21], J. A. Raine[50],
S. Rajagopalan[5], E. Ramakoti[272], I. A. Ramirez-Berend[105], K. Ran[83,158], N. P. Rapheeha[191],
H. Rasheed[48,272], V. Raskina[128], D. F. Rassloff[111], A. Rastogi[137], S. Rave[25], B. Ravina[3],
I. Ravinovich[36], M. Raymond[27], A. L. Read[70], N. P. Readioff[71], D. M. Rebuzzi[23,24], G. Redlinger[5],
A. S. Reed[117], K. Reeves[15], J. A. Reidelsturz[151], D. Reikher[7], A. Rej[4], C. Rembser[27], M. Renda[48],
M. B. Rendel[117], F. Renner[83], A. G. Rennie[95], A. L. Rescia[83], S. Resconi[56], M. Ressegotti[115,116],
S. Rettie[27], J. G. Reyes Rivera[153], E. Reynolds[137], O. L. Rezanova[272], P. Reznicek[139], N. Ribaric[52],
E. Ricci[188,189], R. Richter[117,272], S. Richter[73,74], E. Richter-Was[210], M. Ridel[128], S. Ridouani[253],
P. Rieck[9], P. Riedler[27], E. M. Riefel[73,74], J. O. Rieger[89], M. Rijssenbeek[79], A. Rimoldi[23,24],
M. Rimoldi[27], L. Rinaldi[42,110], T. T. Rinn[5], M. P. Rinnagel[67], G. Ripellino[136], I. Riu[22],
P. Rivadeneira[83], J. C. Rivera Vergara[43], F. Rizatdinova[137], E. Rizvi[96], B. A. Roberts[132],
B. R. Roberts[137], S. H. Robertson[29,187], D. Robinson[112], C. M. Robles Gajardo[59],
M. Robles Manzano[25], A. Robson[125], A. Rocchi[32,33], C. Roda[78,162], S. Rodriguez Bosca[111],
Y. Rodriguez Garcia[248], A. Rodriguez Rodriguez[88], A. M. Rodríguez Vera[231], S. Roe[27],
J. T. Roemer[95], A. R. Roepe-Gier[20], J. Roggel[151], O. Røhne[70], R. A. Rojas[152], C. P. A. Roland[128],
J. Roloff[5], A. Romaniouk[272], M. Romano[23,24], M. Romano[42], A. C. Romero Hernandez[97],
N. Rompotis[54,272], L. Roos[128], S. Rosati[80], B. J. Rosser[21], E. Rossi[21], E. Rossi[61,62], L. P. Rossi[116],
L. Rossini[88], R. Rosten[149], M. Rotaru[48], B. Rottler[88], C. Rougier[1,245], D. Rousseau[57], D. Rousso[112],
A. Roy[97], S. Roy-Garand[166], A. Rozanov[1], Z. M. A. Rozario[125], Y. Rozen[8], A. Rubio Jimenez[34],
A. J. Ruby[54], V. H. Ruelas Rivera[84], T. A. Ruggeri[200], A. Ruggiero[27], A. Ruiz-Martinez[34],
A. Rummler[27], Z. Rurikova[88], N. A. Rusakovich[271], H. L. Russell[43], G. Russo[80,90],
J. P. Rutherfoord[179], S. Rutherford Colmenares[112,271], K. Rybacki[52], M. Rybar[139], E. B. Rye[70],
A. Ryzhov[144], J. A. Sabater Iglesias[50], P. Sabatini[34], H. F-W. Sadrozinski[20], F. Safai Tehrani[80],
B. Safarzadeh Samani[19], M. Safdari[118], S. Saha[117], S. Sahoo[49], M. Sahinsoy[117], A. Saibel[34], M. Saimpert[102],
M. Saito[82], T. Saito[82], D. Salamani[27], A. Salnikov[118], J. Salt[34], A. Salvador Salas[7],
D. Salvatore[146,147], F. Salvatore[58], A. Salzburger[27], D. Sammel[88], D. Sampsonidis[103,234],
D. Sampsonidou[142], J. Sánchez[34], V. Sanchez Sebastian[34], H. Sandaker[70], C. O. Sander[83],
J. A. Sandesara[152], M. Sandhoff[151], C. Sandoval[256], D. P. C. Sankey[19], T. Sano[232], A. Sansoni[44],
L. Santi[80,90], C. Santoni[148], H. Santos[69,213], A. Santra[36], K. A. Saoucha[257], J. G. Saraiva[69,243],
J. Sardain[179], O. Sasaki[81], K. Sato[218], C. Sauer[190], F. Sauerburger[88], E. Sauvan[13], P. Savard[100,166],
R. Sawada[82], C. Sawyer[19], L. Sawyer[198], I. Sayago Galvan[34], C. Sbarra[42], A. Sbrizzi[42,110],
T. Scanlon[122], J. Schaarschmidt[149], U. Schäfer[25], A. C. Schaffer[57,144], D. Schaile[67],
R. D. Schamberger[79], C. Scharf[84], M. M. Schefer[130], V. A. Schegelsky[272], D. Scheirich[139],
F. Schenck[84], M. Schernau[95], C. Scheulen[3,272], C. Schiavi[115,116], E. J. Schioppa[169,170],
M. Schioppa[146,147], B. Schlag[118,206], K. E. Schleicher[88], S. Schlenker[27], J. Schmeing[151],
M. A. Schmidt[151], K. Schmieden[25], C. Schmitt[25], N. Schmitt[25], S. Schmitt[83], L. Schoeffel[102],
A. Schoening[190], P. G. Scholer[88], E. Schopf[117], M. Schott[25], J. Schovancova[27], S. Schramm[50],
T. Schroer[50], H-C. Schultz-Coulon[190], M. Schumacher[88], B. A. Schumm[20], Ph. Schune[102],
A. J. Schuy[64], H. R. Schwartz[20], A. Schwartzman[118], T. A. Schwarz[68], Ph. Schwemling[102],
R. Schwienhorst[153], A. Sciandra[20], G. Sciolla[15], F. Scuri[78], C. D. Sebastiani[54], K. Sedlaczek[17],
P. Seema[84], S. C. Seidel[208], A. Seiden[20], B. D. Seidlitz[41], C. Seitz[83], J. M. Seixas[6],
G. Sekhniaidze[61], L. Selem[45], N. Semprini-Cesari[42,110], D. Sengupta[50], V. Senthilkumar[34],
L. Serin[57], L. Serkin[10,11], M. Sessa[32,33], H. Severini[2], F. Sforza[115,116], A. Sfyrla[50], E. Shabalina[3],
R. Shaheen[238], J. D. Shahinian[99], D. Shaked Renous[36], L. Y. Shan[123], M. Shapiro[137], A. Sharma[27],
A. S. Sharma[207], P. Sharma[205], P. B. Shatalov[272], K. Shaw[58], S. M. Shaw[16], A. Shcherbakova[272],
Q. Shen[138,174,272], D. J. Sheppard[155], P. Sherwood[122], L. Shi[122,272], X. Shi[123], C. O. Shimmin[107],
J. D. Shinner[30], I. P. J. Shipsey[121], S. Shirabe[252], M. Shiyakova[258,271], J. Shlomi[36], M. J. Shochet[21],
J. Shojaii[114], D. R. Shope[70,271], B. Shrestha[2], S. Shrestha[149,259], E. M. Shrif[191], M. J. Shroff[43],
P. Sicho[181], A. M. Sickles[191], E. Sideras Haddad[191], A. Sidoti[42], F. Siegert[140], Dj. Sijacki[108], F. Sili[63],
J. M. Silva[60], M. V. Silva Oliveira[5], S. B. Silverstein[73], S. Simion[57], R. Simoniello[27],
E. L. Simpson[125], H. Simpson[58], L. R. Simpson[68], N. D. Simpson[37], S. Simsek[133], S. Sindhu[3],
P. Sinervo[166], S. Singh[166], S. Sinha[83], S. Sinha[16], M. Sioli[42,110], I. Siral[27], E. Sitnikova[83],
S. Yu. Sivoklokov[272,273], J. Sjölin[73,74], A. Skaf[3], E. Skorda[60], P. Skubic[2], M. Slawinska[85],
V. Smakhtin[19], B. H. Smart[19], S. Yu. Smirnov[272], Y. Smirnov[272], L. N. Smirnova[272], O. Smirnova[272],
A. C. Smith[41,272], E. A. Smith[21,272], H. A. Smith[121,272], J. L. Smith[54], R. Smith[118], M. Smizanska[52],
K. Smolek[51], A. A. Snesarev[272], S. R. Snider[166], H. L. Snoek[89], S. Snyder[5], R. Sobie[43,187,272],
A. Soffer[7], C. A. Solans Sanchez[27], E. Yu. Soldatov[272], U. Soldevila[34], A. A. Solodkov[272],
S. Solomon[15], A. Soloshenko[272], K. Solovieva[88], O. V. Solovyanov[272], V. Solovyev[272],
P. Sommer[27,271], A. Sonay[22], W. Y. Song[231], A. Sopczak[51,272], A. L. Sopio[122], F. Sopkova[101],
J. D. Sorenson[208], I. R. Sotarriva Alvarez[220], V. Sothilingam[111], O. J. Soto Sandoval[195,260],
S. Sottocornola[161], R. Soualah[257], Z. Soumaimi[6], D. South[83], N. Soybelman[36], S. Spagnolo[169,170],
M. Spalla[117], D. Sperlich[88], G. Spigo[27], S. Spiteri[125], M. Spousta[139], E. J. Staats[105],
R. Stamen[111], A. Stampekis[60], M. Standke[39], E. Stanecka[85], M. V. Stange[140], B. Stanislaus[137],
M. M. Stanitzki[83], B. Stapf[83], E. A. Starchenko[272], G. H. Stark[20], J. Stark[1,245], P. Staroba[181],
P. Starovoitov[111,272], S. Stärz[29], R. Staszewski[85], G. Stavropoulos[209], J. Steentoft[136],
P. Steinberg[5], B. Stelzer[155], H. J. Stelzer[113], O. Stelzer-Chilton[100], H. Stenzel[157],
T. J. Stevenson[58], G. A. Stewart[27], J. R. Stewart[47], M. C. Stockton[27], G. Stoicea[48], M. Stolarski[69],
S. Stonjek[117], A. Straessner[140], J. Strandberg[238], S. Strandberg[73,74], M. Stratmann[151],
M. Strauss[2], T. Strebler[1], P. Strizenec[101], R. Ströhmer[141], D. M. Strom[142], R. Stroynowski[144],
A. Strubig[73,74], S. A. Stucci[5], B. Stugu[154], J. Stupak[2], N. A. Styles[83], D. Su[118], S. Su[120],
W. Su[150], X. Su[57,120], K. Sugizaki[82], V. V. Sulin[272], M. J. Sullivan[54], D. M. S. Sultan[188,189],
L. Sultanaliyeva[272], S. Sultansoy[261,272], T. Sumida[232], S. Sun[68], S. Sun[177,272],
O. Sunneborn Gudnadottir[136], N. Sur[1], M. R. Sutton[58], H. Suzuki[218], M. Svatos[181],
M. Swiatlowski[100], T. Swirski[141], I. Sykora[94], M. Sykora[139], T. Sykora[139], D. Ta[25],
K. Tackmann[83,251], A. Taffard[95], R. Tafirout[100], J. S. Tafoya Vargas[57], Y. Takubo[81], M. Talby[1],
A. A. Talyshev[272], K. C. Tam[225], N. M. Tamir[7], A. Tanaka[82], J. Tanaka[82,272], R. Tanaka[57],
M. Tanasini[115,116], Z. Tao[207], S. Tapia Araya[59], S. Tapprogge[25], A. Tarek Abouelfadl Mohamed[153],
S. Tarem[7], K. Tariq[123], G. Tarna[1,48], G. F. Tartarelli[56], P. Tas[139], M. Tasevsky[181], E. Tassi[146,147],
A. C. Tate[97], G. Tateno[82], Y. Tayalati[6,98], G. N. Taylor[114], W. Taylor[231], A. S. Tee[177],
R. Teixeira De Lima[118], P. Teixeira-Dias[30], J. J. Teoh[166], K. Terashi[82], J. Terron[57], S. Terzo[22],
M. Testa[44], R. J. Teuscher[166,187], A. Thaler[233], O. Theiner[50], N. Themistokleous[46],
T. Theveneaux-Pelzer[1], O. Thielmann[151], D. W. Thomas[30], J. P. Thomas[60], E. A. Thompson[137],
P. D. Thompson[60], E. Thomson[99], Y. Tian[3], V. Tikhomirov[272], Yu. A. Tikhonov[272],
S. Timoshenko[272], D. Timoshyn[139], E. X. L. Ting[200,272], P. Tipton[107,272], S. H. Tlou[191,272],
A. Tnourji[148], K. Todome[220], S. Todorova-Nova[139], S. Todt[140], M. Togawa[81], J. Tojo[252],
S. Tokár[94], K. Tokushuku[81], O. Toldaiev[161], R. Tombs[112], M. Tomoto[81,91], L. Tompkins[118,206],

K. W. Topolnicki[210], E. Torrence[142], H. Torres[1,245], E. Torró Pastor[34], M. Toscani[127], C. Tosciri[21],
M. Tost[72], D. R. Tovey[71], A. Traeet[154], I. S. Trandafir[48], T. Trefzger[141], A. Tricoli[5], I. M. Trigger[100],
S. Trincaz-Duvoid[128], D. A. Trischuk[15], B. Trocmé[45], C. Troncon[56], L. Truong[55], M. Trzebinski[85],
A. Trzupek[85], F. Tsai[79], M. Tsai[68], A. Tsiamis[103,234], P. V. Tsiareshka[272], S. Tsigaridas[100],
A. Tsirigotis[103,237], V. Tsiskaridze[166], E. G. Tskhadadze[228,272], M. Tsopoulou[103,234], Y. Tsujikawa[232],
I. I. Tsukerman[272], V. Tsulaia[137], S. Tsuno[81], K. Tsuri[92], D. Tsybychev[79,272], Y. Tu[225],
A. Tudorache[48], V. Tudorache[48], A. N. Tuna[93], S. Turchikhin[115,116], I. Turk Cakir[160], R. Turra[56],
T. Turtuvshin[262,271], P. M. Tuts[41], S. Tzamarias[103,234], P. Tzanis[49], E. Tzovara[25,271], F. Ukegawa[218],
P. A. Ulloa Poblete[195,260], E. N. Umaka[5], G. Unal[27], A. Undrus[5], G. Unel[95], J. Urban[101],
P. Urquijo[114], P. Urrejola[194], G. Usai[109], R. Ushioda[220], M. Usman[87], Z. Uysal[133], V. Vacek[51],
B. Vachon[29], K. O. H. Vadla[70], T. Vafeiadis[27], A. Vaitkus[122], C. Valderanis[67],
E. Valdes Santurio[73,74], M. Valente[100], S. Valentinetti[42,110], A. Valero[34], E. Valiente Moreno[34],
A. Vallier[1,245], J. A. Valls Ferrer[34], D. R. Van Arneman[89], T. R. Van Daalen[64], A. Van Der Graaf[4],
P. Van Gemmeren[173], M. Van Rijnbach[27,70], S. Van Stroud[122], I. Van Vulpen[89], M. Vanadia[32,33],
W. Vandelli[27], E. R. Vandewall[47], D. Vannicola[7], L. Vannoli[115,116], R. Vari[80], E. W. Varnes[179],
C. Varni[263], T. Varol[53], D. Varouchas[57], L. Varriale[34], K. E. Varvell[250], M. E. Vasile[48], L. Vaslin[81],
G. A. Vasquez[43], A. Vasyukov[271], F. Vazeille[148], T. Vazquez Schroeder[27], J. Veatch[216],
V. Vecchio[16,271], M. J. Veen[152], I. Veliscek[121], L. M. Veloce[166], F. Veloso[69,201], S. Veneziano[80],
A. Ventura[169,170], S. Ventura Gonzalez[102], A. Verbytskyi[117], M. Verducci[78,162], C. Vergis[39],
M. Verissimo De Araujo[66], W. Verkerke[89], J. C. Vermeulen[89], C. Vernieri[118], M. Vessella[152],
M. C. Vetterli[100,155], A. Vgenopoulos[103,234], N. Viaux Maira[59], T. Vickey[71], O. E. Vickey Boeriu[71],
G. H. A. Viehhauser[121], L. Vigani[190], M. Villa[42,110], M. Villaplana Perez[34], E. M. Villhauer[46],
E. Vilucchi[44], M. G. Vincter[105], G. S. Virdee[60], A. Vishwakarma[46], A. Visibile[89], C. Vittori[27],
I. Vivarelli[58], E. Voevodina[117], F. Vogel[67], J. C. Voigt[140], P. Vokac[51], Yu. Volkotrub[14],
J. Von Ahnen[83], E. Von Toerne[39], B. Vormwald[27], V. Vorobel[39], K. Vorobev[272], M. Vos[34],
K. Voss[126], M. Vozak[89], L. Vozdecky[106,272], N. Vranjes[108], M. Vranjes Milosavljevic[108],
M. Vreeswijk[89], N. K. Vu[150,174], R. Vuillermet[27], O. Vujinovic[25], I. Vukotic[21], S. Wada[218],
C. Wagner[152], J. M. Wagner[137], W. Wagner[151], S. Wahdan[151], H. Wahlberg[63], M. Wakida[91],
J. Walder[19], R. Walker[67], W. Walkowiak[126], A. Wall[99], T. Wamorkar[173], A. Z. Wang[20], C. Wang[25],
C. Wang[72], H. Wang[137], J. Wang[240], R.-J. Wang[25], R. Wang[173], S. M. Wang[53],
S. Wang[199], T. Wang[120], W. T. Wang[205], W. Wang[123], X. Wang[159], X. Wang[97], X. Wang[174],
Y. Wang[150], Y. Wang[159], Z. Wang[68], Z. Wang[86,150,174], Z. Wang[68], A. Warburton[29], R. J. Ward[60],
N. Warrack[125], S. Waterhouse[30], A. T. Watson[60], H. Watson[125], M. F. Watson[60], E. Watton[19,125],
G. Watts[64], B. M. Waugh[122], C. Weber[5], H. A. Weber[84], M. S. Weber[130], S. M. Weber[111], C. Wei[120],
Y. Wei[121], A. R. Weidberg[121], E. J. Weik[9], J. Weingarten[4], M. Weirich[25], C. Weiser[88],
C. J. Wells[83], T. Wenaus[5], B. Wendland[4], T. Wengler[27], N. S. Wenke[117], N. Wermes[39],
M. Wessels[111], A. M. Wharton[52], A. S. White[93], A. White[109], M. J. White[200], D. Whiteson[95],
L. Wickremasinghe[223], W. Wiedenmann[17], M. Wielers[19], C. Wiglesworth[163], D. J. Wilbern[2],
H. G. Wilkens[27], D. M. Williams[41], H. H. Williams[99], S. Williams[112], S. Willocq[152], B. J. Wilson[16],
P. J. Windischhofer[21], F. I. Winkel[127], F. Winklmeier[142], B. T. Winter[88], J. K. Winter[16],
M. Wittgen[118], M. Wobisch[198], T. Wolffs[89], J. Wollrath[95], M. W. Wolter[85], H. Wolters[69,201],
E. L. Woodward[41], S. D. Worm[83], B. K. Wosiek[85], K. W. Woźniak[85], S. Wozniewski[3],
K. Wraight[125], C. Wu[60], J. Wu[123,158], M. Wu[178], M. Wu[165], S. L. Wu[177], X. Wu[50], Y. Wu[120], Z. Wu[102],
J. Wuerzinger[117,272], T. R. Wyatt[16], B. M. Wynne[46], S. Xella[163], L. Xia[159], M. Xia[175], J. Xiang[240],
M. Xie[120], X. Xie[120], S. Xin[123,158], A. Xiong[142], J. Xiong[137], D. Xu[123], H. Xu[120], L. Xu[120], R. Xu[99],
T. Xu[68], Y. Xu[75], Z. Xu[46], Z. Xu[159], B. Yabsley[250], S. Yacoob[96], Y. Yamaguchi[220], E. Yamashita[82],
H. Yamauchi[218], T. Yamazaki[137], Y. Yamazaki[236], J. Yan[174], S. Yan[121], Z. Yan[156], H. J. Yang[150,174],
H. T. Yang[120], S. Yang[120], T. Yang[240], X. Yang[27], X. Yang[123], Y. Yang[144], Y. Yang[120], Z. Yang[120],
W-M. Yao[137], H. Ye[159], H. Ye[3], J. Ye[123], S. Ye[5], X. Ye[120], Y. Yeh[122], I. Yeletskikh[271], B. Yeo[263],
M. R. Yexley[122], P. Yin[41], K. Yorita[38], S. Younas[48], C. J. S. Young[27,271], C. Young[118], C. Yu[123,158],
Y. Yu[120], M. Yuan[68], R. Yuan[199], L. Yue[122], M. Zaazoua[120], B. Zabinski[85], E. Zaid[46], Z. K. Zak[85],
T. Zakareishvili[34], N. Zakharchuk[105], S. Zambito[50], J. A. Zamora Saa[195,235], J. Zang[82], D. Zanzi[88],
O. Zaplatilek[51], C. Zeitnitz[151], H. Zeng[123], J. C. Zeng[97], D. T. Zenger Jr[15], O. Zenin[272], T. Ženiš[94],
S. Zenz[106], S. Zerradi[6], D. Zerwas[57], M. Zhai[123,158], D. F. Zhang[71,272], J. Zhang[199], J. Zhang[173],
K. Zhang[123,158], L. Zhang[159], P. Zhang[123,158], R. Zhang[177], S. Zhang[144], T. Zhang[82],
X. Zhang[174], X. Zhang[199], Y. Zhang[138,174], Y. Zhang[122], Y. Zhang[159], Z. Zhang[137], Z. Zhang[57],
H. Zhao[64], T. Zhao[199], Y. Zhao[20], Z. Zhao[120], A. Zhemchugov[271], J. Zheng[159], K. Zheng[97],
X. Zheng[120], Z. Zheng[118], D. Zhong[97], B. Zhou[68,271], H. Zhou[179], N. Zhou[174], Y. Zhou[159], Y. Zhou[179],
C. G. Zhu[199], J. Zhu[68], Y. Zhu[174], Y. Zhu[120], X. Zhuang[123], K. Zhukov[272], N. I. Zimine[271],
J. Zinsser[190], M. Ziolkowski[126], L. Živković[108], A. Zoccoli[42,110], K. Zoch[93,272], T. G. Zorbas[71,271],
O. Zormpa[209], W. Zou[41] & L. Zwalinski[27]

[1]CPPM, Aix-Marseille Université, CNRS/IN2P3, Marseille, France. [2]Homer L. Dodge Department of Physics and Astronomy, University of Oklahoma, Norman, OK, USA. [3]II. Physikalisches Institut, Georg-August-Universität Göttingen, Göttingen, Germany. [4]Fakultät Physik, Technische Universität Dortmund, Dortmund, Germany. [5]Physics Department, Brookhaven National Laboratory, Upton, NY, USA. [6]Faculté des sciences, Université Mohammed V, Rabat, Morocco. [7]Raymond and Beverly Sackler School of Physics and Astronomy, Tel Aviv University, Tel Aviv, Israel. [8]Department of Physics, Technion, Israel Institute of Technology, Haifa, Israel. [9]Department of Physics, New York University, New York, NY, USA. [10]INFN Gruppo Collegato di Udine, Sezione di Trieste, Udine, Italy. [11]ICTP, Trieste, Italy. [12]Department of Physics, King's College London, London, UK. [13]LAPP, Université Savoie Mont Blanc, CNRS/IN2P3, Annecy, France. [14]Faculty of Physics and Applied Computer Science, AGH University of Krakow, Krakow, Poland. [15]Department of Physics, Brandeis University, Waltham, MA, USA. [16]School of Physics and Astronomy, University of Manchester, Manchester, UK. [17]Department of Physics, Northern Illinois University, DeKalb, IL, USA. [18]Department of Physics, Istanbul University, Istanbul, Türkiye. [19]Particle Physics Department, Rutherford Appleton Laboratory, Didcot, UK. [20]Santa Cruz Institute for Particle Physics, University of California Santa Cruz, Santa Cruz, CA, USA. [21]Enrico Fermi Institute, University of Chicago, Chicago, IL, USA. [22]Institut de Física d'Altes Energies (IFAE), Barcelona Institute of Science and Technology, Barcelona, Spain. [23]INFN Sezione di Pavia, Pavia, Italy. [24]Dipartimento di Fisica, Università di Pavia, Pavia, Italy. [25]Institut für Physik, Universität Mainz, Mainz, Germany. [26]Department of Physics, Alexandru Ioan Cuza University of Iasi, Iasi, Romania. [27]CERN, Geneva, Switzerland. [28]Institute of Physics, Azerbaijan Academy of Sciences, Baku, Azerbaijan. [29]Department of Physics, McGill University, Montreal, Quebec, Canada. [30]Department of Physics, Royal Holloway University of London, Egham, UK. [31]School of Physics and Microelectronics, Zhengzhou University, Zhengzhou, China. [32]INFN Sezione di Roma Tor Vergata, Rome, Italy.

[33]Dipartimento di Fisica, Università di Roma Tor Vergata, Roma, Italy. [34]Instituto de Física Corpuscular (IFIC), Centro Mixto Universidad de Valencia - CSIC, Valencia, Spain. [35]Faculté des Sciences Ain Chock, Réseau Universitaire de Physique des Hautes Energies – Université Hassan II, Casablanca, Morocco. [36]Department of Particle Physics and Astrophysics, Weizmann Institute of Science, Rehovot, Israel. [37]Fysiska Institutionen, Lunds Universitet, Lund, Sweden. [38]Waseda University, Tokyo, Japan. [39]Physikalisches Institut, Universität Bonn, Bonn, Germany. [40]Department of Physics, Bogazici University, Istanbul, Türkiye. [41]Nevis Laboratory, Columbia University, Irvington, NY, USA. [42]INFN Sezione di Bologna, Bologna, Italy. [43]Department of Physics and Astronomy, University of Victoria, Victoria, British Columbia, Canada. [44]INFN e Laboratori Nazionali di Frascati, Frascati, Italy. [45]LPSC, Université Grenoble Alpes, CNRS/IN2P3, Grenoble INP, Grenoble, France. [46]SUPA - School of Physics and Astronomy, University of Edinburgh, Edinburgh, UK. [47]Department of Physics, Oklahoma State University, Stillwater, OK, USA. [48]Horia Hulubei National Institute of Physics and Nuclear Engineering, Bucharest, Romania. [49]Physics Department, National Technical University of Athens, Zografou, Greece. [50]Département de Physique Nucléaire et Corpusculaire, Université de Genève, Geneva, Switzerland. [51]Czech Technical University in Prague, Prague, Czech Republic. [52]Physics Department, Lancaster University, Lancaster, UK. [53]Institute of Physics, Academia Sinica, Taipei, Taiwan. [54]Oliver Lodge Laboratory, University of Liverpool, Liverpool, UK. [55]Department of Mechanical Engineering Science, University of Johannesburg, Johannesburg, South Africa. [56]INFN Sezione di Milano, Milan, Italy. [57]IJCLab, Université Paris-Saclay, Orsay, France. [58]Department of Physics and Astronomy, University of Sussex, Brighton, UK. [59]Departamento de Física, Universidad Técnica Federico Santa María, Valparaíso, Chile. [60]School of Physics and Astronomy, University of Birmingham, Birmingham, UK. [61]INFN Sezione di Napoli, Napoli, Italy. [62]Dipartimento di Fisica, Università di Napoli, Napoli, Italy. [63]Instituto de Física La Plata, Universidad Nacional de La Plata and CONICET, La Plata, Argentina. [64]Department of Physics, University of Washington, Seattle, WA, USA. [65]Departamento de Física Teorica C-15 and CIAFF, Universidad Autónoma de Madrid, Madrid, Spain. [66]Universidade Federal do Rio De Janeiro COPPE/EE/IF, Rio de Janeiro, Brazil. [67]Fakultät für Physik, Ludwig-Maximilians-Universität München, Munich, Germany. [68]Department of Physics, University of Michigan, Ann Arbor, MI, USA. [69]Laboratório de Instrumentação e Física Experimental de Partículas - LIP, Lisboa, Portugal. [70]Department of Physics, University of Oslo, Oslo, Norway. [71]Department of Physics and Astronomy, University of Sheffield, Sheffield, UK. [72]Department of Physics, University of Texas at Austin, Austin, TX, USA. [73]Department of Physics, Stockholm University, Stockholm, Sweden. [74]Oskar Klein Centre, Stockholm, Sweden. [75]Dipartimento di Fisica, Università di Milano, Milano, Italy. [76]Physics Department, National and Kapodistrian University of Athens, Athens, Greece. [77]Lawrence Livermore National Laboratory, Livermore CA, USA. [78]INFN Sezione di Pisa, Pisa, Italy. [79]Departments of Physics and Astronomy, Stony Brook University, Stony Brook, NY, USA. [80]INFN Sezione di Roma, Rome, Italy. [81]KEK, High Energy Accelerator Research Organization, Tsukuba, Japan. [82]International Center for Elementary Particle Physics and Department of Physics, University of Tokyo, Tokyo, Japan. [83]Deutsches Elektronen-Synchrotron DESY, Hamburg and Zeuthen, Germany. [84]Institut für Physik, Humboldt Universität zu Berlin, Berlin, Germany. [85]Institute of Nuclear Physics Polish Academy of Sciences, Krakow, Poland. [86]Department of Physics, Duke University, Durham, NC, USA. [87]Group of Particle Physics, University of Montreal, Montreal, Quebec, Canada. [88]Physikalisches Institut, Albert-Ludwigs-Universität Freiburg, Freiburg, Germany. [89]Nikhef National Institute for Subatomic Physics and University of Amsterdam, Amsterdam, The Netherlands. [90]Dipartimento di Fisica, Sapienza Università di Roma, Roma, Italy. [91]Graduate School of Science and Kobayashi-Maskawa Institute, Nagoya University, Nagoya, Japan. [92]Ochanomizu University, Otsuka, Bunkyo-ku, Tokyo, Japan. [93]Laboratory for Particle Physics and Cosmology, Harvard University, Cambridge, MA, USA. [94]Faculty of Mathematics, Physics and Informatics, Comenius University, Bratislava, Slovakia. [95]Department of Physics and Astronomy, University of California Irvine, Irvine, CA, USA. [96]Department of Physics, University of Cape Town, Cape Town, South Africa. [97]Department of Physics, University of Illinois, Urbana, IL, USA. [98]Institute of Applied Physics, Mohammed VI Polytechnic University, Ben Guerir, Morocco. [99]Department of Physics, University of Pennsylvania, Philadelphia, PA, USA. [100]TRIUMF, Vancouver, British Columbia, Canada. [101]Department of Subnuclear Physics, Institute of Experimental Physics of the Slovak Academy of Sciences, Košice, Slovak Republic. [102]IRFU, CEA, Université Paris-Saclay, Gif-sur-Yvette, France. [103]Department of Physics, Aristotle University of Thessaloniki, Thessaloniki, Greece. [104]Department of Physics, University of Thessaly, Thessaly, Greece. [105]Department of Physics, Carleton University, Ottawa, Ontario, Canada. [106]School of Physics and Astronomy, Queen Mary University of London, London, UK. [107]Department of Physics, Yale University, New Haven, CT, USA. [108]Institute of Physics, University of Belgrade, Belgrade, Serbia. [109]Department of Physics, University of Texas at Arlington, Arlington, TX, USA. [110]Dipartimento di Fisica e Astronomia A. Righi, Università di Bologna, Bologna, Italy. [111]Kirchhoff-Institut für Physik, Ruprecht-Karls-Universität Heidelberg, Heidelberg, Germany. [112]Cavendish Laboratory, University of Cambridge, Cambridge, UK. [113]Department of Physics and Astronomy, University of Pittsburgh, Pittsburgh, PA, USA. [114]School of Physics, University of Melbourne, Melbourne Victoria, Australia. [115]Dipartimento di Fisica, Università di Genova, Genova, Italy. [116]INFN Sezione di Genova, Genova, Italy. [117]Max-Planck-Institut für Physik (Werner-Heisenberg-Institut), Munich, Germany. [118]SLAC National Accelerator Laboratory, Stanford, CA, USA. [119]Joint Laboratory of Optics, Palacký University, Olomouc, Czech Republic. [120]Department of Modern Physics and State Key Laboratory of Particle Detection and Electronics, University of Science and Technology of China, Hefei, China. [121]Department of Physics, Oxford University, Oxford, UK. [122]Department of Physics and Astronomy, University College London, London, UK. [123]Institute of High Energy Physics, Chinese Academy of Sciences, Beijing, China. [124]An-Najah National University, Nablus, Palestine. [125]SUPA - School of Physics and Astronomy, University of Glasgow, Glasgow, UK. [126]Department Physik, Universität Siegen, Siegen, Germany. [127]Facultad de Ciencias Exactas y Naturales, Departamento de Física, Universidad de Buenos Aires, y CONICET, Instituto de Física de Buenos Aires (IFIBA), Buenos Aires, Argentina. [128]LPNHE, Sorbonne Université, Université Paris Cité, CNRS/IN2P3, Paris, France. [129]Department of Physics and Astronomy, Tufts University, Medford, MA, USA. [130]Albert Einstein Center for Fundamental Physics and Laboratory for High Energy Physics, University of Bern, Bern, Switzerland. [131]Department of Physics, University of Fribourg, Fribourg, Switzerland. [132]Department of Physics, University of Warwick, Coventry, UK. [133]Istinye University, Sariyer, Istanbul, Türkiye. [134]Rio de Janeiro State University, Rio de Janeiro, Brazil. [135]United Arab Emirates University, Al Ain, United Arab Emirates. [136]Department of Physics and Astronomy, University of Uppsala, Uppsala, Sweden. [137]Physics Division, Lawrence Berkeley National Laboratory, Berkeley, CA, USA. [138]APC, Université Paris Cité, CNRS/IN2P3, Paris, France. [139]Charles University, Faculty of Mathematics and Physics, Prague, Czech Republic. [140]Institut für Kern- und Teilchenphysik, Technische Universität Dresden, Dresden, Germany. [141]Fakultät für Physik und Astronomie, Julius-Maximilians-Universität Würzburg, Würzburg, Germany. [142]Institute for Fundamental Science, University of Oregon, Eugene, OR, USA. [143]INFN Sezione di Roma Tre, Rome, Italy. [144]Physics Department, Southern Methodist University, Dallas, TX, USA. [145]Department of Physics Engineering, Gaziantep University, Gaziantep, Türkiye. [146]Dipartimento di Fisica, Università della Calabria, Rende, Italy. [147]Laboratori Nazionali di Frascati, INFN Gruppo Collegato di Cosenza, Frascati, Italy. [148]LPC, Université Clermont Auvergne, CNRS/IN2P3, Clermont-Ferrand, France. [149]Ohio State University, Columbus, OH, USA. [150]Tsung-Dao Lee Institute, Shanghai, China. [151]Fakultät für Mathematik und Naturwissenschaften, Fachgruppe Physik, Bergische Universität Wuppertal, Wuppertal, Germany. [152]Department of Physics, University of Massachusetts, Amherst, MA, USA. [153]Department of Physics and Astronomy, Michigan State University, East Lansing, MI, USA. [154]Department for Physics and Technology, University of Bergen, Bergen, Norway. [155]Department of Physics, Simon Fraser University, Burnaby, British Columbia, Canada. [156]Department of Physics, Boston University, Boston, MA, USA. [157]II. Physikalisches Institut, Justus-Liebig-Universität Giessen, Giessen, Germany. [158]University of Chinese Academy of Science (UCAS), Beijing, China. [159]Department of Physics, Nanjing University, Nanjing, China. [160]Department of Physics, Ankara University, Ankara, Türkiye. [161]Department of Physics, Indiana University, Bloomington, IN, USA. [162]Dipartimento di Fisica E. Fermi, Università di Pisa, Pisa, Italy. [163]Niels Bohr Institute, University of Copenhagen, Copenhagen, Denmark. [164]Department of Physics, Westmont College, Santa Barbara, CA, USA. [165]Institute for Mathematics, Astrophysics and Particle Physics, Radboud University/Nikhef, Nijmegen, The Netherlands. [166]Department of Physics, University of Toronto, Toronto, Ontario, Canada. [167]Departament de Fisica de la Universitat Autonoma de Barcelona, Barcelona, Spain. [168]Departamento de Física, Universidade do Minho, Braga, Portugal. [169]INFN Sezione di Lecce, Lecce, Italy. [170]Dipartimento di Matematica e Fisica, Università del Salento, Lecce, Italy. [171]Departamento de Engenharia Elétrica, Universidade Federal de Juiz de Fora (UFJF), Juiz de Fora, Brazil. [172]High Energy Physics Institute, Tbilisi State University, Tbilisi, Georgia. [173]High Energy Physics Division, Argonne National Laboratory, Argonne, IL, USA. [174]School of Physics and Astronomy, Shanghai Jiao Tong University, Key Laboratory for Particle Astrophysics and Cosmology (MOE), SKLPPC, Shanghai, China. [175]Physics Department, Tsinghua University, Beijing, China. [176]The Collaborative Innovation Center of Quantum Matter (CICQM), Beijing, China. [177]Department of Physics, University of Wisconsin, Madison, WI, USA. [178]Department of Physics, Chinese University of Hong Kong, Shatin, Hong Kong. [179]Department of Physics, University of Arizona, Tucson, AZ, USA. [180]Department of Physics, National Tsing Hua University, Hsinchu, Taiwan. [181]Institute of Physics of the Czech Academy of Sciences, Prague, Czech Republic. [182]Department of Experimental Particle Physics, Jožef Stefan Institute and Department of Physics, University of Ljubljana, Ljubljana, Slovenia. [183] Department of Physics, Ben Gurion University of the Negev, Beer Sheva, Israel. [184]Dipartimento Politecnico di ngegneria e Architettura, Università di Udine, Udine, Italy. [185]Departamento de Física, Instituto Superior Técnico, Universidade de Lisboa, Lisboa, Portugal. [186]Università di Napoli Parthenope, Napoli, Italy. [187]Institute of Particle Physics (IPP), Victoria British Columbia, Canada. [188]INFN-TIFPA, Povo, Italy. [189]Università degli Studi di Trento, Trento, Italy. [190]Physikalisches Institut, Ruprecht-Karls-Universität Heidelberg, Heidelberg, Germany. [191]School of Physics, University of the Witwatersrand, Johannesburg, South Africa. [192]Department of Physics, University of Colorado Boulder, Boulder CO, USA. [193]Dipartimento di Matematica e Fisica, Università Roma Tre, Roma, Italy. [194]Departamento de Física, Pontificia Universidad Católica de Chile, Santiago, Chile. [195]Millennium Institute for Subatomic physics at high energy frontier (SAPHIR), Santiago, Chile. [196]Instituto de Física, Universidade de São Paulo, São Paulo, Brazil. [197]Faculté des Sciences, Université Ibn-Tofail, Kénitra, Morocco. [198]Louisiana Tech University, Ruston, LA, USA. [199]Institute of Frontier and Interdisciplinary Science and Key Laboratory of Particle Physics and Particle Irradiation (MOE), Shandong University, Qingdao, China. [200]Department of Physics, University of Adelaide, Adelaide South Australia, Australia. [201]Departamento de Física, Universidade de Coimbra, Coimbra, Portugal. [202]Borough of Manhattan Community College, City University of New York, New York, NY, USA. [203]National Institute of Physics, University of the Philippines Diliman, Quezon City, Philippines. [204]Department of Financial and Management Engineering, University of the Aegean, Chios, Greece. [205]University of Iowa, Iowa City, IA, USA. [206]Department of Physics, Stanford University, Stanford, CA, USA. [207]Department of Physics, University of British Columbia, Vancouver, British Columbia, Canada. [208]Department of Physics and Astronomy, University of New Mexico, Albuquerque, NM, USA. [209]National Centre for Scientific Research 'Demokritos', Agia Paraskevi, Greece. [210]Marian Smoluchowski Institute of Physics, Jagiellonian University, Krakow, Poland. [211]Department of Physics, University of Alberta, Edmonton, Alberta, Canada. [212]Centro Studi e Ricerche Enrico Fermi, Rome, Italy. [213]Departamento de Física, Faculdade de Ciências, Universidade de Lisboa, Lisboa, Portugal. [214]University of Georgia, Tbilisi, Georgia. [215]West University in Timisoara, Timisoara, Romania. [216]California State University, Los Angeles, CA, USA. [217]Institucio Catalana de Recerca i Estudis Avancats, ICREA, Barcelona, Spain. [218]Division of Physics and Tomonaga Center for the History of the Universe, Faculty of Pure and Applied Sciences, University of Tsukuba, Tsukuba, Japan. [219]Department of Physics, Shinshu University, Nagano, Japan. [220]Department of Physics, Tokyo Institute of Technology, Tokyo, Japan. [221]Department of Physics and Astronomy, Iowa State University, Ames, IA, USA. [222]Technical University of Munich, Munich, Germany. [223]Graduate School of Science, Osaka University, Osaka, Japan. [224]National University of Science and Technology Politechnica, Bucharest, Romania. [225]Department of Physics, University of Hong Kong, Pok Fu Lam, Hong Kong. [226]Physics Department, Yeditepe University, Istanbul, Türkiye. [227]Physics Department, University of Texas at Dallas, Richardson, TX, USA. [228]E. Andronikashvili Institute of Physics, Ivane Javakhishvili Tbilisi State University, Tbilisi, Georgia. [229]Institute of Theoretical Physics, Ilia State University, Tbilisi, Georgia. [230]Instituto de Alta Investigación, Universidad de Tarapacá, Arica, Chile. [231]Department of Physics and Astronomy, York University, Toronto, Ontario, Canada. [232]Faculty of Science, Kyoto University, Kyoto, Japan. [233]Universität Innsbruck, Department of Astro and Particle Physics, Innsbruck, Austria. [234]Center for Interdisciplinary Research and Innovation (CIRI-AUTH), Thessaloniki, Greece. [235]Department of Physics, Universidad Andres Bello, Santiago, Chile. [236]Graduate School of Science, Kobe University, Kobe, Japan. [237]Hellenic Open University, Patras, Greece. [238]Department of Physics,

Royal Institute of Technology, Stockholm, Sweden. [239]Center for High Energy Physics, Peking University, Beijing, China. [240]Department of Physics and Institute for Advanced Study, Hong Kong University of Science and Technology, Kowloon, Hong Kong, China. [241]School of Science, Shenzhen Campus of Sun Yat-sen University, Guangzhou, China. [242]Centro Nacional de Microelectrónica (IMB-CNM-CSIC), Barcelona, Spain. [243]Centro de Física Nuclear da Universidade de Lisboa, Lisboa, Portugal. [244]Department of Physics, Stellenbosch University, Stellenbosch, South Africa. [245]L2IT, Université de Toulouse, CNRS/IN2P3UPS, Toulouse, France. [246]Department of Physics, California State University, Sacramento, CA, USA. [247]Departamento de Física de Materiales, Universidad Complutense de Madrid, Madrid, Spain. [248]Facultad de Ciencias y Centro de Investigaciónes, Universidad Antonio Nariño, Bogotá, Colombia. [249]New York University Abu Dhabi, Abu Dhabi, United Arab Emirates. [250]School of Physics, University of Sydney, Sydney, New South Wales, Australia. [251]Institut für Experimentalphysik, Universität Hamburg, Hamburg, Germany. [252]Research Center for Advanced Particle Physics and Department of Physics, Kyushu University, Fukuoka, Japan. [253]LPMR, Faculté des Sciences, Université Mohamed Premier, Oujda, Morocco. [254]Transilvania University of Brasov, Brasov, Romania. [255]National Institute for Research and Development of Isotopic and Molecular Technologies, Physics Department, Cluj-Napoca, Romania.

[256]Departamento de Física, Universidad Nacional de Colombia, Bogotá, Colombia. [257]University of Sharjah, Sharjah, United Arab Emirates. [258]Institute for Nuclear Research and Nuclear Energy (INRNE) of the Bulgarian Academy of Sciences, Sofia, Bulgaria. [259]Washington College, Chestertown, MD, USA. [260]Instituto de Investigación Multidisciplinario en Ciencia y Tecnología, y Departamento de Física, Universidad de La Serena, La Serena, Chile. [261]Division of Physics, TOBB University of Economics and Technology, Ankara, Türkiye. [262]Institute of Physics and Technology, Mongolian Academy of Sciences, Ulaanbaatar, Mongolia. [263]University of California, Berkeley, CA, USA. [264]Faculty of Physics, University of Bucharest, Bucharest, Romania. [265]iThemba Labs, Western Cape, South Africa. [266]University of South Africa, Department of Physics, Pretoria, South Africa. [267]University of Zululand, KwaDlangezwa, South Africa. [268]Faculté des Sciences Semlalia, Université Cadi Ayyad, LPHEA-Marrakech, Morocco. [269]Departamento de Física Teórica y del Cosmos, Universidad de Granada, Granada, Spain. [270]Faculty of Physics, Sofia University, 'St. Kliment Ohridski', Sofia, Bulgaria. [271]Affiliated with an international laboratory covered by a cooperation agreement with CERN, Geneva, Switzerland. [272]Affiliated with an institute covered by a cooperation agreement with CERN, Geneva, Switzerland. [273]Deceased: N. D. Hehir, M. Lokajicek, B. Lund-Jensen, S. Yu. Sivoklokov.

# Methods

## Object identification in the ATLAS detector

ATLAS uses a right-handed coordinate system with its origin at the nominal interaction point in the centre of the detector and the $z$-axis along the beam pipe. The $x$-axis points from the interaction point to the centre of the LHC ring, and the $y$-axis points upwards. Cylindrical coordinates $(r, \phi)$ are used in the transverse plane, where $\phi$ is the azimuth angle around the $z$-axis. The pseudorapidity is defined in terms of the polar angle $\theta$ as $\eta = -\ln\tan(\theta/2)$. Angular distance is measured in units of $\Delta R \equiv \sqrt{(\Delta\eta)^2 + (\Delta\phi)^2}$.

Reconstructed (detector-level) objects are defined as follows. Electron candidates are required to satisfy the 'tight' likelihood-based identification requirement as well as calorimeter- and track-based isolation criteria[56] and have pseudorapidity $|\eta| < 1.37$ or $1.52 < |\eta| < 2.47$. Muon candidates are required to satisfy the 'medium' identification requirement as well as track-based isolation criteria[57–59] and have $|\eta| < 2.5$. Electrons and muons must have a minimum transverse momentum ($p_T$) of 25–28 GeV, depending on the data-taking period. Showers of particles (jets) that arise from the hadronization of quarks and gluons[60] are reconstructed from particle-flow objects[61], using the anti-$k_t$ algorithm[62,63] with a radius parameter $R = 0.4$, a $p_T$ threshold of 25 GeV and a $|\eta| < 2.5$ requirement. Objects can fulfil the criteria for both jet and lepton selections, necessitating the implementation of an overlap removal procedure. This way, objects are associated with a singular hypothesis. First, any electron candidates that share a track with a muon candidate are removed. Subsequently, jets within $\Delta R = 0.2$ of an electron are removed, and afterwards, electrons within a region $0.2 < \Delta R < 0.4$ around any remaining jet are rejected. Jets that have fewer than three tracks and are within $\Delta R = 0.2$ of a muon candidate are removed, and muons within $\Delta R = 0.4$ of any remaining jet are discarded. A Jet-Vertex-Tagger (JVT) requirement is applied to jets with $p_T < 60$ GeV and $|\eta| < 2.4$ to suppress jets originating from additional interactions in the same or neighbouring bunch crossings (pile-up)[64]. Jets are tagged as containing $b$-hadrons using the DL1r tagger[65] with a $b$-tagging efficiency of 85%. Missing transverse momentum ($\mathbf{p}_T^{miss}$) (refs. 66,67) is determined from the imbalance in the transverse momenta of all reconstructed objects.

To measure $D$, the top quarks must be reconstructed from their measured decay products. In the $t\bar{t}$ dileptonic decay, apart from charged leptons and jets, there are two neutrinos that are not measured by the detector. Several methods are available to reconstruct the top quarks from the detector-level charged leptons, jets and $\mathbf{p}_T^{miss}$. The main method used in this work is the Ellipse method[68], which is a geometric approach to analytically calculate the neutrino momenta. This method yields at least one real solution in 85% of events. We always choose the solution with the lowest top-quark pair invariant mass, to populate the region that is close to the threshold. If this method fails (for example, the resultant solutions are all complex), the Neutrino Weighting method[69] is used. The Neutrino Weighting method assigns a weight to each possible solution by assessing the compatibility of the neutrino momenta and the $\mathbf{p}_T^{miss}$ in the event, after scanning possible values of the pseudorapidities of the neutrinos. In this analysis, the Neutrino Weighting method is only used in a small fraction of events (about 5%). Furthermore, in ref. 24, it was used in all events and the performance was found to be the same between samples that include and exclude spin correlation. If both methods fail, a simple pairing of each lepton with its closest $b$-tagged jet is used as proxies for the top- and antitop-quark, and no attempt is made to reconstruct the neutrinos. If a second $b$-tagged jet is not present in the event, the leading (highest) $p_T$ untagged jet is used instead. In all cases, a $W$ boson mass of 80.4 GeV and a top-quark mass of 172.5 GeV are used as input parameters.

In simulated events, parton-level objects are taken directly from the Monte Carlo history information and are required to have a status code of 1, indicating that they are the fundamental particles (partons) of the interaction. Top quarks are required to be partons that decay to a $W$ boson and a $b$ quark, whereas charged leptons are required to be the immediate decay parton from the $W$ boson from the top quark. Particle-level objects are reconstructed using simulated stable particles in the Monte Carlo simulation before their reconstruction in the detector but after hadronization. A particle is defined as stable if it has a mean lifetime greater than 30 ps, within the pseudorapidity acceptance of the detector. The selection criteria for the particle-level objects are chosen to correspond as closely as possible to the criteria applied to the detector-level objects. Electrons, muons and neutrinos are required to come from the electroweak decay of a top quark and are discarded if they arise from the decay of a hadron or a $\tau$-lepton. Electrons and muons are then 'dressed' by summing their four momenta with any prompt photons within $\Delta R = 0.1$. Electrons and muons must also be well separated from jet activity. If they lie within $\Delta R < 0.4$ from a jet, they are removed from the event. Leptons are also required to have $p_T > 10$ GeV and $|\eta| < 2.5$, and at least one lepton must have $p_T > 25$ GeV. Jets are built by clustering all stable particles, using the anti-$k_t$ algorithm with a radius parameter of $R = 0.4$ and are tagged as containing $b$-hadrons if they have at least one ghost-matched $b$-hadron[70,71] with $p_T > 5$ GeV. Jets are also required to have $p_T > 25$ GeV and $|\eta| < 2.5$. Each $W$ boson is reconstructed by combining an available electron and electron neutrino or muon and muon neutrino. The top quark and antitop quark are reconstructed by pairing the two leading $b$-tagged jets, or the $b$-tagged jet and the highest-$p_T$ untagged jet in events with only one $b$-tag, with the reconstructed $W$ bosons. Both potential jet–lepton combinations are formed and the one that minimizes $|m_t - m(W_1 + b_{1/2})| + |m_t - m(W_2 + b_{2/1})|$ is taken as the correct pairing, where $m_t$ denotes the mass of the top quark, $b_{1/2}$ denotes the two jets selected for the reconstruction, $W_{1/2}$ refers to the reconstructed $W$ bosons and $m$ is the invariant mass of the objects in brackets.

## Monte Carlo simulation

The production of $t\bar{t}$ events was modelled using the POWHEG BOX v.2 heavy-quark (hvq) (refs. 42–45) event generator. This generator uses matrix elements calculated at next-to-leading-order (NLO) precision in a strong coupling constant power expansion in QCD with the NNPDF3.0NLO (ref. 72) parton distribution function (PDF) set and the $h_{damp}$ parameter set to $1.5m_t$ (ref. 73). The $h_{damp}$ parameter is a resummation damping factor and one of the parameters that control the matching of POWHEG matrix elements to the parton shower and thus effectively regulates the high-$p_T$ radiation against which the system recoils. The decays of the top quarks, including their spin correlations, were modelled at leading-order (LO) precision in QCD. As an alternative, the POWHEG BOX RES (refs. 49,50) event generator, developed to treat decaying resonances within the POWHEG BOX framework and including off-shell and non-resonant effects in the matrix element calculation, was used to produce an additional event sample, labelled as $bb4\ell$ in the following. Although $bb4\ell$ is the higher-precision Monte Carlo sample, it cannot be compared directly with the data after they are corrected for detector effects as it is not possible to remove its off-shell component in a formally correct way. However, the effect of using this model was tested approximately and was found to not significantly change the conclusions of the measurement.

In the $bb4\ell$ event sample, spin correlations are calculated at NLO, and full NLO accuracy in $t\bar{t}$ production and decays is attained. To model the parton shower, hadronization and underlying event, the events from both POWHEG BOX v.2 and POWHEG BOX RES were interfaced to PYTHIA 8.230 (ref. 46), with parameters set according to the A14 set of tuned parameters[74] and using the NNPDF2.3LO set of PDFs[75]. Similarly, the events from POWHEG BOX v.2 (hvq) were also interfaced with HERWIG 7.2.1 (refs. 47,48), using the HERWIG 7.2.1 default set of tuned parameters. The decays of bottom and charm hadrons were performed by EVTGEN 1.6.0 (ref. 76). The spin information from the matrix element

calculation is not passed to the parton shower programs and, therefore, is not fully preserved during the shower.

All simulated event samples include pile-up interactions, and the events are reweighted to reproduce the observed distribution of the average number of collisions per bunch crossing.

## Reweighting the cos φ distribution

To construct the calibration curve, templates for alternative scenarios with different degrees of entanglement, and therefore with different values of $D$, must be extracted. The degree of entanglement is intrinsic in the calculations of the Monte Carlo event generators. However, the effects of entanglement can be directly accessed using $D$, measured from the average of the cos φ distribution in the event. Therefore, an event-by-event reweighting based on $D$ is used to vary the degree of entanglement. Although the measurement uses detector-level and particle-level objects, the observable $D$ is changed at the parton level, at which it is directly related to the entanglement between the top and antitop spins. Therefore, each event is reweighted according to its parton-level values of $m_{t\bar{t}}$ and cos φ, as described below.

The entanglement marker $D$ is extracted at the parton level from the cos φ distribution by using either the mean of the distribution $D = -3 \cdot \langle \cos\varphi \rangle$ or the slope of the normalized differential cross-section $(1/\sigma)\mathrm{d}\sigma/\mathrm{d}\cos\varphi = (1/2)(1 - D\cos\varphi)$.

For simplicity, the analysis always uses the mean of the distribution, although the two methods are equivalent. Thus, for the purpose of reweighting, we must change the slope of the cos φ distribution at the parton level. Each event is reweighted according to this slope, which in turn changes the distributions at the particle level and detector level.

The observable $D$ depends on the invariant mass of the $t\bar{t}$ system, $m_{t\bar{t}}$. To perform the reweighting, the differential value of $D$ per mass unit as a function of $m_{t\bar{t}}$, $D_\Omega(m_{t\bar{t}})$, has to be calculated. This is achieved by fitting a third-order polynomial of the form

$$D_\Omega(m_{t\bar{t}}) = x_0 + x_1 \cdot m_{t\bar{t}}^{-1} + x_2 \cdot m_{t\bar{t}}^{-2} + x_3 \cdot m_{t\bar{t}}^{-3},$$

where $x_0, x_1, x_2$ and $x_3$ are constants. This parametrization was found to describe well the value of $D_\Omega(m_{t\bar{t}})$, in good agreement with the Monte Carlo prediction. The values of the parameters of $D_\Omega(m_{t\bar{t}})$ depend on the Monte Carlo event generator and have to be calculated for the nominal sample and for the effect of each of the $t\bar{t}$ theory systematic uncertainties, as they change the parton-level cos φ values and thus $D_\Omega(m_{t\bar{t}})$.

The reweighting method is a simple scaling of the cos φ distribution according to the desired new value of $D$. This is done by assigning a weight $w$ to each event at parton level as

$$w = \frac{1 - D_\Omega(m_{t\bar{t}}) \cdot \mathcal{X} \cdot \cos\varphi}{1 - D_\Omega(m_{t\bar{t}}) \cdot \cos\varphi},$$

with $\mathcal{X}$ as the scaling hypothesis of $D$. If, for example, $\mathcal{X} = 1.2$, it means that $D$ is scaled up by 20% relative to its nominal value. To build the calibration curve, four alternative values of $D$ are considered, with $\mathcal{X} = 0.4, 0.6, 0.8, 1.2$, in addition to the nominal value without reweighting ($\mathcal{X} = 1.0$). It is important to note that these $\mathcal{X}$ values change $D$ across the entire $m_{t\bar{t}}$ spectrum. In Extended Data Fig. 1, the parton-level distribution of $D$ is shown in the signal region before and after reweighting.

## Background modelling

Simulated data in the form of Monte Carlo samples were produced using either the full ATLAS detector simulation[77] based on the GEANT4 framework[78] or, for the estimation of some of the systematic uncertainties, a faster simulation with parameterized showers in the calorimeters[79]. The effect of pile-up was modelled by overlaying each hard-scattering event with inelastic $pp$ collisions generated with PYTHIA 8.186 (ref. 80) using the NNPDF2.3LO set of PDFs[75] and the A3 set of tuned parameters[81].

Except for the events simulated with SHERPA, the EVTGEN program was used to simulate bottom and charm hadron decays. If not mentioned otherwise, the top-quark mass was set to $m_t = 172.5$ GeV. All event samples that were interfaced with PYTHIA used the A14 set of tuned parameters[74] and the NNPDF2.3LO PDF set.

Single-top quark $tW$ associated production was modelled using the POWHEG BOX v.2 (refs. 43–45,82) event generator, which provides matrix elements at NLO in the strong coupling constant $\alpha_s$ in the five-flavour scheme with the NNPDF3.0NLO (ref. 72) PDF set. The functional form of the renormalization and factorization scales was set to the default scale, which is equal to the top-quark mass. The diagram-removal scheme[83] was used to handle the interference with $t\bar{t}$ production[73]. The inclusive cross-section was corrected to the theoretical prediction calculated at NLO in QCD with next-to-next-leading-logarithm (NNLL) soft-gluon corrections[84,85]. For $pp$ collisions at a centre-of-mass energy of $\sqrt{s} = 13$ TeV, this cross-section corresponds to $\sigma(tW)_{\mathrm{NLO+NNLL}} = 71.7 \pm 3.8$ pb. The uncertainty in the cross-section due to the PDF was estimated using the MSTW2008NNLO 90%CL (refs. 86,87) PDF set and was added in quadrature to the effect of the scale uncertainty.

Samples of diboson final states ($VV$), where $V$ denotes a $W$ or $Z$ boson, were simulated with the SHERPA 2.2.2 (ref. 88) event generator, including off-shell effects and Higgs boson contributions, where appropriate. Fully leptonic final states and semileptonic final states, in which one boson decays leptonically and the other hadronically, were generated using matrix elements at NLO accuracy in QCD for up to one additional parton and at LO accuracy for up to three additional parton emissions. Samples for the loop-induced processes $gg \rightarrow VV$ were generated using LO-accurate matrix elements for up to one additional parton emission for both the cases of fully leptonic and semileptonic final states. The matrix element calculations were matched and merged with the SHERPA parton shower based on Catani–Seymour dipole factorization[89,90] using the MEPS@NLO prescription[91–94]. The virtual QCD corrections were provided by the OPENLOOPS library[95–97]. The NNPDF3.0NNLO set of PDFs was used[72], along with the dedicated set of tuned parton-shower parameters developed by the SHERPA authors.

The production of $V$ + jets events was simulated with the SHERPA 2.2.11 (ref. 88) event generator using NLO matrix elements for up to two partons, and LO matrix elements for up to five partons, calculated with the Comix (ref. 89) and OPENLOOPS 2 (refs. 95–98) libraries. They were matched with the SHERPA parton shower[90] using the MEPS@NLO prescription[91–94]. The set of tuned parameters developed by the SHERPA authors was used, along with the NNPDF3.0NNLO set of PDFs[72].

The production of $t\bar{t}V$ events was modelled using the MADGRAPH5_AMC@NLO 2.3.3 (ref. 99) event generator, which provides matrix elements at NLO in the strong coupling constant $\alpha_s$ with the NNPDF3.0NLO (ref. 72) PDFs. The functional form of the renormalization and factorization scales was set to $0.5 \times \sum_i \sqrt{m_i^2 + p_{\mathrm{T},i}^2}$, where the sum runs over all the particles generated from the matrix element calculation. Top quarks were decayed at LO using MADSPIN (refs. 100,101) to preserve spin correlations. The events were interfaced with PYTHIA 8.210 (ref. 46) for the simulation of parton showering and hadronization. The cross-sections were calculated at NLO QCD and NLO EW accuracy using MADGRAPH5_AMC@NLO as reported in ref. 102. For $t\bar{t}\ell\ell$ events, the cross-section was scaled by an off-shell correction estimated at one-loop level in $\alpha_s$.

The production of $t\bar{t}H$ events was modelled using the POWHEG BOX v.2 (refs. 42–45,103) event generator, which provides matrix elements at NLO in the strong coupling constant $\alpha_s$ in the five-flavour scheme with the NNPDF3.0NLO (ref. 72) PDF set. The functional form of the renormalization and factorization scales was set to $\sqrt[3]{m_{\mathrm{T}}(t) \cdot m_{\mathrm{T}}(\bar{t}) \cdot m_{\mathrm{T}}(H)}$. The events were interfaced with PYTHIA 8.230. The cross-section was calculated at NLO QCD and NLO EW accuracy using MADGRAPH5_AMC@NLO as reported in ref. 102. The predicted

value at $\sqrt{s} = 13$ TeV is $507^{+35}_{-50}$ fb, for which the uncertainties were estimated from variations of both $\alpha_s$ and the renormalization and factorization scales.

The background from non-prompt or fake leptons was modelled using simulated Monte Carlo events to describe the shape of the kinematic distributions. Monte Carlo event generator information is used to distinguish events with prompt leptons from events with non-prompt or fake leptons. The normalization of this background was obtained from data by using a dedicated control region. This control region uses the same basic event selection as the signal and validation regions, the only difference being that the electric charges of the electron and muon must have the same sign. Within this control region, the number of simulated prompt-lepton events is subtracted from the observed number of data events. The number of events remaining is then divided by the number of simulated fake-lepton events, resulting in a normalization factor of 1.4. This scale factor is then applied to the simulated fake-lepton events in the signal and validation regions.

## Systematic uncertainties

The systematic uncertainties can be divided into three separate categories: signal modelling uncertainties, which stem from the theory prediction of $t\bar{t}$ production; object systematic uncertainties, which arise from the uncertainty in the detector response to objects used in the analysis; and background modelling systematic uncertainties, which are related to the theory prediction of the standard model backgrounds. All systematic uncertainties, grouped according to their sources, are described in the following sections. The signal modelling uncertainties were found to dominate the overall uncertainty of this measurement.

For each source of systematic uncertainty, a new calibration curve is created and the simulated (or observed) data are corrected, resulting in a shifted corrected result. In most cases, the systematic uncertainty is taken to be the difference between the nominal expected and observed result and the systematically shifted result. In cases in which a systematic shift only affects the background model (for example, background cross-section uncertainties), the systematically shifted background sample is subtracted from the data instead before the calibration is performed. In cases in which the systematic uncertainty is one-sided, the uncertainty is symmetrized. In cases in which the uncertainties are asymmetric, the larger of the two variations is symmetrized. The signal modelling uncertainties dominate the measurement, and their estimated sizes are presented in Extended Data Table 1.

**Signal modelling uncertainties.** Signal modelling uncertainties are those related to the choice of POWHEG BOX + PYTHIA as the nominal Monte Carlo setup as well as those affecting the theoretical calculation itself. These systematic uncertainties are considered in two forms: alternative event generators and weights. For the alternative-generator uncertainties, the difference between the calibrated values of $D$ is taken as the systematic uncertainty. For the systematic uncertainties involving weights, the difference between the calibrated $D$ values for the nominal sample and the weight-shifted sample is taken as the uncertainty. These uncertainties follow the description in ref. 104 and are enumerated as follows:

- pThard setting: the region of phase space that is vetoed in the showering when matched to a parton shower is varied by changing the internal pThard parameter of POWHEG BOX from 0 to 1, as described in ref. 105.
- Top-quark decay: the uncertainty in the modelling of the decay of the top quarks and of the $m_{t\bar{t}}$ line shape is estimated by comparing the nominal decay in POWHEG BOX with the decays modelled with MADSPIN (refs. 100,101). The effect of this uncertainty is to shift the $m_{t\bar{t}}$ line shape to lower or higher values that alter the degree of entanglement entering the signal region. Thus, this is one of the most impactful sources of systematic uncertainty.

- NNLO QCD + NLO EW reweighting: the uncertainty due to missing higher-order corrections is estimated by reweighting the $p_T$ of the top quarks, the $p_T$ of the $t\bar{t}$ system and the $m_{t\bar{t}}$ spectra at parton level to match the predicted NNLO QCD and NLO EW differential cross-sections[106,107].
- Parton shower and hadronization: this uncertainty is estimated by comparing two different parton-shower and hadronization algorithms, PYTHIA and HERWIG, interfaced with the same matrix element event generator (POWHEG BOX).
- Recoil scheme: the nominal sample uses a recoil scheme in which the partons recoil against $b$-quarks. This recoil scheme changes the modelling of the second and subsequent gluon emissions from quarks produced by coloured resonance decays, such as the $b$-quark in a top-quark decay, and therefore affects how the momentum is rearranged between the $W$ boson and the $b$-quark. An alternative sample is produced in which the recoil is set to be against the top quark itself for the second and subsequent emissions[108].
- Scale uncertainties: the renormalization and factorization scales are raised and lowered by a factor of 2 in the nominal POWHEG setup, including simultaneous variations in the same direction. The envelope of results from all of these variations is taken as the final uncertainty.
- Initial-state radiation: The uncertainty due to initial-state radiation is estimated by choosing the Var3c up/down variations of the A14 tune as described in ref. 109.
- Final-state radiation: the impact of final-state radiation is evaluated by doubling or halving the renormalization scale for emissions from the parton shower.
- PDF: the systematic uncertainty due to the choice of PDF is assessed using the PDF4LHC15 eigenvector decomposition[110]. The full difference between the results from the nominal PDF and the varied PDF is taken and symmetrized for each of the 30 eigenvectors. The quadrature sum of all result variations is provided in Extended Data Table 1.
- $h_{\mathrm{damp}}$ setting: the $h_{\mathrm{damp}}$ parameter is a resummation damping factor and one of the parameters that control the matching of POWHEG BOX matrix elements to the parton shower and thus effectively regulates the high-$p_T$ radiation against which the $t\bar{t}$ system recoils. The systematic uncertainty due to the chosen value of the $h_{\mathrm{damp}}$ parameter is assessed by comparing the nominal POWHEG+ PYTHIA result with one in which the $h_{\mathrm{damp}}$ parameter is increased by a factor of two.
- Top-quark mass: the effect of the top-quark mass uncertainty is examined by comparing the nominal sample with alternative samples that use $m_t = 172$ GeV or 173 GeV in the simulation.

**Object systematic uncertainties.** Systematic uncertainties that originate from the uncertainty in the detector response to the objects used in the analysis are estimated.
- Electrons: The systematic uncertainties considered for electrons arise mainly from uncertainties in their trigger, reconstruction, identification and isolation efficiencies and are estimated using tag-and-probe measurements in $Z$ and $J/\psi$ decays[56,111]. Electron-related systematic uncertainties have a negligible impact on the final measurement, with a total contribution of about 0.2%.
- Muons: The systematic uncertainties considered for muons arise from uncertainties in their trigger, identification and isolation efficiencies, and their energy scale and resolution, and are estimated using tag-and-probe measurements in $Z$ and $J/\psi$ decays[57–59]. Muon-related systematic uncertainties have a negligible impact on the final measurement, with a total contribution of about 0.3%.
- Jets: The systematic uncertainties associated with jets are separated into those related to the jet-energy scale and resolution (JES and JER)[60] and those related to the JVT algorithm[64]. The JES uncertainty consists of 31 individual components and the JER uncertainty consists of 13 individual components that are added in quadrature with the JVT uncertainty to obtain the total jet uncertainty. The largest contribution from a single source is 0.2%.

- *b*-Tagging: The estimation of these uncertainties is described in ref. 112. A total of 17 independent systematic variations are considered: 9 related to *b*-hadrons, 4 related to *c*-hadrons, and 4 related to light-jet misidentification. Furthermore, two high-$p_T$ extrapolation uncertainties are taken into account. The largest contribution from a single systematic variation is 0.4%.

- $E_T^{miss}$: All object-based uncertainties are fully correlated with the reconstruction of the $E_T^{miss}$ object of the event, the magnitude of the $\mathbf{p}_T^{miss}$ vector. However, there are some uncertainties specific to the reconstruction of $E_T^{miss}$ that concern soft tracks not matched to leptons or jets. These uncertainties are divided into parallel and perpendicular response components as well as a scale uncertainty[66]. These have a negligible effect on the measurement.

- Pile-up: The effect of pile-up was modelled by overlaying the simulated hard-scattering events with inelastic *pp* events. To assess the systematic uncertainty due to pile-up, the reweighting performed to match simulation to data is varied within its uncertainty[64]. The resulting uncertainty has an effect of less than 0.1%.

- Luminosity: The luminosity uncertainty only changes the normalization of the signal and background samples. The value of *D* is calculated from the normalized cos φ distribution and, therefore, is not affected by varying the sample normalization. However, the total expected statistical uncertainty can be affected by the luminosity uncertainty. This analysis uses the latest integrated luminosity estimate of $140.1 \pm 1.2$ fb$^{-1}$ (ref. 113). Its uncertainty affects the measurement by less than 0.1%.

**Background modelling systematic uncertainties.** Background events are a relatively small source of uncertainty in this measurement because the event selection and top-quark reconstruction, especially the $m_{t\bar{t}}$ constraint, tend to suppress them. The uncertainties and their sources are listed in the following:

- Single top quark: two uncertainties are considered for the single-top quark background: a cross-section uncertainty of 5.3% based on the NNLO cross-section uncertainty[85] and an uncertainty for the choice of schemes used to remove higher-order diagrams that overlap with the $t\bar{t}$ process. For the latter, the nominal POWHEG + PYTHIA sample, generated with the diagram-removal scheme[83], was compared with an alternative sample generated using the diagram-subtraction scheme[73,83]. The cross-section uncertainty has a 0.4% effect on the measurement, whereas the choice of diagram scheme has less than 0.1% effect on the measurement.

- $t\bar{t} + X$: a normalization uncertainty is considered for each of the $t\bar{t} + X$ backgrounds: a cross-section uncertainty of $^{+10\%}_{-12\%}$ for $t\bar{t} + Z$ and $^{+13\%}_{-12\%}$ for $t\bar{t} + W$. Both are based on the NLO cross-section uncertainty derived from the renormalization and factorization scale variations and PDF uncertainties in the matrix element calculation. These uncertainties have a negligible effect on the measurement because the $t\bar{t} + X$ processes make a very small contribution to the signal region.

- Diboson: a normalization uncertainty of ±10% is considered for the diboson process to account for the difference between the NLO precision of the SHERPA event generator and the precision of the theoretical cross-sections calculated to NNLO in QCD with NLO EW corrections. This simple *K*-factor approach is taken, rather than a more elaborate prescription, because the diboson background is small and the phase space selected by the analysis ($m_{t\bar{t}} < 380$ GeV) is unlikely to be sensitive to shape effects in the EW corrections, typically observed in high-$p_T$ tails. This uncertainty has less than 0.1% effect on the measurement.

- $Z \to \tau\tau$: a conservative cross-section uncertainty of ±20% is applied to the $Z \to \tau\tau$ background to account for the uncertainty in the cross-section prediction (which is much smaller than this variation) as well as to account for some mismodelling of the rate of associated heavy-flavor production, which is typically seen in *ee* and *μμ* dileptonic $t\bar{t}$ analyses and was estimated to be a 5% (3%) effect in previous iterations of this analysis that included the *ee* (*μμ*) channel. This assumption is conservative as it is not possible to isolate a pure $Z \to \tau\tau$ control region in which to estimate this effect, and therefore additional lepton-flavor-related effects present in the *ee* and *μμ* channels are also being included. This uncertainty has a noticeable impact on the final measurement, becoming the largest background-related uncertainty. It becomes large, despite this background being relatively small, because the reconstruction-level $Z \to \tau\tau \cos\varphi$ distribution is quite flat and, therefore, subtracting even a relatively small amount of $Z \to \tau\tau$ background can noticeably affect the mean of the overall cos φ distribution and therefore the *D* observable. This uncertainty has an impact of 0.8% on the measurement.

- Fake and non-prompt leptons: a normalization uncertainty of ±50% is assigned to account for the uncertainty in the total yield of fake or non-prompt leptons in the signal region compared with the same-sign control region to ensure adequate coverage for our understanding of the rates of these types of events. It is a conservative uncertainty based on the observed level of data and Monte Carlo agreement in the same-sign region. The uncertainty has only a 0.1% effect on the final measurement.

Most of the systematic uncertainties that are considered are inconsequential to the measurement, and the dominant systematic uncertainties arise mostly from the signal modelling. These findings are true for the validation regions as well.

### Parton shower and hadronization effects

The studies described in the following were performed to gain a more detailed understanding of why the different parton-shower and hadronization algorithms yield different values for the entanglement- and spin-correlation-related observables. The nominal Monte Carlo sample was produced with the NLO matrix element implemented in POWHEG BOX (hvq). The four momenta produced with POWHEG BOX were interfaced with either PYTHIA or HERWIG for the parton shower, hadronization and underlying-event model.

At the parton level, the two predictions are nearly identical, whereas at the stable-particle and detector levels, the two predictions show larger differences in the shape of the cos φ distributions. A parton-level measurement would, therefore, suffer from the ambiguity in cos φ, whereas the particle-level measurement presented in this paper does not. An extensive suite of studies was performed to understand the origin of this difference.

Apart from using different parameter-tuning strategies, there are two main differences between the two parton-shower algorithms: their hadronization model and the shower ordering. Whereas PYTHIA is based on the Lund string model and uses a $p_T$-ordered dipole shower[114–116], the HERWIG samples used in this study are based on a cluster model and use an angular-ordered shower as the default[117].

A comparison between Monte Carlo simulations with different hadronization models was performed. For one study, SHERPA was used with either a string or a cluster model for hadronization. For the other study, HERWIG 7 was used, again comparing the effects of using either a string or a cluster model. Changing the hadronization model has shown in both cases to have a negligible effect on the cos φ distribution, both when not placing a cut on $m_{t\bar{t}}$ and when using a smaller part of phase space close to the signal region of the analysis, with $m_{t\bar{t}} < 380$ GeV. Instead, most of the differences seem to originate from the different orderings in the parton shower. To illustrate this, different event generator setups were used for simulation and the corresponding cos φ distributions were compared at particle level. The cos φ distributions for the POWHEG + PYTHIA and POWHEG + HERWIG samples used in the analysis are shown in Extended Data Fig. 2a, together with distributions for two different setups of HERWIG 7 in Extended Data Fig. 2b. In these setups, HERWIG 7 was used both for the production of the $t\bar{t}$ events and for the parton shower, hadronization and underlying event.

The samples were produced at LO, using either a dipole shower or an angular-ordered shower. All distributions are normalized to unity. A difference of up to 6% is observed when examining the ratio of POWHEG + HERWIG to POWHEG + PYTHIA distributions. The same behaviour is observed when comparing the two different showering orders for HERWIG.

The similarities between the samples used in this analysis and the HERWIG samples with different showering orders imply that the ordering of the shower is the main cause of the observed differences. It has to be noted, however, that POWHEG does not pass the spin correlation information to the parton shower algorithms, whereas this is done in the LO HERWIG setup used to study these hadronization effects.

These findings lead to the conclusion that performing the measurement at the particle level is more attractive because the difference in the predictions while extrapolating from the parton to particle level can be isolated and not taken as full systematic uncertainty. In the validation regions, the level of agreement between either POWHEG + PYTHIA or POWHEG + HERWIG and the data are similar. As the measurement is performed at the stable-particle level, the parton-level prediction for the entanglement limit was folded to the particle level as well, using a special calibration curve for this step. The prediction for the entanglement limit with POWHEG + HERWIG is further away from the data measurement than the one for POWHEG + PYTHIA. This difference is not symmetrized. All uncertainties in the POWHEG + PYTHIA prediction itself are folded to the particle level as well and are included in the grey uncertainty band in Fig. 2b.

The procedure used in Monte Carlo event generators to combine the matrix element with a parton-shower algorithm requires special attention in future higher-precision quantum information studies at the LHC.

## Data availability

Raw data were generated by the ATLAS experiment. Derived data supporting the findings of this study are available from the ATLAS Collaboration upon request.

## Code availability

The ATLAS data reduction software is available at *Zenodo* (https://doi.org/10.5281/zenodo.4772550) (ref. 118). Statistical modelling and analysis are based on the ROOT software and its embedded RooFit and RooStats modules, available at *Zenodo* (https://doi.org/10.5281/zenodo.3895852) (ref. 119). Code to configure these statistical tools and to process their output is available upon request.

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

**Acknowledgements** We thank CERN for the very successful operation of the LHC and its injectors, as well as the support staff at CERN and at our institutions worldwide without whom ATLAS could not be operated efficiently. The crucial computing support from all WLCG partners is acknowledged, in particular, from CERN, the ATLAS Tier-1 facilities at TRIUMF/SFU (Canada), NDGF (Denmark, Norway and Sweden), CC-IN2P3 (France), KIT/GridKA (Germany), INFN-CNAF (Italy), NL-T1 (the Netherlands), PIC (Spain), RAL (UK) and BNL (USA), the tier-2 facilities worldwide and large non-WLCG resource providers. Major contributors to computing resources are listed in ref.120. We acknowledge the support of ANPCyT, Argentina; YerPhI, Armenia; ARC, Australia; BMWFW and FWF, Austria; ANAS, Azerbaijan; CNPq and FAPESP, Brazil; NSERC, NRC and CFI, Canada; CERN; ANID, Chile; CAS, MOST and NSFC, China; Minciencias, Colombia; MEYS CR, Czech Republic; DNRF and DNSRC, Denmark; IN2P3-CNRS and CEA-DRF/IRFU, France; SRNSFG, Georgia; BMBF, HGF and MPG, Germany; GSRI, Greece; RGC and Hong Kong SAR, China; ISF and Benoziyo Center, Israel; INFN, Italy; MEXT and JSPS, Japan; CNRST, Morocco; NWO, the Netherlands; RCN, Norway; MEiN, Poland; FCT, Portugal; MNE/IFA, Romania; MESTD, Serbia; MSSR, Slovakia; ARRS and MIZŠ, Slovenia; DSI/NRF, South Africa; MICINN, Spain; SRC and Wallenberg Foundation, Sweden; SERI, SNSF and Cantons of Bern and Geneva, Switzerland; MOST, Taipei; TENMAK, Türkiye; STFC, UK; DOE and NSF, USA. Individual groups and members have received support from BCKDF, CANARIE, CRC and DRAC, Canada; CERN-CZ, PRIMUS 21/SCI/017 and UNCE SCI/013, Czech Republic; COST, ERC, ERDF, Horizon 2020, ICSC-NextGenerationEU and Marie Skłodowska-Curie Actions, European Union; Investissements d'Avenir Labex, Investissements d'Avenir Idex and ANR, France; Herakleitos, Thales and Aristeia programmes co-financed by EU-ESF and the Greek NSRF, Greece; BSF-NSF and MINERVA, Israel; Norwegian Financial Mechanism 2014–2021, Norway; NCN and NAWA, Poland; La Caixa Banking Foundation, CERCA Programme Generalitat de Catalunya and PROMETEO and GenT Programmes Generalitat Valenciana, Spain; Göran Gustafssons Stiftelse, Sweden; The Royal Society and Leverhulme Trust, UK. Moreover, individual members wish to acknowledge support from CERN: European Organization for Nuclear Research (CERN PJAS); Chile: Agencia Nacional de Investigación y Desarrollo (FONDECYT 1190886, FONDECYT 1210400, FONDECYT 1230812 and FONDECYT 1230987); China: National Natural Science Foundation of China (NSFC - 12175119, NSFC 12275265, NSFC-12075060); Czech Republic: PRIMUS Research Programme (PRIMUS/21/SCI/017); European Union: European Research Council (ERC - 948254), Horizon 2020 Framework Programme (MUCCA - CHIST-ERA-19-XAI-00), European Union, Future Artificial Intelligence Research (FAIR-NextGenerationEU PE00000013), Italian Center for High Performance Computing, Big Data and Quantum Computing (ICSC, NextGenerationEU), Marie Sklodowska–Curie Actions (EU H2020 MSC IF grant no. 101033496); France: Agence Nationale de la Recherche (ANR-20-CE31-0013, ANR-21-CE31-0013 and ANR-21-CE31-0022), Investissements d'Avenir Idex (ANR-11-LABX-0012), Investissements d'Avenir Labex (ANR-11-LABX-0012); Germany: Baden–Württemberg Stiftung (BW Stiftung-Postdoc Eliteprogramme), Deutsche Forschungsgemeinschaft (DFG - 469666862, DFG - CR 312/5-1); Italy: Istituto Nazionale di Fisica Nucleare (FELLINI G.A. no. 754496, ICSC, NextGenerationEU); Japan: Japan Society for the Promotion of Science (JSPS KAKENHI JP21H05085, JSPS KAKENHI JP22H01227, JSPS KAKENHI JP22H04944 and JSPS KAKENHI JP22KK0227); the Netherlands: The Netherlands Organisation for Scientific Research (NWO Veni 2020 - VI.Veni.202.179); Norway: Research Council of Norway (RCN-314472); Poland: Polish National Agency for Academic Exchange (PPN/PPO/2020/1/00002/U/00001), Polish National Science Centre (NCN 2021/42/E/ST2/00350, NCN OPUS no. 2022/47/B/ST2/03059, NCN UMO-2019/34/E/ST2/00393, UMO-2020/37/B/ST2/01043, UMO-2021/40/C/ST2/00187); Slovenia: Slovenian Research Agency (ARIS grant J1-3010); Spain: BBVA Foundation (LEO22-1-603), Generalitat Valenciana (Artemisa, FEDER, IDIFEDER/2018/048), La Caixa Banking Foundation (LCF/BQ/PI20/11760025), Ministry of Science and Innovation (MCIN and NextGenEU PCI2022-135018-2, MICIN and FEDER PID2021-125273NB, RYC2019-028510-I, RYC2020-030254-I, RYC2021-031273-I and RYC2022-038164-I), PROMETEO and GenT Programmes Generalitat Valenciana (CIDEGENT/2019/023, CIDEGENT/2019/027); Sweden: Swedish Research Council (VR 2018-00482, VR 2022-03845, VR 2022-04683 and VR grant 2021-03651), Knut and Alice Wallenberg Foundation (KAW 2017.0100, KAW 2018.0157, KAW 2018.0458 and KAW 2019.0447); Switzerland: Swiss National Science Foundation (SNSF - PCEFP2_194658); UK: Leverhulme Trust (Leverhulme Trust RPG-2020-004); USA: US Department of Energy (ECA DE-AC02-76SF00515), Neubauer Family Foundation.

**Author contributions** All authors have contributed to the publication, being variously involved in the design and the construction of the detectors, in writing the software, calibrating subsystems, operating the detectors and acquiring data and finally analysing the processed data. The ATLAS Collaboration members discussed and approved the scientific results. This Article was prepared by a subgroup of authors appointed by the ATLAS Collaboration and subjected to an internal collaboration-wide review process. All authors reviewed and approved the final version of the paper.

**Competing interests** The authors declare no competing interests.

**Additional information**
**Correspondence and requests for materials** should be addressed to The ATLAS Collaboration.

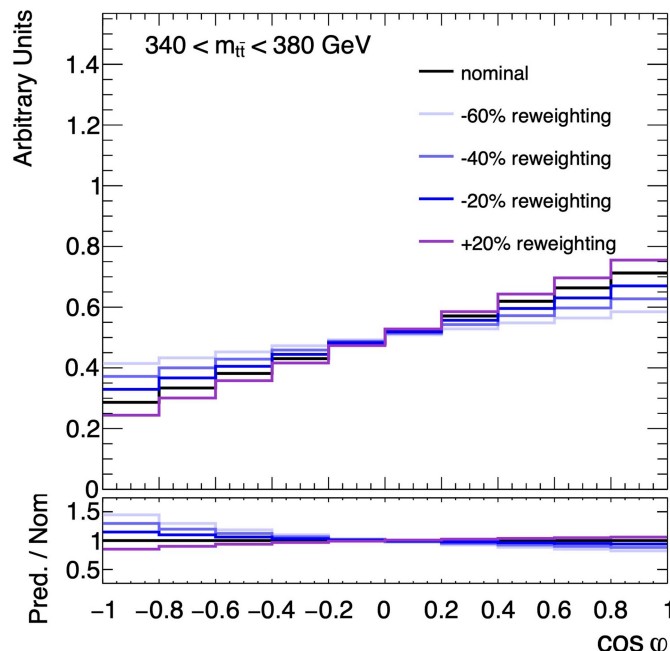

**Extended Data Fig. 1 | Example of reweighting technique.** Example of the nominal cos$\varphi$ distribution and the results of applying the reweighting technique with $\mathcal{X}$ = 0.4, 0.6, 0.8, 1.2 in the signal region at parton level. The lower panel shows the ratio of each $D$ value after reweighting ('Pred.') to the nominal $D$ value ('Nom.').

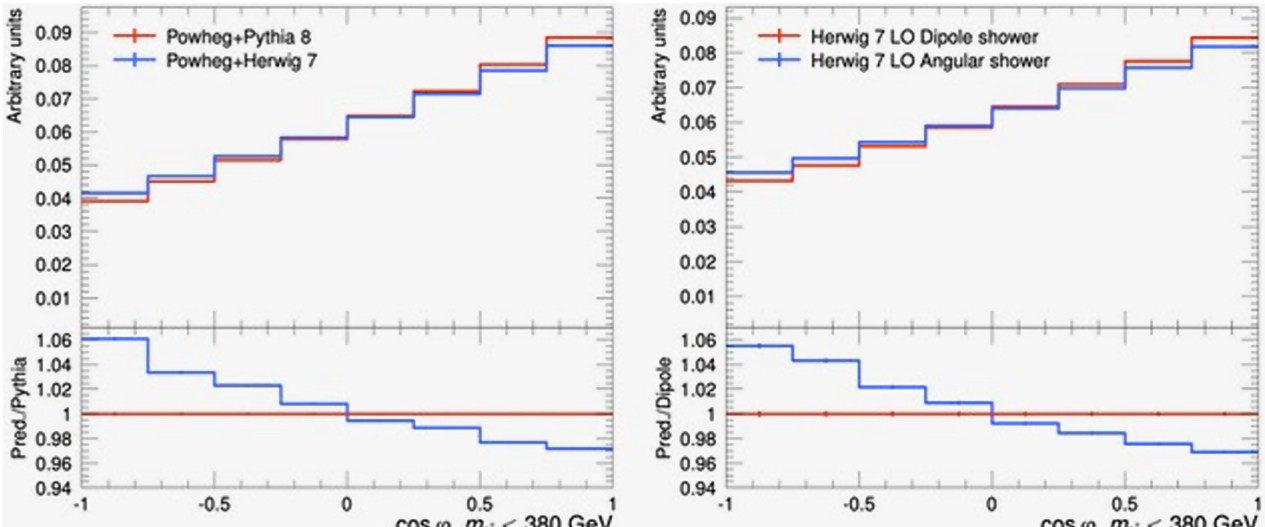

**Extended Data Fig. 2 | Parton shower generator studies.** Comparison between $\cos\varphi$ distributions in the signal region with $m_{t\bar{t}} < 380$ GeV for different Monte Carlo event generator setups at stable-particle level. Figure (a) compares events simulated with POWHEG BOX which are interfaced with either PYTHIA (red line, $p_T$-ordered dipole shower) or HERWIG (blue line, angular-ordered shower) while figure (b) compares events simulated with HERWIG using either a dipole-ordered shower (red line) or an angular-ordered shower (blue line).

**Extended Data Table 1 | Summary of modelling uncertainties**

| Systematic uncertainty source | Relative size (for SM $D$ value) |
|---|---|
| Top-quark decay | 1.6% |
| Parton distribution function | 1.2% |
| Recoil scheme | 1.1% |
| Final-state radiation | 1.1% |
| Scale uncertainties | 1.1% |
| NNLO QCD + NLO EW reweighting | 1.1% |
| pThard setting | 0.8% |
| Top-quark mass | 0.7% |
| Initial-state radiation | 0.2% |
| Parton shower and hadronization | 0.2% |
| $h_{damp}$ setting | 0.1% |

Relative sizes of the signal modelling uncertainties at the standard model expectation point $D_{particle} = -0.47$ for the nominal POWHEG BOX sample.