## [Peer Review file · Nature]

Manuscript Title: Observation of quantum entanglement with top quarks at the ATLAS detector

Reviewer Comments & Author Rebuttals

Reviewer Reports on the Initial Version:

Referee #1 (Remarks to the Author):

Dear authors,

Thank you for submitting this very well written paper. It was a pleasure to read. With very few exceptions, the steps of the analysis are well explained and easy to follow.

The paper describes an extraction of the entanglement marker D at particle-level through a measurement of the spin correlations between the two leptons stemming from top-quark pair decays, where both W bosons decay leptonically. For that purpose, the angle between those leptons in the top-quark-pair rest frame is determined and interpreted in terms of the entanglement marker D . While strictly speaking, the relation of this quantity to the entanglement between the top quark pairs is only defined at parton level, it is here evaluated at particle level to avoid large differences observed between angular and p_T ordered parton showers affecting the sensitivity of the result. The particle level result is then interpreted by translating the expected parton level D to particle level. To maximise the sensitivity to entanglement in the system, events are selected that are close to the top-quark pair production threshold, where entanglement is maximal.

The result is relevant and represents a new way to interpret spin-correlation measurements (that are among the standard top-quark pair analyses since the Tevatron), here focusing on the threshold region. Otherwise, the measurement is following well established procedures and as such is valid and well presented. Also, statistical methods are used appropriately and uncertainties are assigned following the standard procedures.

However, there are some points that I think need to be investigated or better described before a possible publication. They can be divided into two major aspects: one is the modelling of the threshold region, the other one is the calibration between parton, particle, and reconstruction level.

1) The modelling of the threshold region needs additional studies:

This measurement focuses on the threshold region to enhance the sensitivity. In turn, effects that might not play a significant role in the bulk of the events used in a typical top-quark pair analysis can be significantly enhanced here and need to be accounted for.

On page 9, you clearly state yourselves that non-relativistic effects can lead to (pseudo) bound state effects close to threshold. These bound state effects increase the entanglement significantly, not only by changing the $\cos(\phi)$ distribution, but also by enhancing the contribution of the region most sensitive to entanglement in the signal model. The argument made here - that there are other effects that also change the $\cos(\phi)$ distribution and therefore the uncertainty is covered - is in itself invalid. All systematic uncertainties that have an effect on D change the $\cos(\phi)$ distribution (otherwise they would not have an effect); therefore the individual sources need to be accounted for. While many other systematic uncertainties are described in detail and quantitatively, this important aspect is only mentioned shortly, without giving the reader a quantitative estimate.

N.B: In this context, it should not be forgotten that also electroweak effects contribute to the threshold region kinematics. They are strong enough to allow constraints on the top-quark Yukawa coupling in other analyses, but don't seem to be discussed in this paper explicitly, nor are they explicitly accounted for.

The bound-state effects can change the conclusion of the paper quite significantly: (a) they can affect the calibration curves themselves by enhancing particular regions of phase space with events and therefore lead to different acceptance and efficiency corrections. This is in particular noteworthy given the (poor) resolution in terms of $m_{T\bar{T}}$ and the very narrow window around the threshold.

And (b), the bound-state effects will change the translation of the entanglement limit from parton to particle level, by giving rise to a potentially significantly lower D limit on particle level - potentially affecting the significance and therefore changing the main result of the paper.

Therefore, I consider it as crucial to further study and quantify (upper limits on) these effects before the paper can be accepted. Even though the picture is still incomplete from a theory perspective, the results of the following references could be used to estimate limits on the potential effects: e.g. arXiv:1007.0075, arXiv:2004.03088, arXiv:2102.11281.

The submitted paper already includes a thorough study of the different parton shower models that are much less directly related to entanglement, and I would request a similar study on these threshold effects that are much more directly related to the final result.

2) The calibration of D should be better explained or revisited.

Here, you describe how the different levels - parton, particle, and detector level - are defined and can be related to each other via calibration curves. However, it still becomes confusing at times. Since this is a central part of the paper, I would suggest to define very clearly what all levels are in a central place, even though it may seem obvious, e.g. that $D_X = \dots$ (or similar). The purpose is to write more clearly that e.g. D on reconstruction level "uses the $\cos(\phi)$ distribution on reconstruction level". In particular for readers outside of high energy physics, this would improve the readability. Then, I would suggest to consistently use D_X wherever applicable. This may also apply to the abstract. While "particle level" is written in the text there, using a subscript could help to avoid people glossing over it and interpreting the " $D=\dots$ " as parton-level D .

What I don't fully understand in the context of calibration is the distinction made between the different parton shower models. The differences are considered a "real" uncertainty when going from reconstruction level to particle level. However, then their role is diminished to being alternative models to compare the particle level D value to. Wouldn't it be more consistent to treat them as one or the other throughout the extraction (irrespective of the question if indeed such a 2-point comparison represents a well-defined uncertainty)? If the authors do not want to consider the differences a full uncertainty, one could e.g. give different results for the different models; or give one result where these effects are treated consistently as uncertainties. At the moment, and from the outside, the justification for the current scheme is not entirely clear.

Another aspect of the calibration curves is that there are correlations between uncertainties when performing the calibration. Some uncertainties will appear when going from reconstruction to particle level, and similar ones when going from particle level to parton level. This discussion is currently missing from the paper, but also affects the final significance. I am assuming the authors have already quantified the level of correlation and I think it should be quoted in the paper text.

For the parametrisation described in A.3, would it be possible to visualise or quantify how well this parametrisation works, or justify the choice of polynomial more such that the reader can understand what level of possible approximation can be expected from this parametrisation?

Also, a discussion of the limitations of this parameterisation in only two dimensions would be useful. The detector and reconstruction resolution is insufficient to neglect detector smearing effects, that are meant to be accounted for fully by the calibration curves for D. However, the detector smearing could also lead to a situation where variables other than only D and m_{tt} may play a role for the total effect of reduced or enhanced entanglement on the reconstruction level or particle level distributions. This could be part of the reason we see such large differences between the parton shower models.

While the statement that "The degree of entanglement is intrinsic in the calculations of the MC event generators and cannot be changed" is not wrong when applied to the calculations that do/have to incorporate entanglement, it is technically possible to break the entanglement by hand in the generator. I suggest to use such an approach as cross-check against the reweighting method employed in this paper, which could also shed more light on the parton shower discussion.

3) Minor aspects:

In the introduction, it would be useful to acknowledge that in the context of spin-correlation measurements, the D parameter has been measured before.

In the description of the systematic uncertainties (A.5.1), some uncertainties are well explained including the reason why they are considered uncertainties, while others lack explanation. In particular, the top-quark decay uncertainty, which is leading, could use a more in-depth explanation why the difference between Powheg and Madspin is considered an uncertainty - the same is true for the parton shower uncertainty (but here the authors could simply refer to section A.6). In general, it would help in particular the non-expert reader to understand the rationale behind choosing these variations. This is explained to some extent e.g. for the recoil uncertainty, but also here a better justification why both choices could represent reality (or are edge-cases with the true value likely in-between) would be very helpful.

I would suggest to weaken the statement "The degree of entanglement is intrinsic in the calculations of the MC event generators and cannot be changed" for the reasons outlined above.

Referee #2 (Remarks to the Author):

In the article “Observation of quantum entanglement in top-quark pairs using the ATLAS detector”, the authors claim the highest-energy observation of spin entanglement in top-antitop pairs measured in the ATLAS detector at the LHC. The measurement is based on an entanglement witness parameter, D , which is proportional to the event average of the opening angle between the two leptons, originating from the top and antitop decays, measured at the rest frame of each parent top (antitop). To reconstruct the top (antitop) rest frame, unknown neutrino momenta must be reconstructed/inferred consistently with the event kinematics. The D observable is significantly affected by the effects of parton-shower+hadronization as well as the finite detector resolution. To correct these effects, the authors introduced a calibration curve, which is a linear map from the detector(particle)-level D to the particle(parton)-level D . An event reweighting method was used to obtain these calibration curves. In the signal region, the authors found that the measured (particle-level) D value indicates the presence of entanglement beyond the 5-sigma level. The authors also observed that the simulated (particle-level) D values using Pythia’s pT-ordered shower and Herwig’s angular-ordered shower are significantly different. The entanglement threshold for particle-level D suffers from large uncertainty stemming from the dependency of parton-shower algorithms: the D threshold values are -0.322 for the Pythia shower, while it is -0.27 for the Herwig shower.

Overall, the analysis presented in the article is serious and thorough, and the measurement and result are innovative and impactful. The references appropriately credited the previous works. On these bases, the paper should, in principle, be published. However, in my eyes, some parts of the analysis description are not clear, and they should be improved before publication. I would like the authors to clarify the following points.

1) In the analysis, it is crucial to reconstruct each top (antitop) rest frame. However, the recipe for this reconstruction is not well described. The authors should give the following information.

[1-1] First of all, the main problem is the following. There are 6 unknown neutrino momentum components. Ignoring the W and top widths, they are constrained by the 2 W -boson mass-shell constraints, 2 top (antitop) mass-shell constraints and 2 momentum imbalance constraints (corresponding to the two transverse directions). Since the mass-shell constraints are quadratic equations for the neutrino momenta, generally, there are two solutions for neutrino momenta. However, due to imperfect detector resolution, there are cases where the measured event provides complex solutions (rather than real). I understand this is the essence of the ellipse method. A gist of the ellipse method should be provided.

Which solution is taken if there are two real solutions?

[1-2] If the ellipse method does not give real solutions, the neutrino weighting method is used. A brief description of this method should be provided. In ref. [63], I have an impression that this method relies on the probability density of the lepton energy predicted by the Standard Model. My concern is that if the analysis uses Standard Model (SM) information, reconstructed distributions will be biased such that they show entanglement because the entanglement is present in the SM. The authors should justify their method in this regard.

[1-3] On page 11, the authors write, "If both methods fail, a simple pairing of each lepton with its closest b-tagged jet is used." This is about the combinatorics and not neutrino momentum reconstruction. How are the neutrino momenta reconstructed if both methods fail?

2) In section A.6, a study on parton-shower and hadronization effects is provided.

[2-1] Several parton-shower algorithms are compared in Fig.4. Based on this, the authors concluded that the main cause of the MC generator dependence is the parton-shower algorithms, rather than the hadronization models. To conclude this, the same hadronization models must be used among the distributions shown in Fig.4. Which hadronization models are used?

[2-2] If different hadronization models are used in Fig.4, I have to rely on a statement on page 19, "A comparison between MC simulations with different hadronization models has shown that these have a negligible effect on the $\cos(\phi)$ distribution". This sentence should be elaborated. Which MC generators (parton-shower algorithms) are used for this comparison? How small was the effect?

3) On page 19, the authors write, "parton-level measurement would therefore suffer from the ambiguity in $\cos(\phi)$, while the particle-level measurement presented in this paper does not." On page 20, "These findings lead to the conclusion that performing the measurement at particle level is more attractive, since the overall uncertainties are smaller".

I do not understand these sentences. In the particle level, measured D has a smaller error, but the entanglement threshold has a larger error. In the parton level, measured D has a larger error, while the entanglement threshold has no error ($= -1/3$). It seems to me that they are equally powerful (less powerful) in terms of the entanglement measurement. The authors should clarify this point.

4) This is a minor point. On page 11, "they must then lie within $\Delta R > 0.4$ from a jet to avoid being removed from the event."

Probably, the authors want to mean that "they must not lie within $\Delta R < 0.4$ " (?)

In the detector-level analysis, I do not find the corresponding lepton removal procedure.

Why is this procedure implemented only in the parton-level analysis?

Referee #3 (Remarks to the Author):

Summary of key results: The submitted draft "Observation of quantum entanglement in top-quark pairs using the ATLAS detector" reports the observation of entanglement by the ATLAS collaboration at the Large Hadron Collider (LHC) in top-antitop quark events. Entanglement is observed for spin part of the wave function of the top-antitop quark pair through the measurement of a certain observable D , which is related to the averaged cosine of the (boosted) angle between the charged lepton directions; the latter are the end products of the subsequent decays of top quarks into a bottom quark and a W boson, which then decays into a charged lepton and a neutrino. Entanglement is observed for $D < -1/3$, while the ATLAS collaboration finds $D = -0.547 \pm 0.002$ (statistical error) ± 0.021 (systematic error). The obtained result for D is therefore more than 5 standard deviations away from the scenario without entanglement ($D > -1/3$).

Originality and significance: The result presented by the authors provides a first demonstration that quantum entanglement can be studied in the environment of a high energy collider. I am convinced that the result is both original and significant. I however would like the authors to clarify some points, before I would like to give my final judgement on this point (see "Suggested Improvements")

Data & methodology: validity of approach, quality of data, quality of presentation:

The authors mention that they use LO QCD matrix elements in their description. Also Ref. 18 (on which much/all of the theory part is based on) uses LO QCD matrix for the discussion. The authors should clarify whether the condition $D < -1/3$ is valid within a leading order QCD approximation only or whether it is a non-perturbative/all order statement i.e. a generic feature. If the statement is only valid within leading order QCD perturbation theory, they should provide (if possible) an estimate of uncertainties due to possible higher order corrections or (if not possible) acknowledge this in their article. The latter case would of course diminish the relevance of the observed result (at least if no estimate of higher order corrections can be provided)

Suggested improvements:

I understand that the result presented by the authors should be understood as a first demonstration that quantum entanglement can be studied in the environment of a high energy collider. While I agree that observation of spin entanglement at LHC is already a noteworthy achievement, I wonder whether the result has deeper implication i.e. which future explorations might be possible in a collider environment which cannot be achieved in conventional laboratory experiments (e.g. setups in which quantum information experiments are being carried out usually). This is partly addressed in Ref. [17], but the importance of their result would become more apparent if they could also point that out in the draft (if this is the case).

I miss in the introduction an explanation why the LHC is a particularly useful environment for the study of entanglement of pairs of quarks? Naively one might assume that an electron positron collider would be more suitable since it allows a better control of the initial state. The only possible answer I can find is that LHC allows for the production of top antitop pairs due to its high center of mass energy, while the top does not hadronize and therefore constitutes the only particle that allows for observation of those effects. While all this is somehow mentioned, I would recommend to highlight these points or supplement them if appropriate.

On page 3, 3rd paragraph, the authors state that "Entanglement is observed with a significance of more than five standard deviations for the first time in pairs of quarks." Entanglement is however only defined as "If two particles are entangled, the quantum state of one particle cannot be described independently of the other." This is of course true, but it is not directly clear what has been confirmed with 5 sigma (apart from the entanglement marker D whose relation to entanglement is not very clear from the presentation given in the paper; it has been mainly worked out in Ref. 18).

For the necessary theory background, the authors mainly refer to Ref. 18 and 19. Ref. 19 tries to demonstrate a violation of the Clauser-Horner-Shimony-Holt inequality (as far as I understand), which is identified in ref. 19 "as a particular useful form of the Bell inequality for 2x2 systems." In this sense the presented result would imply the possibility to discard a local hidden variable scenario. Is this the case also for the ATLAS result? Following ref. 18, entanglement is shown in this reaction following the Peres-Horodecki criterion. It seems to me that this is actually what is being measured by the ATLAS collaboration (while the criterion is not cited in the text). As far as I understand the criterion merely demonstrates that entanglement is present (assuming that a quantum mechanical description applies) but not discard any local hidden variable scenario.

To make this point more clear: I think everybody agrees that physics at a collider is a highly non-trivial quantum system where states of particles are in general not factorized i.e. cannot be described independently from each other. In this sense it would be important to clarify what the authors mean exactly by "observing entanglement" to make the relevance of their work more apparent. I understand that the authors would not like to spell out the entire derivation which has been presented elsewhere. They should however explain better the logic behind the entanglement marker D and what exactly it demonstrates.

page 4, 3rd paragraph, second equation: the authors state that "vectors B_{\pm} are analogous to those that appear in the general form of ρ ". Ref. 18 states that they are identical up to an overall normalization factor. If this is the case this should be also stated like that, since it implies that the expression allows to determine the elements of a density matrix (up to normalization) while "analogous" merely states that they play a similar role, but are not necessarily connected.

Related, on a more technical level: I understand that the spin density matrix ρ (as given in the first equation) is a standard parameterization for the two qubit density matrix. I understand that the expression clarifies the relation between density matrix and normalized differential cross-section (in particular if the relation between parameters is clarified as done in Ref. 18), which is then finally used to determine D , which is related to the $\text{tr}[C]$ i.e. certain parameters of the density matrix. It remains however completely obscure why this demonstrates entanglement. The expression given for ρ is merely a parameterization with the correct normalization, so it doesn't clarify this.

The authors state, spin correlation have been observed previously by ATLAS and CMS collaboration, their ref. 22-26. What distinguishes the current measurement from previous ones and what makes it special? Why does the observation of spin correlation not imply observation of entanglement?

Some minor details:

page 4, 4th paragraph: "entanglement arises only at threshold ..." Following the cited literature, it is true that the spin singlet (which is clearly the object of interest) is only observable at threshold, but the statement that there is only entanglement at threshold is misleading. It would imply that the $t\bar{t}$ state is separable away from threshold; I don't see any indication which would justify such a statement. This should be clarified

References: appropriate credit to previous work has been given as far as I can tell.

Clarity and context: lucidity of abstract/summary, appropriateness of abstract, introduction and conclusions.

Abstract, introduction and conclusion are appropriate, but I would like the authors to consider the comments made under "Suggested improvements"

Appropriate use of statistics and treatment of uncertainties: I have in general no doubt on appropriate use of statistics and treatment of uncertainties. I have however a few minor doubts on which I would like the authors to comment on:

The measured and expected result for D in the entanglement region are not consistent within errors. Do the authors attribute this to an incorrect determination by Powheg+Pythia (since the Monte Carlo it does not contain the complete spin information), the use of LO QCD matrix elements or do they have any other possible explanation for this (if not this should be also stated).

page 12: I am a bit surprised that the decay of top quarks, including their spin correlations, were modeled at LO precision in QCD only. The authors comment in footnote 3 that $bb4l$ cannot be directly compared to data since it is not possible to remove its offshell component in a formally correct way. I am not sure what this really means, but I understand that the differences are not substantial. It would be however useful to provide an estimate of uncertainties due to using leading order matrix elements only i.e. what does "not significantly change the conclusion of the measurement" imply. Is it a 20%, a 10% or a 1% correction etc.?

Author Rebuttals to Initial Comments:

Reviewer 1

We thank the reviewer for the constructive feedback. Below, we address the points raised by the reviewer.

1) The modelling of the threshold region needs additional studies:

Details of the bound state effects study:

In the following we describe a test on the bound state effects that is relevant to many of our responses to the reviewers questions. It is placed here before the responses themselves for clarity.

- The effects of bound state effects were evaluated already by performing essentially the same test as is proposed by the reviewer. We introduced a reweighting at the parton level based on the results in arXiv:0812.0919. Specifically, 2, where the cross-section is divided into various spin and colour states. We tried the following independent tests: We reweighted the truth $m_{T\bar{T}}$ distribution in the MC to show the increase in cross-section at ~ 342 GeV in the spin singlet gluon-gluon curve (we essentially imposed the ‘bump’ on the MC). Secondly, we reweighted the entire cross-section in a 5 GeV region around the bump to match the increase due to the bound state effects (essentially, a flat increase in cross-section in a 5 GeV bin centered on 342 GeV of $\sim 20\%$). This reweighted MC was then treated as an alternative signal sample and used to create a ‘bound state effect’ uncertainty in the calibration curve, in exactly the same way that other signal modeling systematics were used. In both of these tests, the largest systematic uncertainty on the calibration curve was 0.5%. This is smaller than the variation due to top mass (0.7%) and due to the lineshape uncertainty from MadSpin (1.6%), two uncertainties that are from different sources but which affect the underlying $m_{T\bar{T}}$ shape in a similar way (within our experimental resolution). The justification for our statement that this effect is covered by other uncertainties is simply that the quadrature sum of 0.5, 0.7, and 1.6 is essentially the same as 0.7 and 1.6. When considering the sum of all uncertainties, the addition of bound state effects doesn’t change the data result. The reason we have not included it, despite it not affecting the final sensitivity, is because the way we have introduced the bound state effect signal in these tests is not formally correct. The correct way to do this would be to only reweight the colour singlet gluon-gluon fusion events to show the same ‘bump’ (as these have the same $1s$ spin state as the bound state) or to have an alternative MC that includes the bound state effects. No such MC exists at present nor is it possible to isolate the colour-singlet spin-singlet gluon-gluon fusion events from the others in order to reweight them. At best, we can reweight all of the gluon-gluon events together, but this incorrectly changes the colour octet events as well as the colour singlet ones. Furthermore, isolating the gluon-gluon events in this way in the NLO MCs that we use for the measurements is not theoretically safe due to the

presence of higher order diagrams. Therefore, since we are confident that, even with a conservative reweighting of ALL of the types of events would not affect the overall uncertainty on the final result, and since we would not try to defend an ill-defined uncertainty to theory colleagues, we chose not to add it as a systematic uncertainty and this remains our strong preference.

Furthermore, the effect of including the bound state effects in the parton-particle level calibration curve was also tested and was found to essentially be negligible (a 0.02% effect) so we do not include it as an uncertainty on the entanglement limit.

This measurement focuses on the threshold region to enhance the sensitivity. In turn, effects that might not play a significant role in the bulk of the events used in a typical top-quark pair analysis can be significantly enhanced here and need to be accounted for.

On page 9, you clearly state yourselves that non-relativistic effects can lead to (pseudo) bound state effects close to threshold. These bound state effects increase the entanglement significantly, not only by changing the $\cos(\phi)$ distribution, but also by enhancing the contribution of the region most sensitive to entanglement in the signal model. The argument made here - that there are other effects that also change the $\cos(\phi)$ distribution and therefore the uncertainty is covered - is in itself invalid. All systematic uncertainties that have an effect on D change the $\cos(\phi)$ distribution (otherwise they would not have an effect); therefore the individual sources need to be accounted for. While many other systematic uncertainties are described in detail and quantitatively, this important aspect is only mentioned shortly, without giving the reader a quantitative estimate.

- The statement here is referring to the particular way in which bound-state effects enter the measurement, which is to change the lineshape, and was not intended to be a general statement about how all systematics change D . The experimental resolution on $m_{t\bar{t}}$ in this result is ~ 40 GeV, therefore, the impact of non-relativistic effects (from the point of view of the measurement) is to increase the cross-section at low $m_{t\bar{t}}$. From our studies (described in the following responses), the impact of non-relativistic effects on the measurement is 0.5%. Two other uncertainties also change the lineshape in this way, and to a greater extent. The top quark decay uncertainty (which is the largest uncertainty at 1.6%) shifts the line-shape higher at low $m_{t\bar{t}}$ and lower at high $m_{t\bar{t}}$ (and is then symmetrised) as does the top quark mass uncertainty (which is a 0.7% effect). To improve the clarity, we have changed the paragraph in the discussion to read: "It is important to note that close to the threshold, non-relativistic QCD processes, such as Coulomb bound state effects, affect the production of $t\bar{t}$ events~\cite{Kiyoy:2008bv} and are not accounted for in the MC generators. The main impact of these effects is to change the line-shape of the $m_{t\bar{t}}$ spectrum. The impact of these missing effects was tested by introducing them with an ad-hoc reweighting of the MC based on theoretical predictions and the effect was found to be 0.5%. Other

systematic uncertainties on the top quark decay (1.6%) and top quark mass (0.7%) also change the lineshape in a similar way within our experimental resolution and have a much larger impact. Therefore, the ad-hoc reweighting is not included by default in the measurement since including it would not change the sensitivity of the result within the precision quoted.”

N.B: In this context, it should not be forgotten that also electroweak effects contribute to the threshold region kinematics. They are strong enough to allow constraints on the top-quark Yukawa coupling in other analyses, but don't seem to be discussed in this paper explicitly, nor are they explicitly accounted for.

- This is a very good point raised by the referee. In ‘Methods’, under ‘Signal modeling uncertainties’, we have mentioned previously the following: “\textbf{NNLO reweighting}: The uncertainty due to missing higher-order corrections is estimated by reweighting the \sqrt{s} of the top quarks, the \sqrt{s} of the $t\bar{t}$ system, and the $m_{t\bar{t}}$ spectra at parton level to match the predicted NNLO differential cross-sections~\cite{Czakon:2015owf}.”
However, this reweighting actually also includes NLO EW effects. We now mention it specifically and have added the relevant citation: “\textbf{NNLO QCD + NLO EW reweighting}: The uncertainty due to missing higher-order corrections is estimated by reweighting the \sqrt{s} of the top quarks, the \sqrt{s} of the $t\bar{t}$ system, and the $m_{t\bar{t}}$ spectra at parton level to match the predicted NNLO QCD and NLO EW differential cross-sections~\cite{Czakon:2015owf,Czakon:2017wor}.”

The bound-state effects can change the conclusion of the paper quite significantly: (a) they can affect the calibration curves themselves by enhancing particular regions of phase space with events and therefore lead to different acceptance and efficiency corrections. This is in particular noteworthy given the (poor) resolution in terms of $m_{t\bar{t}}$ and the very narrow window around the threshold.

- We believe that this point is sufficiently covered by the previous responses, but we take the opportunity to reiterate that the bound state effects on the calibration curve do not impact the conclusions of the paper. To make this point even more clear: these effects can impact the prediction, but have a negligible effect on the uncertainty, compared to other systematic uncertainties.

And (b), the bound-state effects will change the translation of the entanglement limit from parton to particle level, by giving rise to a potentially significantly lower D limit on particle level - potentially affecting the significance and therefore changing the main result of the paper.

- The effect of including the bound state effects on the translation of the entanglement limit from parton to particle level was tested using the same reweighting used to test the effect on the reco-particle calibration curve. The difference in the result when including the bound state effects compared to not including them was 0.02% so it does not affect the significance.

Therefore, I consider it as crucial to further study and quantify (upper limits on) these effects before the paper can be accepted. Even though the picture is still incomplete from a theory perspective, the results of the following references could be used to estimate limits on the potential effects: e.g. arXiv:1007.0075, arXiv:2004.03088, arXiv:2102.11281.

- We agree that the theory picture is still incomplete and we hope that the publication of this paper will encourage work to be performed in this area. We have added to the paper the studies that have been performed on these topics and we have explained, we hope, the limitations of these assumptions. We do not believe, based on our studies, that these limitations significantly impact uncertainty and therefore on the observation of entanglement.

The submitted paper already includes a thorough study of the different parton shower models that are much less directly related to entanglement, and I would request a similar study on these threshold effects that are much more directly related to the final result.

- We hope that the previous responses go some way to addressing this very reasonable observation. However, we do think it is important to highlight that the theoretical understanding and the technical implementation of the parton shower is orders of magnitude more advanced than this relatively unexplored area of threshold effects w.r.t quantum entanglement. The parton shower study was performed over the course of a year and with the assistance of the generator authors themselves in order to understand the effect. No such study would be possible (at present) for the threshold effects that aren't yet included in the MC simulations at all. Whilst we don't think that this significantly limits the simple conclusion of observing entanglement, we hope that this work will encourage development of more sophisticated tools that will surely be necessary in order to take the field of Quantum Information at hadron colliders further and we feel that requiring the same level of sophistication in these studies would not only be impossible at present, but would deter the very work needed to address them in the future.

2) The calibration of D should be better explained or revisited.

Here, you describe how the different levels - parton, particle, and detector level - are defined and can be related to each other via calibration curves. However, it still becomes confusing at times.

Since this is a central part of the paper, I would suggest to define very clearly what all levels are in a central place, even though it may seem obvious, e.g. that $D_X = \dots$ (or similar). The purpose is to write more clearly that e.g. D on reconstruction level “uses the $\cos(\phi)$ distribution on reconstruction level”. In particular for readers outside of high energy physics, this would improve the readability. Then, I would suggest to consistently use D_X wherever applicable. This may also apply to the abstract. While “particle level” is written in the text there, using a subscript could help to avoid people glossing over it and interpreting the “ $D=\dots$ ” as parton-level D.

- We have added the following sentences to improve readability as suggested:
“The distribution of $\cos\phi$ in the signal region and the detector-level D_{detector} value, built from the $\cos\phi$ at the reconstructed detector-level and after background subtraction, are shown in the left and right panels of Figure~\ref{fig:cosphi}, respectively.” which matches the nomenclature used in the figures for the reco level D. Whilst we agree that a central area where these three definitions are alliterated would be nice, we had comments from the editor to significantly remove the word count and couldn’t find a way to include this whilst also reducing word count overall. Hopefully, the reco and truth level descriptions are close enough together in the text that these new changes sufficiently resolve the referees comment.

What I don’t fully understand in the context of calibration is the distinction made between the different parton shower models. The differences are considered a “real” uncertainty when going from reconstruction level to particle level. However, then their role is diminished to being alternative models to compare the particle level D value to. Wouldn’t it be more consistent to treat them as one or the other throughout the extraction (irrespective of the question if indeed such a 2-point comparison represents a well-defined uncertainty)? If the authors do not want to consider the differences a full uncertainty, one could e.g. give different results for the different models; or give one result where these effects are treated consistently as uncertainties. At the moment, and from the outside, the justification for the current scheme is not entirely clear.

- We agree that the difference between Pythia and Herwig is treated somewhat differently. On the measurement, correcting from reco level to data, it is fully symmetrised as it covers many uncertainties related to the parton shower (such as the hadronisation modeling), which is standard for all ATLAS top results. However, for the prediction (and the error on the prediction) we identified the source of difference between the shower models to be the ordering of the shower (which would not be reasonable to symmetrise) and that these other effects (such as hadronisation) do not play a significant role here. Thus, we think it is better to show both predictions. It amounts to the same as what the reviewer suggests as all the information is available for the reader (if they wished, they

could subtract the parton shower uncertainty in Table 2 from the total uncertainty on the result and compare to either prediction, though since the parton shower uncertainty is very small it doesn't result in any different conclusions from how the paper currently presents the result).

Another aspect of the calibration curves is that there are correlations between uncertainties when performing the calibration. Some uncertainties will appear when going from reconstruction to particle level, and similar ones when going from particle level to parton level. This discussion is currently missing from the paper, but also affects the final significance. I am assuming the authors have already quantified the level of correlation and I think it should be quoted in the paper text.

- The correlations between the uncertainties that appear in both calibrations were assessed and are almost entirely weakly correlated (with no statistically significant sources of anti-correlation). Therefore, the current procedure of treating them as uncorrelated results in the most conservative estimate of significance (which is well above 5 sigma). We also performed the test where we assumed maximal anti-correlation and even in that extreme case, the result remained significantly above 5 sigma in significance.

For the parametrisation described in A.3, would it be possible to visualise or quantify how well this parametrisation works, or justify the choice of polynomial more such that the reader can understand what level of possible approximation can be expected from this parametrisation?

- The parametrization described in A.3 was found to describe the parameter very well. We have added a sentence in A.3 to make it clear: "This parametrization was found to describe well the value of $D_{\Omega}(m_{\bar{t}})$, in good agreement with the MC prediction." Furthermore, we have added here one figure which shows the parametrization and the actual values from the MC, so the reviewer can see the strong agreement.

Also, a discussion of the limitations of this parameterisation in only two dimensions would be useful. The detector and reconstruction resolution is insufficient to neglect detector smearing effects, that are meant to be accounted for fully by the calibration curves for D. However, the detector smearing could also lead to a situation where variables other than only D and m_{tt} may play a role for the total effect of reduced or enhanced entanglement on the reconstruction level or particle level distributions. This could be part of the reason we see such large differences between the parton shower models.

- We believe that the limitations of the reweighting are already apparent from the methods section (as evidenced by the reviewer commenting on it). The reweighting is clearly described as being a function of only two variables and it is not possible to do anything other than this to change entanglement with currently available MC tools. We note, however, that this reweighting is not the cause of the difference in the parton showers which is already observed before any reweighting (as shown in Figure 4). The difference between the parton shower generators was observed at particle level, before detector effects can impact the observable.

While the statement that “The degree of entanglement is intrinsic in the calculations of the MC event generators and cannot be changed” is not wrong when applied to the calculations that do/have to incorporate entanglement, it is technically possible to break

the entanglement by hand in the generator. I suggest to use such an approach as cross-check against the reweighting method employed in this paper, which could also shed more light on the parton shower discussion.

- We have considered this comment carefully and we don't believe that what the referee suggests is possible. There is no way to identify the spin structure of the event based on MC truth histories, the spin correlation matrix isn't recorded or passed to the parton shower, so we don't see how what the referee suggests can be done and note that the referee hasn't explained how to achieve this with some existing literature as they have in other comments, so we assume it is a technique that is not publicly available to replicate. We can turn off spin correlations entirely in the generators, which would disable entanglement as the reviewer suggests, but would also modify the predictions for lepton observables so the test would not be particularly informative. It would not be possible to isolate if any differences that are seen are due to a change in entanglement or just to breaking assumptions in the top quark decay and it wouldn't produce a sample with $D=-\frac{1}{3}$ but $D=0$. We believe that studies such as the one the reviewer suggests are necessary in the future, but need to be performed by the MC experts (after they have developed the tools to do them). For this measurement, the difference in the parton shower behavior, whilst intriguing, does not limit the claim of observation.

In the introduction, it would be useful to acknowledge that in the context of spin-correlation measurements, the D parameter has been measured before.

- We have added a sentence: "It should be noted that the CMS collaboration has already measured $D = -0.237 \pm 0.011$ inclusively~\cite{CMS-TOP-18-006}, showing no signal of entanglement."

In the description of the systematic uncertainties (A.5.1), some uncertainties are well explained including the reason why they are considered uncertainties, while others lack explanation. In particular, the top-quark decay uncertainty, which is leading, could use a more in-depth explanation why the difference between Powheg and Madspin is considered an uncertainty - the same is true for the parton shower uncertainty (but here the authors could simply refer to section A.6). In general, it would help in particular the non-expert reader to understand the rationale behind choosing these variations. This is explained to some extent e.g. for the recoil uncertainty, but also here a better justification why both choices could represent reality (or are edge-cases with the true value likely in-between) would be very helpful.

- We have added a citation to the PUB note that describes these uncertainties to the first part of A.5.1 and expanded the description of the madspin uncertainty.

I would suggest to weaken the statement “The degree of entanglement is intrinsic in the calculations of the MC event generators and cannot be changed” for the reasons outlined above.

- We have removed the last part of the sentence, changing it to be: “The degree of entanglement is intrinsic in the calculations of the MC event generators.”

Reviewer 2

We thank the reviewer for the constructive feedback. Below, we address the points raised by the reviewer.

1) In the analysis, it is crucial to reconstruct each top (antitop) rest frame. However, the recipe for this reconstruction is not well described. The authors should give the following information.

[1-1] First of all, the main problem is the following. There are 6 unknown neutrino momentum components. Ignoring the W and top widths, they are constrained by the 2 W-boson mass-shell constraints, 2 top (antitop) mass-shell constraints and 2 momentum imbalance constraints (corresponding to the two transverse directions). Since the mass-shell constraints are quadratic equations for the neutrino momenta, generally, there are two solutions for neutrino momenta. However, due to imperfect detector resolution, there are cases where the measured event provides complex solutions (rather than real). I understand this is the essence of the ellipse method. A gist of the ellipse method should be provided.

- The referee described well how the Ellipse method works. We elaborate on the Ellipse method under 'Methods'. We don't think that adding more info in the paper body is relevant here since it is described in 'Methods' and in more detail in the reference we cite. Furthermore, we were asked to reduce the paper length, and we don't feel that adding more text on the Ellipse method is crucial.

Which solution is taken if there are two real solutions?

- We always take the solution with the lowest value of M_{tt} , in order to populate the region which is close to the production threshold. We now mention it specifically in 'Methods', specifically in the second paragraph of A.1.

[1-2] If the ellipse method does not give real solutions, the neutrino weighting method is used. A brief description of this method should be provided. In ref. [63], I have an impression that this method relies on the probability density of the lepton energy predicted by the Standard Model. My concern is that if the analysis uses Standard Model (SM) information, reconstructed distributions will be biased such that they show entanglement because the entanglement is present in the SM. The authors should justify their method in this regard.

- The NW method scans neutrino pseudo-rapidity distributions and can either do so completely agnostically (e.g. scanning between -5 and +5 in equal steps) but this is very inefficient and so we instead scan a very broad gaussian whose mean is determined based on a weak correlation with the charged lepton's pseudo-rapidity. This is indeed a SM assumption but it is not one that introduces a significant bias here because the difference in eta between the charged lepton and neutrino isn't a very spin-sensitive observable (and thus, is not sensitive to spin entanglement) and because the neutrino only carries low spin analyzing power (which dilutes the sensitivity to spin even further). The NW method is only used in a small fraction of events (~5%) but in Eur. Phys. J. C 80 (2020) 754 it was used in all events and the performance was explicitly checked between samples that include and exclude spin correlation and was found to be the same.

To make it more clear, we have added under methods the following: "In this analysis, the Neutrino Weighting method is only used in a small fraction of events ($\sim 5\%$). Furthermore, in Ref.~\cite{TOPQ-2016-10} it was used in all events and the performance was found to be the same between samples that include and exclude spin correlation."

[1-3] On page 11, the authors write, "If both methods fail, a simple pairing of each lepton with its closest b-tagged jet is used." This is about the combinatorics and not neutrino momentum reconstruction. How are the neutrino momenta reconstructed if both methods fail?

- If both methods fail we don't attempt to reconstruct the neutrinos at all and treat the $b\bar{b}$ system as " $t\bar{t}$ " and the $l\bar{b}$ pairs as top/anti-tops. We have added the following text to make this clearer "If both methods fail, a simple pairing of each lepton with its closest b -tagged jet are used as proxies for the top and anti-top quark and no attempt is made to reconstruct the neutrinos".

[2-1] Several parton-shower algorithms are compared in Fig.4. Based on this, the authors concluded that the main cause of the MC generator dependence is the parton-shower algorithms, rather than the hadronization models. To conclude this, the same hadronization models must be used among the distributions shown in Fig.4. Which hadronization models are used?

- We referred to this in the 3rd paragraph of section A.6. In principle Herwig also offers the possibility to use a string model now, so we made clear in the text what we use for the study: "While \PYTHIA is based on the Lund string model and uses a \pt-ordered dipole shower~\cite{Gustafson:1994cd,Lonnblad:1995yk,Friberg:1996xc}, the \HERWIG samples used in this study are based on a cluster model and uses an angular-ordered shower as the default~\cite{Webber:1997iw}."

[2-2] If different hadronization models are used in Fig.4, I have to rely on a statement on page 19, “A comparison between MC simulations with different hadronization models has shown that these have a negligible effect on the $\cos(\phi)$ distribution”. This sentence should be elaborated. Which MC generators (parton-shower algorithms) are used for this comparison? How small was the effect?

- We have rephrased this part to make more clear what we did exactly: “A comparison between MC simulations with different hadronization models was performed. For one study, \SHERPA was used with either a string or a cluster model for hadronization. For the other study \HERWIG[7] was used, again comparing the effects of using either a string or a cluster model. Changing the hadronization model has shown in both cases to have a negligible effect on the $\cos\phi$ distribution, both when not placing a cut on $m_{\bar{t}t}$ and when using a smaller part of phase space close to the signal region of the analysis, with $m_{\bar{t}t} < 380 \text{ GeV}$.”

3) On page 19, the authors write, “parton-level measurement would therefore suffer from the ambiguity in $\cos(\phi)$, while the particle-level measurement presented in this paper does not.”

On page 20, “These findings lead to the conclusion that performing the measurement at particle level is more attractive, since the overall uncertainties are smaller”.

I do not understand these sentences. In the particle level, measured D has a smaller error, but the entanglement threshold has a larger error. In the parton level, measured D has a larger error, while the entanglement threshold has no error ($= -1/3$). It seems to me that they are equally powerful (less powerful) in terms of the entanglement measurement. The authors should clarify this point.

- The choice of correcting to particle or parton level results in either taking the full difference between the two shower predictions as a systematic uncertainty, or isolating the difference as separate predictions, so the two correction methods are not equally powerful (one contains an uncertainty that the other does not). We have changed the text from:

“These findings lead to the conclusion that performing the measurement at particle level is more attractive, since the overall uncertainties are smaller”

to

“These findings lead to the conclusion that performing the measurement at particle level is more attractive, since the difference in the predictions when extrapolating from parton to particle level can be isolated and not taken as full systematic uncertainty.”

4) This is a minor point. On page 11, “they must then lie within $\Delta R > 0.4$ from a jet to avoid being removed from the event.”

Probably, the authors want to mean that “they must not lie within $\Delta R < 0.4$ ” (?)

In the detector-level analysis, I do not find the corresponding lepton removal procedure. Why is this procedure implemented only in the parton-level analysis?

- This cut is only applied at particle level and mimics a more elaborate procedure that is applied at reco level that we had neglected to describe in the text. To clarify this, we’ve added the following text, where we define the reconstructed objects:

“Objects can fulfill the criteria for both jet and lepton selection, necessitating the implementation of an overlap removal procedure. This way, objects are associated with a singular hypothesis. First, any electron candidates that share a track with a muon candidate are removed. Subsequently, jets within $\Delta R = 0.2$ of an electron are removed, and afterwards, electrons within a region $0.2 < \Delta R < 0.4$ around any remaining jet are rejected. Jets that have fewer than three tracks and are within $\Delta R = 0.2$ of a muon candidate are removed, and muons within $\Delta R = 0.4$ of any remaining jet are discarded.”

and rephrased the sentence where we defined the particle level objects to read:

“Electrons and muons must also be well separated from jet activity. If they lie within $\Delta R < 0.4$ from a jet they are removed from the event.”

Reviewer 3

We thank the reviewer for the constructive feedback. Below, we address the points raised by the reviewer.

Data & methodology: validity of approach, quality of data, quality of presentation:

The authors mention that they use LO QCD matrix elements in their description. Also Ref. 18 (on which much/all of the theory part is based on) uses LO QCD matrix for the discussion. The authors should clarify whether the condition $D < -1/3$ is valid within a leading order QCD approximation only or whether it is a non-perturbative/all order statement i.e. a generic feature. If the statement is only valid within leading order QCD perturbation theory, they should provide (if possible) an estimate of uncertainties due to possible higher order corrections or (if not possible) acknowledge this in their article. The latter case would of course diminish the relevance of the observed result (at least if no estimate of higher order corrections can be provided)

- Indeed the criterion $D < -1/3$ is universal, and is independent from the order of the calculation. This is now mentioned in the text when introducing this criterion: “The existence of an entangled state is demonstrated if the measurement satisfies $D < -1/3$, derived from the Peres–Horodecki criterion~\cite{Peres:1996dw,Horodecki:1997vt}, and is independent from the order of the calculation.”
Furthermore, we note that for the production of top-antitop pairs, we use NLO estimation from Monte Carlo simulations and assign an uncertainty due to higher-order corrections, in particular at NNLO QCD + NLO EW (see methods).

I understand that the result presented by the authors should be understood as a first demonstration that quantum entanglement can be studied in the environment of a high energy collider. While I agree that observation of spin entanglement at LHC is already a noteworthy achievement, I wonder whether the result has deeper implication i.e. which future explorations might be possible in a collider environment which cannot be achieved in conventional laboratory experiments (e.g. setups in which quantum information experiments are being carried out usually). This is partly addressed in Ref. [17], but the importance of their result would become more apparent if they could also point that out in the draft (if this is the case).

- We agree and this is a topic of a great deal of debate in many papers published in recent months. Since the discussion and implications of this measurement and other proposed measurements is a rapidly evolving discussion, we believe that the discussion in the final paragraph is as far as is reasonable to go. Additionally, we are conscious of length

limitations of the paper which practically also limits the depth of discussion that is possible.

I miss in the introduction an explanation why the LHC is a particularly useful environment for the study of entanglement of pairs of quarks? Naively one might assume that an electron positron collider would be more suitable since it allows a better control of the initial state. The only possible answer I can find is that LHC allows for the production of top antitop pairs due to its high center of mass energy, while the top does not hadronize and therefore constitutes the only particle that allows for observation of those effects. While all this is somehow mentioned, I would recommend to highlight these points or supplement them if appropriate.

- The LHC is the only experiment which we have that allows us to produce top quark pairs and measure the top quark properties with good precision. We have added a sentence in the introduction to make it clear: "The experiments at the LHC ring, such as ATLAS, are the only ones currently taking data which is able to produce and study the properties of the top quark."

On page 3, 3rd paragraph, the authors state that "Entanglement is observed with a significance of more than five standard deviations for the first time in pairs of quarks." Entanglement is however only defined as "If two particles are entangled, the quantum state of one particle cannot be described independently of the other." This is of course true, but it is not directly clear what has been confirmed with 5 sigma (apart from the entanglement marker D whose relation to entanglement is not very clear from the presentation given in the paper; it has been mainly worked out in Ref. 18). For the necessary theory background, the authors mainly refer to Ref. 18 and 19. Ref. 19 tries to demonstrate a violation of the Clauser-Horner-Shimony-Holt inequality (as far as I understand), which is identified in ref. 19 "as a particular useful form of the Bell inequality for 2x2 systems." In this sense the presented result would imply the possibility to discard a local hidden variable scenario. Is this the case also for the ATLAS result? Following ref. 18, entanglement is shown in this reaction following the Peres-Horodecki criterion. It seems to me that this is actually what is being measured by the ATLAS collaboration (while the criterion is not cited in the text). As far as I understand the criterion merely demonstrates that entanglement is present (assuming that a quantum mechanical description applies) but not discard any local hidden variable scenario.

- Indeed what we measure is derived from the Peres-Horodecki criterion, and we forgot to cite it. We thank the referee for pointing that out, as clearly it is important to mention it in the paper. We now mention it specifically in the text and cite the relevant paper: "The existence of an entangled state is demonstrated if the measurement satisfies $S_D < -1/3$, derived from the Peres-Horodecki criterion~\cite{Peres:1996dw,Horodecki:1997vt}, and is independent from the order of the calculation."

We note that this result is only capable of showing that top spin correlation cannot be described by classical probabilities and instead requires QM. The stronger statement on local hidden variables would need to be tested with a Bell type operator, and the result presented in this paper cannot formally make a statement on this.

To make this point more clear: I think everybody agrees that physics at a collider is a highly non-trivial quantum system where states of particles are in general not factorized i.e. cannot be described independently from each other. In this sense it would be important to clarify what the authors mean exactly by "observing entanglement" to make the relevance of their work more apparent. I understand that the authors would not like to spell out the entire derivation which has been presented elsewhere. They should however explain better the logic behind the entanglement marker D and what exactly it demonstrates.

- We note that we have already partly addressed this point above, by referring to the Peres–Horodecki criterion. To make it even more appealing, we have added that measuring D to be smaller than $-\frac{1}{3}$ violates a Cauchy–Schwarz inequality, which is a well known criterion from other physics systems, which in addition for spins is well-known from condensed-matter setups: “It can be understood as a violation of a Cauchy–Schwarz inequality, a notable entanglement criterion in fields such as quantum optics, condensed matter or analogue gravity~\cite{Walls2008,Wolk2014,deNova:2012hm}.”

page 4, 3rd paragraph, second equation: the authors state that "vectors B_{\pm} are analogous to those that appear in the general form of ρ ". Ref. 18 states that they are identical up to an overall normalization factor. If this is the case this should be also stated like that, since it implies that the expression allows to determine the elements of a density matrix (up to normalization) while "analogous" merely states that they play a similar role, but are not necessarily connected.

- These terms are the same as those that appear in the general form of ρ . We made it clear in the text: “These terms are the same as those that appear in the general form for ρ .”

Related, on a more technical level: I understand that the spin density matrix ρ (as given in the first equation) is a standard parameterization for the two qubit density matrix. I understand that the expression clarifies the relation between density matrix and normalized differential cross-section (in particular if the relation between parameters is clarified as done in Ref. 18), which is then finally used to determine D, which is related to the $\text{tr}[C]$ i.e. certain parameters of the density matrix. It remains however completely obscure why this demonstrates entanglement. The expression given for ρ is merely a parameterization with the correct normalization, so it doesn't clarify this.

- This criterion is a derivation of the Peres–Horodecki criterion. We have now made it clear in the text and cited the relevant references: “The existence of an entangled state is demonstrated if the measurement satisfies $D < -1/3$, derived from the Peres–Horodecki criterion~\cite{Peres:1996dw,Horodecki:1997vt}, and is independent from the order of the calculation.”
- To make it even more appealing, we have added that measuring D to be smaller than $-1/3$ violates a Cauchy–Schwarz inequality, which is a well known criterion from other physics systems, which in addition for spins is well-known from condensed-matter setups: “It can be understood as a violation of a Cauchy–Schwarz inequality, a notable entanglement criterion in fields such as quantum optics, condensed matter or analogue gravity~\cite{Walls2008,Wolk2014,deNova:2012hm}.”

The authors state, spin correlation have been observed previously by ATLAS and CMS collaboration, their ref. 22-26. What distinguishes the current measurement from previous ones and what makes it special? Why does the observation of spin correlation not imply observation of entanglement?

- It is possible to have correlated spins that can be explained by classical probabilities, however, once these correlations become strong enough, they require QM to explain them. In all of the previous measurements from ATLAS and CMS, the observed correlations were not strong enough to claim that they couldn't be explained without the presence of QM. What differs in this measurement is that we have isolated a region of phase space (using \bar{m}_T cuts) where the spin correlations are so strong that there is no way to account for them without QM (via the quintessential property of QM, entanglement).

page 4, 4th paragraph: "entanglement arises only at threshold ..." Following the cited literature, it is true that the spin singlet (which is clearly the object of interest) is only observable at threshold, but the statement that there is only entanglement at threshold is misleading. It would imply that the the $\bar{t}\bar{t}$ state is separable away from threshold; I don't see any indication which would justify such a statement. This should be clarified

- We have changed this statement to mention that entanglement survives only close to threshold. The sentence now reads: “After averaging over all possible top-quark directions, entanglement only survives close to threshold because of the rotational invariance of the spin singlet.” We would like to point out that in the referenced literature cited by the paper, indeed it is shown that entanglement is present close to the production threshold and in other regions of phase space. However, the latter disappears after averaging over all possible top-quark directions, and we're left with entanglement only close to the production threshold.

Abstract, introduction and conclusion are appropriate, but I would like the authors to consider the comments made under "Suggested improvements"

- The Suggested improvements have been addressed above. We thank the referee for making these proposals.

The measured and expected result for D in the entanglement region are not consistent within errors. Do the authors attribute this to an incorrect determination by Powheg+Pytha (since the Monte Carlo it does not contain the complete spin information), the use of LO QCD matrix elements or do they have any other possible explanation for this (if not this should be also stated).

- A discussion of this may be found in the second paragraph of the discussion. In particular, we note that non-relativistic QCD effects, which are important close to the production threshold, affect the production of top-quark pairs. These effects are not modeled by the MC generators, and therefore cannot be added to the prediction. We note, however, that the methods used in this paper are solid in order to extract the result. It's merely the prediction which is affected by these effects, and since these effects enhance the contribution of a spin-0 state close to threshold, it is indeed expected to make the prediction for entanglement to be stronger. We do hope that this measurement will motivate the theory community to work on more reliable predictions, to be accounted in tools which can be used by the experimental community.

page 12: I am a bit surprised that the decay of top quarks, including their spin correlations, were modeled at LO precision in QCD only. The authors comment in footnote 3 that bb4l cannot be directly compared to data since it is not possible to remove its offshell component in a formally correct way. I am not sure what this really means, but I understand that the differences are not substantial. It would be however useful to provide an estimate of uncertainties due to using leading order matrix elements only i.e. what does "not significantly change the conclusion of the measurement" imply. Is it a 20%, a 10% or a 1% correction etc.?

- We direct the reviewer to <https://arxiv.org/pdf/1907.03729.pdf> Table 3 which shows the difference between an exact calculation at fixed order for spin correlation parameters (including D) vs. Monte Carlo predictions which include the described caveats. The effect is on the order of 5% percent. It is also not the case that this directly translates into an uncertainty on the data as the calibration curve is sensitive to changes in the decay modeling in the MC on the order of 2% (see the top quark decay uncertainty in Table 2 in the Methods section, which includes spin correlation as well as other decay effects, so 2% is probably a conservative upper bound on the effect).

Reviewer Reports on the First Revision:

Referee #1 (Remarks to the Author):

Dear authors,

Thank you for the comprehensive answers to my questions and comments. The text reads better now, in particular I was pleased to see the more clear distinction between parton, particle, and detector level despite the word limit, and the additional information on the electroweak effects. I would also like to thank the authors for providing the plot showing the parametrisation of D .

However, I am not fully satisfied with the answers on the threshold modelling, yet, and would therefore request another revision.

I am not convinced that the MC sample used in the analysis contains the relevant events (with any weight) when approaching the threshold, leading to large corrections as a_s/β becomes large, requiring resummation. As the authors write themselves, the composition at threshold, e.g. of octet/singlet contributions, matters. Effects that can affect not just the predicted entanglement (which the measurement should be insensitive to), but also the acceptance, are not limited to the line shape only. E.g. in the reference I provided [arXiv:2102.11281], Fig. 2 shows a significant effect on the kinematics of the system, and therefore possible effects on the acceptance corrections.

This is backed up by the recent result from CMS [cds 2893854], where the respective authors seem to attempt an inclusion of bound state effects, following the model in Ref. [arXiv:2102.11281]. While the CMS measurement and the ATLAS measurement follow a different approach, the theory modelling should have at least a comparable effect. However, CMS claims 10% uncertainty on the measured D_{parton} due to (possibly not even conservative enough) variations of the η_t contribution, while you find a sub-percent effect on D_{particle} . This uncertainty does not grow when going to D_{parton} with the reweighting method.

Of course small differences are normal, maybe even up to a factor of two. However, comparing the numbers at D_{parton} directly, the CMS uncertainty is a factor of 20 larger, while you claim that your reweighting should in principle cover the same, if not even more, effects. There is no guarantee that the preliminary CMS result is free of issues, but the size of this discrepancy does require further studies in my opinion, as a possibly dominant source of uncertainty may be vastly underestimated, and I am worried that this could - in the end - constitute a major weakness of the paper under review here.

As a minor note, I am not sure I understand the discussion of the correlations between the different levels of calibration. For a given (theory) uncertainty, its effect on both calibration steps is - by definition - fully correlated. So I am not sure I understand the answer, and the conclusion to treat both steps fully uncorrelated in this context. Could the authors please clarify?

Referee #2 (Remarks to the Author):

The authors' reply and the revised manuscript properly address all questions and points raised by reviewers 1 and 2. I now recommend the article for publication.

Referee #3 (Remarks to the Author):

I would like to thank the authors for their detailed answer to my questions and concerns. The current modifications to the paper address these concerns and I therefore believe that the paper is suitable for publication in its current form. I also studied the concerns raised by referee 1 and 2 and as well as the answers provided by the authors. To the best of my understanding, the authors address these questions to a satisfactory level.

Author Rebuttals to First Revision:

Referee #1 (Remarks to the Author):

We thank the reviewer for their time and careful review of our revised manuscript. Below, we have addressed the comments and concerns raised by the reviewer.

Dear authors,

Thank you for the comprehensive answers to my questions and comments. The text reads better now, in particular I was pleased to see the more clear distinction between parton, particle, and detector level despite the word limit, and the additional information on the electroweak effects. I would also like to thank the authors for providing the plot showing the parametrisation of D . However, I am not fully satisfied with the answers on the threshold modeling, yet, and would therefore request another revision.

I am not convinced that the MC sample used in the analysis contains the relevant events (with any weight) when approaching the threshold, leading to large corrections as a_s/β becomes large, requiring resummation. As the authors write themselves, the composition at threshold, e.g. of octet/singlet contributions, matters.

- We agree that the MC does not contain any bound state effects. However, what we are most interested in is the effect that bound state effects would have on the reco-particle level calibration curve as only this information would impact the actual measurement. For this purpose we believe that the events in the MC are sufficient for this task. From the point of view of detector effects, the toponium simply increases the fraction of spin and colour singlet events (or at least, this is the assumption underlying our test). Whilst we cannot isolate these specific events, in the region where bound state effects are expected, we can increase the total cross-section by some amount (and 90% of events in this region are the spin and colour singlet events that we want) either by matching the resonance structure in the theory literature, or the more simple “flat” reweighting described in our responses in the previous round of comments. To account for the 10% of events we didn’t want to reweight, we added additional variations above the expected amount of toponium. Even when using unphysical and extreme variations (e.g. the flat cross-section reweighting), which we would assume would have a much larger effect than toponium alone, it does not alter the significance of the result.

Effects that can affect not just the predicted entanglement (which the measurement should be insensitive to), but also the acceptance, are not limited to the line shape only. E.g. in the reference I provided [arXiv:2102.11281], Fig. 2 shows a significant effect on the kinematics of the system, and therefore possible effects on the acceptance corrections.

- The figures in the paper you refer to show the reconstructed top mass and rapidity difference of the tops, neither of which are used as selection cuts in the measurement.

The rapidity difference, in particular, is not correlated with any selection requirement that we use. We remind the reviewer that the selection criteria used to select the events are very loose. With the exception of $m_{T\bar{T}}$ (which our reweighting will approximate the effect of correctly by definition) we only use p_T and η cuts on selected objects, all of which are well covered by a multitude of other systematic uncertainties. We don't dispute that we cannot perfectly test the effect of toponium but we don't think that it is possible to do this at present and we don't believe it would significantly change the conclusions of this paper.

This is backed up by the recent result from CMS [cds 2893854], where the respective authors seem to attempt an inclusion of bound state effects, following the model in Ref. [arXiv:2102.11281].

- We contacted the authors of 2102.11281, and they informed us that the model is not public. There are also aspects of the model, such as the Green's function re-weighting, that involve tabulated data that isn't clear how to apply to the prediction, and again, is not shared by the authors. It appears that CMS have generated their own model to use, for which we don't have enough information to comment on how different it is from the prediction of 2102.11281, nor can we comment on its validity. As far as we know, from our interaction with a variety of authors, the theory community is currently working on improved models for toponium, but it is not clear when all of these will be public. We note that one of the goals of our work was actually to motivate the theory community to improve the modelling of top pair production near threshold, and we are happy that this is indeed the case. By necessity, this result had to come first to induce such developments. We hope the reviewer will agree, at least, that it is not possible for us to use this model as it is not publicly available.

While the CMS measurement and the ATLAS measurement follow a different approach, the theory modelling should have at least a comparable effect. However, CMS claims 10% uncertainty on the measured D_{parton} due to (possibly not even conservative enough) variations of the η_t contribution, while you find a sub-percent effect on D_{particle} . This uncertainty does not grow when going to D_{parton} with the reweighting method. Of course small differences are normal, maybe even up to a factor of two. However, comparing the numbers at D_{parton} directly, the CMS uncertainty is a factor of 20 larger, while you claim that your reweighting should in principle cover the same, if not even more, effects. There is no guarantee that the preliminary CMS result is free of issues, but the size of this discrepancy does require further studies in my opinion, as a possibly dominant source of uncertainty may be vastly underestimated, and I am worried that this could - in the end - constitute a major weakness of the paper under review here.

- We are uncomfortable that this comment requires us to comment on the recent CMS result (which was made public after we had submitted our previous responses). The CMS measurement is, as yet, preliminary, not peer reviewed, and we have no access to internal CMS information. Based on recent discussion in the LHC Top WG forum, the result is still preliminary and may change before final publication. We find it awkward to speculate upon the analysis methods of another experiment. Nevertheless, we understand that the question is an obvious one and we have attempted to respond to your comments as best as we are able.
- We believe that some (understandable) confusion has arisen here based on how CMS have chosen to quote their uncertainties, which are not uncertainties in the typical sense. What CMS quotes, in Table 3 of their conference note, is the relative change in the size of the total uncertainty of a profile likelihood fit if they fix the cross-section of their toponium model to the SM expectation and do not allow it to float in their fit. This is not the same as what would typically be called an uncertainty and this is self-evident from the fact that if they had a 10% uncertainty on their central value from bound state effects, as they quote in their table, they would not have sufficient sensitivity to claim observation from the entanglement limit.

What we quote as an uncertainty, and what we would claim is a more commonly understood definition, is the change in the value of D due to a source of systematic uncertainty (in this case, the inclusion of bound state effects in our signal model). This information is not tabulated in the CMS PAS note.

Fortunately, we can work out something similar to the size that their toponium uncertainty would be in our formalism as they have provided the central value change in Figure 8 when they do not include toponium in their model. Their bound state effect uncertainty would then be something like 2.2% in our definition (the relative difference in the two data points), compared to our 0.5%, though this is still not exactly an apples-to-apples comparison.

It is crucial to point out that the statement “the CMS uncertainty is a factor of 20 larger” comes from different definitions of “uncertainty” between the two results and does not correspond to the real difference in uncertainty/precision of the results.

- Furthermore, as the reviewer points out, the methods ATLAS and CMS have used are very different in methodology, data sets employed, and the definition of the signal region. In particular, it is well known that Profile Likelihood fits can be more sensitive to systematic uncertainties than detector-corrected results (e.g. those that use unfolding), which can be readily seen from CMS’s large JES uncertainty (defined as the relative change in the uncertainty, as mentioned above) which has a far larger effect on the total uncertainty than in our measurement. A-priori, ATLAS and CMS have similar JES performance and precision, so this nicely highlights the importance of optimal experimental methodology choices and the difference in what ATLAS and CMS each

quote as “uncertainties” in their tables.

- Finally, If we were to use CMS’s bound state effect uncertainty (or 2.2%) instead of our own, it would not reduce the significance of the measurement sufficiently to alter the conclusions of this paper. In fact, the precision of our measurement can accommodate an uncertainty 3 times larger than what CMS have (and 14 times larger than our own) before our significance would no longer be above 5 sigma.
- We believe that these points address the concern of the reviewer, which we reiterate, is an understandable one due to the confusion in what CMS defines as an uncertainty relative to our paper. The treatment of bound state effects does not significantly alter the statement of observation of entanglement. We further note that CMS are also claiming observation even with their larger toponium uncertainty and using a smaller set of the available run 2 data-set, corresponding to about 36 inverse fb.

As a minor note, I am not sure I understand the discussion of the correlations between the different levels of calibration. For a given (theory) uncertainty, its effect on both calibration steps is - by definition - fully correlated. So I am not sure I understand the answer, and the conclusion to treat both steps fully uncorrelated in this context. Could the authors please clarify?

- As an example to clarify what we mean, we use the parton shower uncertainty. **At parton level**, there is no difference between the Powheg + Pythia model and the Powheg + Herwig model. At the matrix element level, they are identical. When looking at the particle level, the D predictions for the Powheg + Pythia model don’t change much compared to the parton level, whereas the Powheg + Herwig model dilutes the spin correlation significantly. Ergo, the calibration curve for Powheg + Herwig has a shallower slope than Powheg + Pythia when going from parton to particle level and a systematic uncertainty here (were we to assign one) would be large. Going from particle to reco level, the detector dilutes the D by a factor of roughly 3. However, it does this almost identically for the Powheg + Herwig D values as it does for the Powheg + Pythia D values. Even though those underlying values are different, the resulting calibration curves for particle to reco level are nearly identical. The points themselves lie in different places along the curve, but the curve itself is the same. Thus, the systematic uncertainty is very small when going from particle to reco level.
- What we meant by saying that they are weakly correlated is that the change in the calibration when going from parton to particle level would be large, but the change in the calibration curve going from reco to particle level is very small. Equivalently, a large change in parton-particle level doesn’t necessarily lead to a large change in particle to reco level, nor does it even have to go in the same direction so we don’t treat these uncertainties as correlated when calculating the significance. We did explicitly check these correlations and found them to be weak and positive in most cases (within the MC

stat uncertainties). When performing the significance test we also tested keeping these correlations and the result was that the significance increases (very slightly) so we think it is more conservative and simpler to leave them as uncorrelated. Either way, it doesn't change the conclusion of the observation.

Referee #2 (Remarks to the Author):

The authors' reply and the revised manuscript properly address all questions and points raised by reviewers 1 and 2. I now recommend the article for publication.

- We thank the referee for their time and careful review of our revised manuscript.

Referee #3 (Remarks to the Author):

I would like to thank the authors for their detailed answer to my questions and concerns. The current modifications to the paper address these concerns and I therefore believe that the paper is suitable for publication in its current form. I also studied the concerns raised by referee 1 and 2 and as well as the answers provided by the authors. To the best of my understanding, the authors address these questions to a satisfactory level.

- We thank the referee for their time and careful review of our revised manuscript.

Reviewer Reports on the Second Revision:

Referee #1 (Remarks to the Author):

Dear authors,

Thank you for your replies to my previous comments.

However, I am afraid they are still not addressing the point in a satisfactory manner that I have made since the very first round of comments, and that I have been repeating ever since: Within the SM, quasi bound states exist (this is not new physics) and we know they affect the threshold region. The possibility to at least attempt to quantify these effects beyond line-shape reweighting exists, as the preliminary CMS result shows. Irrespective of the definition of the uncertainty table in the CMS result, it is their leading uncertainty. I do accept that this likely does not change your (rightful) claim of observation of entanglement. However, you also report a measurement of D . I could accept that the effects do not translate to the particle-level definition that you are employing. However, this requires - in my opinion - a quantitative simulation study that I have been explicitly asking for. Your current replies do not address this request in a manner satisfactory for me.

I can understand if it requires a non-negligible effort to pass a (rough) toponium sample through the detector simulation, which is what I assume the push-back is coming from. I can also understand that a constructive approach is needed at this point such that the review can conclude.

Therefore, I would be satisfied if you could show that its effect on the calibration from D_{particle} to D_{parton} is similar to the impact it has on the preliminary CMS result within a factor of about two (to give a quantitative threshold and avoid unnecessary iterations). This would at least hint that the effect between D_{reco} and D_{particle} is likely small (enough) to not significantly change the uncertainty on the measured D_{particle} .

In the following you can find more detailed comments to your responses.

“We agree that the MC does not contain any bound state effects ... it does not alter the significance of the result.”

-> I am not debating that the significance will remain above 5 sigma. Clearly and as you point out yourself later, even a larger uncertainty does not change the fact that you observe entanglement. However, you also quote a result on D , and that is my concern. For the preliminary result from CMS, toponium is the leading uncertainty. This alone - to me - justifies a quantification for your measurement - or to refrain from claiming a (precise) measurement.

“The figures in the paper you refer to ... we don't think that it is possible to do this at present and we don't believe it would significantly change the conclusions of this paper.”

-> The rapidity difference and top mass distribution are distributions the authors of that paper picked. Concluding that it is **only** these distributions that are affected and - since you are not using

these exact distributions in your analysis (or strongly correlated ones) - the impact must be negligible is not valid.

“We contacted the authors of 2102.11281 [...] By necessity, this result had to come first to induce such developments. [...] it is not possible for us to use this model as it is not publicly available.”

-> Indeed, a more detailed model of the effects would be very useful and it is great that your result is stimulating these developments. That paper, however, came out way before this article was submitted. Whether it is publicly available or not, the paper showed that there is an impact in exactly the difficult region you are using for your measurement. If there could be an effect, then the best possible attempt needs to be made to estimate it, and assign an uncertainty to it when measurements are performed in the affected region of phase space. The availability of code has - strictly speaking - no scientific relevance. In the meantime we learned that your attempt may underestimate the effect (admittedly not by a factor of 20 but possibly still by a factor of 4-5), and that such a simulation can be run.

[on the CMS uncertainties]. Thank you for the clarification. Indeed, that is a very suboptimal way to present the uncertainties. However, whether it is relative or not, it does remain the leading uncertainty and therefore there is still a strong motivation to also study it in context of your measurement.

“Profile Likelihood fits can be more sensitive to systematic uncertainties than detector-corrected results“

-> It is irrelevant for the review here, but this statement is not valid.

“Finally, If we were to use CMS’s bound state effect uncertainty ... [still] above 5 sigma“

-> As I already pointed out earlier, I am not questioning this. However, if I naively add the 2.2% to your total uncertainty on D, its uncertainty does change significantly; and that is my concern.

Regarding the correlations, thank you for the detailed explanation. Either way it was a minor comment and in addition I am fully satisfied with the answer.

To summarise, I am - personally - not fully satisfied with the answers, yet. I hope that my proposal to resolve my hesitation does imply a lot of additional work or delay, as also I would be happy to see this important result being published in a timely manner. As the other referees seem to be satisfied, I will also accept if the editor decides to overrule my input. I just personally think that such a clearly important result to appear in a high-quality journal actually deserves a high amount of scrutiny, also taking into account recent developments.

Best wishes.

Referee #3 (Remarks to the Author):

In the following I will only comment on the concerns raised by referee 1 and the replies provided by the authors of the submitted draft. I would further would like to stress that I am not an expert on toponium and can only give my personal impression on the discussion. I also do not possess any information on the mentioned CMS result.

It is my understanding that both referee 1 and the authors agree that there are open theory questions concerning the correct description of $t\bar{t}$ production in the threshold region, which is the region of interest for the presented draft. To my understanding the authors argue convincingly that these uncertainties do not affect the significance of their result.

I also agree with the authors that it is very difficult to comment on a preliminary result by another collaboration. I nevertheless find the arguments of the authors convincing. I therefore believe that the larger uncertainties reported by the CMS experiment should not be an obstacle for publication of this article.

Author Rebuttals to Second Revision:

Reviewer 1 comments

Dear reviewer,

Thank you once again for the time you have invested in reviewing our result. Please find below our in-line responses to your comments.

With best regards,

The Authors.

Dear authors,

Thank you for your replies to my previous comments.

However, I am afraid they are still not addressing the point in a satisfactory manner that I have made since the very first round of comments, and that I have been repeating ever since: **Within the SM, quasi bound states exist (this is not new physics) and we know they affect the threshold region. The possibility to at least attempt to quantify these effects beyond line-shape reweighting exists, as the preliminary CMS result shows. Irrespective of the definition of the uncertainty table in the CMS result, it is their leading uncertainty. I do accept that this likely does not change your (rightful) claim of observation of entanglement. However, you also report a measurement of D. I could accept that the effects do not translate to the particle-level definition that you are employing. However, this requires - in my opinion - a quantitative simulation study that I have been explicitly asking for. Your current replies do not address this request in a manner satisfactory for me.**

I can understand if it requires a non-negligible effort to pass a (rough) toponium sample through the detector simulation, which is what I assume the push-back is coming from. I can also understand that a constructive approach is needed at this point such that the review can conclude. Therefore, I would be satisfied if you could show that its effect on the calibration from D_{particle} to D_{parton} is similar to the impact it has on the preliminary CMS result within a factor of about two (to give a quantitative threshold and avoid unnecessary iterations). This would at least hint that the effect between D_{reco} and D_{particle} is likely small (enough) to not significantly change the uncertainty on the measured D_{particle} .

- We again reiterate that we don't believe that making comparisons to an unpublished CMS result, such as you suggest with *"I would be satisfied if you could show that its effect on the calibration from D_{particle} to D_{parton} is similar to the impact it has on the*

preliminary CMS result within a factor of about two” to be reasonable, given that we have almost no information about the implementation of the CMS topoponium model. Once again, however, we will attempt to respond to the reviewer’s points.

- Following the last round of comments, we successfully obtained a topoponium model. It is not the same model that CMS created but is the model that was used in the Maltoni et. al. JHEP 03 (2024) 099 paper. We created a new calibration curve from parton to particle level incorporating this model, a visual representation of which is attached to these responses. The topoponium was included using the cross-section assumptions from <https://arxiv.org/pdf/0812.0919> (the same result used in our topoponium reweighting tests). The result was that the topoponium inclusion does shift the D parameter but does so in almost the same way between parton and particle level, and thus, one obtains a very similar calibration curve and a very very small systematic uncertainty due to this effect (~0.1%). This is not unexpected as the madspin lineshape uncertainty behaves similarly to this at parton-particle level, as stated in our previous responses. However, there are some significant caveats to our study, which also serve to highlight why making a fair comparison to the CMS results is not trivial (or possible, with the information contained in the CMS note).
- Firstly, the definition of a “top” is unclear in these models. One or both of the tops must be off-shell in a topoponium scenario and, therefore, there are never two top quark four vectors written out into the LHE files event record produced by the generator (since virtual particles are not documented in the record). Our parton level definition used in this observation relies on finding these particles in the event record. We have to identify the tops to boost the leptons into their rest frames, and we require the invariant mass of the top system in order to reweight the entanglement in the build the calibration curve. How CMS tackled this problem and how it affects their result is not stated in their conference note. This also means that the two calibration curves, shown in our figure, are not technically identical. The parton-level definition for the orange curve is the one used in the measurement, where tops are directly identified in the event record. The blue curve is built by constructing a parton level top using the sum of a lepton, b, and neutrino, since the off-shell top itself is not present in the record. The difference between these two definitions is likely to be extremely small but it is present.
- Secondly, all of the events in a topoponium sample have $D = -1$. They are perfectly entangled and their spins are 100% correlated. Topoponium with “less entanglement” (and indeed, less spin correlation) is not something easily defined. Though our reweighting procedure that we use to build the calibration curve can technically be performed on these events, it is unclear how meaningful that would be. It is not stated how CMS handled this with their sample, since they mix samples containing SM spin correlated tops and uncorrelated tops (and, presumably, topoponium should be entirely absent from the uncorrelated sample).

- Thirdly, the sample we have created mixes SM $t\bar{t}$ and topoponium, with some ratio of cross-sections. We understand that CMS took a similar approach, based on their preliminary note. We have a reasonable idea what this ratio of cross-sections should be in the SM case, but we have no idea if this assumption should hold were entanglement to be 'changed'. One possibility is that if the tops weren't entangled, the topoponium cross-section should get smaller, probably to the point of vanishing. Another would be to fix the ratio of the cross-sections to keep them constant. We have no guidance for how to scale this (e.g. what should the cross-section at our reweighting point of -60% entanglement). Again, it is not stated how CMS have done this. It should be stated that with different assumptions of this cross-section ratio, we could change the calibration curve. However, without guidance of how to do this correctly, it isn't obvious that this would be a meaningful result.
- In summary, we performed (our interpretation of) the study requested by the reviewer, but it does not cover the factor of two that the reviewer determined to be the point of reasonable agreement. However, the study did behave how we expected it to based on the behaviour of similar systematic uncertainties. We highlight that a precision measurement of D is not the goal of this observation and, as the reviewer agrees, no matter what approach is taken to address topoponium, it does not affect the main observation statement of the paper.

In the following you can find more detailed comments to your responses.

“We agree that the MC does not contain any bound state effects ... it does not alter the significance of the result.”

-> I am not debating that the significance will remain above 5 sigma. Clearly and as you point out yourself later, even a larger uncertainty does not change the fact the you observe entanglement. However, you also quote a result on D, and that is my concern. For the preliminary result from CMS, toponium is the leading uncertainty. This alone - to me - justifies a quantification for your measurement - or to refrain from claiming a (precise) measurement.

- We agree with the reviewer and do not claim to be making a ‘precision’ measurement in the paper of D. Statements to that effect, or words that could be interpreted in that way,

never appear in the text. On the contrary, the paper was written specifically to avoid making such claims. For example, at the end of the discussion section we explicitly state “the current precision of the measurements in the validation regions does not allow us to rule out any of the MC setups that were used” as well as “It is important to note that close to the threshold, non-relativistic QCD processes, such as Coulomb bound state effects, affect the production of $t\bar{t}$ events and are not accounted for in the MC generators”. Furthermore, at the end of section A.6, the concluding paragraph directly states that there are issues to be addressed if we wish to make future quantum information measurements: “The procedure used in MC event generators to combine the matrix element with a parton-shower algorithm requires special attention in future higher-precision quantum information studies at the LHC”.

We therefore suggest that we have been very careful in quantifying the validity of this measurement and in not overstating the sensitivity or the implications of it. We don't believe any further qualifications or refrains are necessary.

“The figures in the paper you refer to ... we don't think that it is possible to do this at present and we don't believe it would significantly change the conclusions of this paper.”

-> The rapidity difference and top mass distribution are distributions the authors of that paper picked. Concluding that it is *only* these distributions that are affected and - since you are not using these exact distributions in your analysis (or strongly correlated ones) - the impact must be negligible is not valid.

- We certainly don't claim that these are the only comparisons to make and we apologise if we have given this impression. We were attempting to say that these comparisons were the most relevant ones we actually can make based on the information in that particular paper.

“We contacted the authors of 2102.11281 [...] By necessity, this result had to come first to induce such developments. [...] it is not possible for us to use this model as it is not publicly available.”

-> Indeed, a more detailed model of the effects would be very useful and it is great that your result is stimulating these developments. That paper, however, came out way before this article was submitted. Whether it is publicly available or not, the paper showed that there is an impact in exactly the difficult region you are using for your measurement. If there could be an effect, then the best possible attempt needs to be made to estimate it, and assign an uncertainty to it when measurements are performed in the affected region of phase space. The availability of code has - strictly speaking - no scientific relevance. In the meantime we learned that your attempt may underestimate the effect (admittedly

not by a factor of 20 but possibly still by a factor of 4-5), and that such a simulation can be run.

- Once again, we do not agree that there is any evidence that we are underestimating the effect. The statement that CMS may be overestimating the effect (or, more likely, that their analysis method makes them more susceptible to it) is equally valid. The CMS preliminary result methodology is very different from our observation and it is different in almost every aspect of the experimental methodology. The invariant mass range is different, the selection cuts (such as the beta cut) are different, the way topoponium has been estimated is different, the way that D is extracted (contrary to other comments from the reviewer, this does matter and has a large effect on the impact of systematics) is entirely different. The reviewer's implication on an equivalency between the results is not justified in our opinion.

[on the CMS uncertainties]. Thank you for the clarification. Indeed, that is a very suboptimal way to present the uncertainties. However, whether it is relative or not, it does remain the leading uncertainty and therefore there is still a strong motivation to also study it in context of your measurement.

- We agree. However, we also highlight again that CMS's result doesn't contain the same uncertainties as ours. They do not include an uncertainty on the lineshape of the $t\bar{t}$ invariant mass, whereas we do, and these two uncertainties are almost certainly covering similar effects from a detector response perspective. And that result is *our* leading uncertainty.

“Profile Likelihood fits can be more sensitive to systematic uncertainties than detector-corrected results“

-> It is irrelevant for the review here, but this statement is not valid.

- It is not valid if we had said “are always more sensitive” but it is certainly true that detector level profile likelihood fits can be more sensitive to systematic uncertainties than detector corrected ones. The same measurement, performed with a reconstructed level template fit, vs an unfolded template fit, vs a reco level profile likelihood fit, vs a calibration curve correction would have the same central value but entirely different systematic uncertainty behaviour. The optimisation of such choices is one of the key aspects of experimental physics and, in the context of this result, could easily explain the differences between the ATLAS observation and CMS preliminary result.

“Finally, If we were to use CMS's bound state effect uncertainty ... [still] above 5 sigma“

-> As I already pointed out earlier, I am not questioning this. However, if I naively add the 2.2% to your total uncertainty on D, its uncertainty does change significantly; and that is my concern.

- It does not change the uncertainty in such a way as to alter the conclusions of the paper. The central message and result of this article is the statement of observation, as the reviewer notes. We cannot make this statement without presenting the measurement of D with its uncertainty, but these values are not the main result of the paper and are not claimed to be so.

Regarding the correlations, thank you for the detailed explanation. Either way it was a minor comment and in addition I am fully satisfied with the answer.

To summarise, I am - personally - not fully satisfied with the answers, yet. I hope that my proposal to resolve my hesitation does imply a lot of additional work or delay, as also I would be happy to see this important result being published in a timely manner. As the other referees seem to be satisfied, I will also accept if the editor decides to overrule my input. I just personally think that such a clearly important result to appear in a high-quality journal actually deserves a high amount of scrutiny, also taking into account recent developments.

- We agree with the sentiments of the reviewer and are grateful at the time and effort they have clearly invested in the review.

Reviewer 3 comments

In the following I will only comment on the concerns raised by referee 1 and the replies provided by the authors of the submitted draft. I would further would like to stress that I am not an expert on toponium and can only give my personal impression on the discussion. I also do not possess any information on the mentioned CMS result.

It is my understanding that both referee 1 and the authors agree that there are open theory questions concerning the correct description of $t\bar{t}b\bar{b}$ production in the threshold region, which is the region of interest for the presented draft. To my understanding the authors argue convincingly that these uncertainties do not affect the significance of their result.

I also agree with the authors that it is very difficult to comment on a preliminary result by another collaboration. I nevertheless find the arguments of the authors convincing. I therefore believe that the larger uncertainties reported by the CMS experiment should not be an obstacle for publication of this article.

Dear reviewer,

We thank you once again for your comments and your continued endorsement of the result.

With best regards,

The Authors.